# MTNL: A Unified Modeling Perspective for Enhancing Tensor Network Learning

Junhua Zeng [1]   Yuning Qiu [2]   Binghua Li [3]   Chao Li [2]   Qibin Zhao [2]   Guoxu Zhou [4]

## Abstract

Over the years, the unsupervised and supervised learning research directions of tensor networks (TNs) have mainly developed in parallel. In this paper, we provide a view for their cooperative advancement through a novel mixed tensor network learning (MTNL) framework that unifies the two fields. Specifically, inspired by supervised TN learning tasks, multiple TNs are fused in a deep-network style in MTNL, enhancing the expressive power for the unsupervised TN learning tasks. We then develop a more flexible TN structure search prior with theoretical guarantees for learning multiple TN structures, aligning with trends in many supervised learning setups. More interestingly, by combining these components within a Bayesian framework, we show that MTNL induces a lightweight uncertainty quantification mechanism that is theoretically guaranteed by its connection to the dropout-based counterpart problem, making the mechanism a potential alternative for large-scale learning problems. Finally, we demonstrate the effectiveness of the MTNL framework on tensor recovery, parameter-efficient fine-tuning, and tensor regression experiments.

## 1. Introduction

Tensor network (TN) provides a powerful framework for addressing high-dimensional problems. Over the years, TNs have been applied across various domains (Anandkumar et al., 2014; Novikov et al., 2015; Cichocki et al., 2015; Sidiropoulos et al., 2017; Kossaifi et al., 2020; Panagakis

et al., 2021; Loeschcke et al., 2024; Wang et al., 2024; Nguyen et al., 2024). Their main advantage lies in representing a high-order tensor as a network of low-order tensors connected through contraction (Cichocki et al., 2016; 2017). This approach mitigates the curse of dimensionality when handling high-order tensors directly.

TNs are typically used under two main problem setups. In unsupervised learning, TNs mostly aim to approximate a given high-dimensional tensor data with its low-rank TN components in order to perform tensor analysis, recovery, and so on. Commonly used TNs include tensor train (TT) (Oseledets, 2011; Bengua et al., 2017), tensor ring (TR) (Zhao et al., 2016; Wang et al., 2017; Yuan et al., 2019), and fully-connected tensor network (FCTN) (Zheng et al., 2021). Recently, growing attention has been given to tensor network structure search (TN-SS) (Li & Sun, 2020), which aims to identify the optimal TN structure for a given task. Theoretically, TN-SS reduces both variance and bias in TN learning, making it essential for effective TN modeling (Nie et al., 2021b; Li et al., 2023; Liu et al., 2024).

Supervised learning setup leverages the multi-dimensional modeling capability of TNs to capture high-order correlations in the training data, thereby enhancing learning performance while reducing the number of model parameters (Novikov et al., 2015; Hou et al., 2019; Kossaifi et al., 2020). One representative example is deep learning (DL) model compression (Wang et al., 2018; Pan et al., 2019; Panagakis et al., 2021; Li et al., 2022b; Wang et al., 2023). Recently, parameter-efficient fine-tuning (PEFT) of large models has gained significant attention (Hu et al., 2022; Zanella & Ben Ayed, 2024; Tao et al., 2025). TN-based PEFT methods further compress the fine-tuning parameters by exploiting the multilinear structure of pre-trained weights using TNs (Chen et al., 2024a; Bershatsky et al., 2024; Yang et al., 2024; Tao et al., 2025).

Although these two setups seem to be separated by their different learning nature, they are actually connected, and studies have shown that ideas can be borrowed between them to enhance performance. Specifically, in tensor recovery, the concepts of depth and nonlinearity have motivated the development of deeper (Fan & Cheng, 2018; Milanesi et al., 2021; Fan, 2022) or nonlinear (Zhou et al., 2023) decom-

---

[1]Generative Machine Learning Lab, College of Computer Science, Inner Mongolia University, Hohhot, China [2]RIKEN-AIP, Tokyo, Japan [3]Department of Electrical Engineering and Computer Science, Tokyo University of Agriculture and Technology, Tokyo, Japan [4]School of Automation, Guangdong University of Technology, Guangzhou, China. Correspondence to: Guoxu Zhou <gx.zhou@gdut.edu.cn>.

*Proceedings of the 43$^{rd}$ International Conference on Machine Learning*, Seoul, South Korea. PMLR 306, 2026. Copyright 2026 by the author(s).

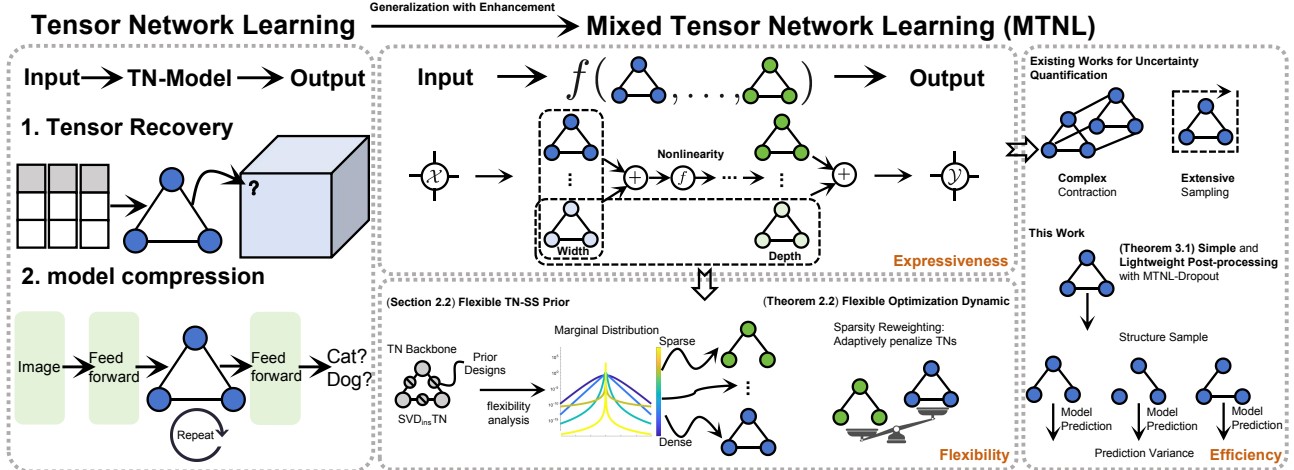

*Figure 1.* The overall structure of MTNL.

position models to boost recovery performance. Moreover, the latent factor decomposition model (Nie et al., 2021a) also shares a similar spirit with the mixture-of-experts architecture in the large language model (LLM) (Fedus et al., 2022). From the other perspective, advances in tensor recovery for identifying more suitable TN structures have also contributed to a higher model compression level (Yin et al., 2021; Li et al., 2021; Hawkins & Zhang, 2021; Dai et al., 2023).

However, hindered by the modeling gap between these two fields, existing integration of ideas may lead to sub-optimal performance. First, most existing TN-SS methods are designed to search for a single TN structure (Li et al., 2023; Zheng et al., 2024), which may lack the flexibility needed to accommodate the diverse structural requirements of TN representations in model compression, where parameters at different layers may exhibit varying degrees of redundancy (Zhang et al., 2023; Ye et al., 2026). Second, while core concepts from DL have been incorporated to enhance the TN model, these attempts remain shallow and still fall short of fully exploring the insights of the modern DL framework, leaving room for further improvements.

To close the gap and enhance TN learning, this paper adopts a unified modeling perspective and intends to propose a model that satisfies two essential criteria. *First, the model should incorporate the key characteristics of modern DL frameworks, such as depth, nonlinearity, and width. Second, it should offer a flexible solution to the TN-SS problem.* In this paper, the key contributions are:

- We propose the mixed tensor network learning (MTNL) that satisfies the general requirements of TN learning, specifically by developing a novel Bayesian TN learning framework that fuses multiple TNs in a deep-network style, with provable multiple TN-SS ability. Moreover,

by solving the maximum a posteriori (MAP) problem, it induces a lightweight uncertainty quantification algorithm based on model dropout, requiring only simple post-processing of the solution, given its potential to improve uncertainty quantification efficiency in large-scale TN learning. The overall structure of MTNL is illustrated in Fig. 1.

- Theoretically, we provide a detailed analysis of the flexibility of the proposed TN-SS prior in Section 2.2. Beyond the explicit TN-SS capability, we also reveal the implicit TN-SS potential of the proposed model by studying its lower-bound optimization problem, as established in Theorem 2.2. To support effective dropout-based uncertainty quantification, we establish a theoretical connection between the MAP solution of MTNL and its dropout-based optimization counterpart in Theorem 3.1.

- In experiments, we show that MTNL significantly outperforms existing tensor recovery methods. Furthermore, MTNL improves the LLM PEFT performances with more expressive TN structures. Additionally, it estimates output uncertainty with minimal overhead, substantially reducing the computation budget compared with existing methods.

### 1.1. Related Works

Our work is related to studies on TN-SS. Existing methods for solving the TN-SS can be broadly divided into three categories. The first category includes discrete optimization methods (Hayashi et al., 2019; Hashemizadeh et al., 2020; Li & Sun, 2020; Li et al., 2022a; Solgi et al., 2022; Li et al., 2023; Zeng et al., 2024a; Guo et al., 2025), which formulate and solve TN-SS as a discrete optimization problem. However, when addressing the multiple TN-SS needs, the solution space becomes extremely large, making discrete optimization difficult to converge to a satisfactory solution.

The second category of methods relies on the spectral information of tensors to solve the TN-SS problem (Nie et al., 2021b; 2023; Chen et al., 2024b). However, these methods impose structural constraints on tensor modes and ranks, which may lead to sub-optimal performance in practice (Li et al., 2022a). The proposed method belongs to the third category: continuous optimization methods (Cheng et al., 2022; Kodryan et al., 2023; Yang et al., 2023; Zheng et al., 2024; Zeng et al., 2024b). These methods impose sparsity constraints on TN-ranks to continuously learn the structure. Compared with the two former categories, they do not require sampling in the solution space and do not impose constraints on the problem, making them more suitable for solving the MTNL problem. Compared to existing continuous optimization studies, our method introduces a more flexible TN-SS mechanism for learning multiple TN structures and theoretically demonstrates its effectiveness in Section 2.

Several studies have integrated TN-SS with model compression and explored both TN-SS methods (Li et al., 2021; Yin et al., 2021; Hawkins & Zhang, 2021; Dai et al., 2023). However, these works mainly focus on optimizing TN-ranks within a fixed structure, limiting their ability to fully exploit parameter redundancy. In contrast, this work explores learning TN structures for large model PEFT and provides a theoretical analysis of the implicit structure searching capability that benefits PEFT. Moreover, as improving the reliability of TN learning methods becomes increasingly important (Xia et al., 2025) and existing algorithms face scaling issues (Hawkins & Zhang, 2021), this work sheds light on a lightweight uncertainty quantification algorithm with a theoretical guarantee.

## 2. Mixed Tensor Network Learning

### 2.1. Notations

In this paper, we primarily follow the notations of (Kolda & Bader, 2009). Lowercase letters (e.g., $x$) denote scalars and vectors, while boldface capital letters (e.g., $\mathbf{X}$) represent matrices. Calligraphic letters (e.g., $\mathcal{X}$) denote higher-order tensors (order greater than 2). For sequences of tensors, we use the notation $\{\mathcal{X}_i^j\}_{i=1,j=1}^{I,J}$, where subscripts $i$ and $j$ enumerate the elements from 1 to $I$ and 1 to $J$, respectively. Specifically, $\{\mathcal{X}_i^j\}_{i=1,j=1}^{I,J} := \{\mathcal{X}_1^1, \ldots, \mathcal{X}_I^J\}$. Matrix and vector sequences are represented analogously.

### 2.2. Model

In this subsection, we introduce the modeling details of MTNL. Firstly, input and output data pairs of the model are represented as $\{\mathcal{X}_i, \mathcal{Y}_i\}_{i=1}^N, \mathcal{X}_i \in \mathbb{R}^{I_1 \times I_2 \times \cdots \times I_d}, \mathcal{Y}_i \in \mathbb{R}^{O_1 \times O_2 \times \cdots \times O_c}$. Given the MTNL model illustrated in Fig. 1, we assume for notational simplicity that the model consists of $L$ layers, each containing $E$ TNs with the same

structure. However, our results can be easily extended to more general cases. To model each TN, in this paper, we borrow the SVD-inspired form of TN decomposition (Zheng et al., 2024). As we discuss later, this form is beneficial for introducing the flexible TN-SS prior. Therefore, the $e$-th TN at the $l$-th layer is composed of tensor cores $\{\mathcal{G}_i^{(l,e)}\}_{i=1}^T$, where $\mathcal{G}_i^{(l,e)} \in \mathbb{R}^{R_{1,i}^{(l,e)} \times \cdots \times R_{i-1,i}^{(l,e)} \times J_i^{(l,e)} \times R_{i,i+1}^{(l,e)} \times \cdots \times R_{i,T}^{(l,e)}}$, and also the threshold matrices $\{\mathbf{S}_{k_1,k_2}^{(l,e)}\}_{1 \le k_1 < k_2 \le T}$, where $\mathbf{S}_{k_1,k_2}^{(l,e)} = \text{diag}(s_{k_1,k_2}^{(l,e)}), s_{k_1,k_2}^{(l,e)} \in \mathbb{R}^{R_{k_1,k_2}^{(l,e)}}$. Here, we assume that all TNs in the MTNL have the same order $T$, though the results can be easily extended to more general cases. To incorporate the model structure with input-output data pair, the order of the TN must satisfy $T \ge \max(d, c)$. Moreover, for the first layer, $J_1^{(1,e)} = I_1, J_2^{(1,e)} = I_2, \ldots, J_d^{(1,e)} = I_d, e = 1, \cdots, E$. For the last layer, $J_{T-c+1}^{(L,e)} = O_1, J_{T-c+2}^{(L,e)} = O_2, \ldots, J_T^{(L,e)} = O_c, e = 1, \cdots, E$. For the middle layers, the dimension patterns of the TNs are as follow: for the second layer, the first $T - d$ dimensions of the TNs are equal to the last $T - d$ dimensions of the TNs in the first layer, and for the third layer, the first $d$ dimensions of the TNs are equal to the last $d$ dimensions of the TNs in the second layer, and the pattern alternates similarly in following layers.

The high-level idea conveyed by the above dimensional constraint is to ensure that contractions between layers of the MTNL can be performed. As long as contractions can be performed, the resulting model is considered an MTNL model.

Given a data pair $\{\mathcal{X}_i, \mathcal{Y}_i\}$ and the parameters set $\Theta = \{\{\mathcal{G}_i^{(l,e)}\}_{i=1}^T, \{\mathbf{S}_{k_1,k_2}^{(l,e)}\}_{1 \le k_1 < k_2 \le T}\}_{e=1,l=1}^{E,L}$, the forward propagation process of the MTNL is expressed as follows:

$$
\mathcal{Y}_i = \mathcal{F}(\mathcal{X}_i; \Theta) = f^L(\{\mathcal{G}_i^{(L,e)}\}, \{\mathbf{S}_{k_1,k_2}^{(L,e)}\}, \overbrace{\underbrace{\cdots}_{\times L}}^{\text{forward}} \quad (1)
$$
$$
f^1(\{\mathcal{G}_i^{(1,e)}\}, \{\mathbf{S}_{k_1,k_2}^{(1,e)}\}, \mathcal{X}_i) \cdots).
$$

In Eq. 1, we omit the subscripts on the braces to simplify the expression. The function $f^l(\cdot)$ implements the $l$-th layer's forward propagation by: contracting the input feature with the weight TN represented by $\{\mathcal{G}_i^{(l,e)}\}_{i=1}^T$ and $\{\mathbf{S}_{k_1,k_2}^{(l,e)}\}_{1 \le k_1 < k_2 \le T}$ for $e = 1, \ldots, E$, separately, summing the results, and applying an activation. It is worth noting that the MTNL model in Eq. 1 is a general TN learning model that includes the tensor recovery as a special case by considering the input $\mathcal{X}$ as the outer product of basis vectors representing the tensor element's position. For example, given a three-order tensor, the element value at location $(2, 3, 4)$ is 10, then the input tensor $\mathcal{X} = e_2 \circ e_3 \circ e_4$, output tensor (scalar) $y = 10$, and $e_i$ is the basic vector.

**Likelihood.** Given the training data pairs and the MTNL

model, the likelihood is given by:

$$p(\mathcal{Y} \mid \mathcal{X}, \Theta) = \prod_{i=1}^{N} p(\mathcal{Y}_i \mid \mathcal{X}_i, \Theta), \qquad (2)$$

where $\mathcal{Y}$ and $\mathcal{X}$ are tensors that respectively stack output and input tensors together. For the likelihood function $p(\mathcal{Y}_i \mid \mathcal{X}_i, \Theta)$, it is flexible enough to model different types of learning problems, including regression problems, classification problems, and so on. Taking the regression problem as an example, we can define $p(\mathcal{Y}_i \mid \mathcal{X}_i, \Theta) = \mathcal{N}(\text{vec}(\mathcal{Y}_i) \mid \text{vec}(\mathcal{F}(\mathcal{X}_i; \Theta)), \epsilon^{-1}\mathbf{I}_{|\text{vec}(\mathcal{Y}_i)|})$, where $|\text{vec}(\mathcal{Y}_i)|$ denotes the dimension of the vector, and $\epsilon$ is the noise precision.

**Prior and hyperprior.** In this paragraph, we introduce the prior settings of the MTNL model. For TN cores $\mathcal{G}_i^{(l,e)}, i = 1, \ldots T, l = 1, \ldots L, e = 1, \ldots E$, its prior is given as follow:

$$p(\text{vec}(\mathcal{G}_i^{(l,e)})) = \mathcal{N}\left(\text{vec}(\mathcal{G}_i^{(l,e)}) \mid 0, (\tau_i^{(l,e)})^{-1}\mathbf{I}_{|\text{vec}(\mathcal{G}_i^{(l,e)})|}\right),$$
$$(3)$$

where $\tau_i^{(l,e)}$ is the hyper-parameter. We adopt the Gaussian prior here to avoid overfitting.

To be able to learn the structure of the TN, similarly to existing study, we need to constrain the sparsity level of the vector $s_{k_1,k_2}^{(l,e)}, 1 \le k_1 < k_2 \le T, l = 1, \ldots L, e = 1, \ldots E$, which serves as the threshold for the TN-ranks. However, in the existing study (Zheng et al., 2024), the authors only use the $\ell_1$-norm to induce sparsity, which may lead to sub-optimal results when handling a large network with many TNs exhibiting different sparsity patterns. Therefore, in this paper, we intend to place a more flexible sparsity-inducing prior over vector $s_{k_1,k_2}^{(l,e)}$. The prior over $s_{k_1,k_2}^{(l,e)}$ is given by:

$$p(s_{k_1,k_2}^{(l,e)} \mid \lambda_{k_1,k_2}^{(l,e)}) = \prod_{i=1}^{R_{k_1,k_2}^{(l,e)}} \mathcal{N}(s_{k_1,k_2}^{(l,e)}(i) \mid 0, \lambda_{k_1,k_2}^{(l,e)}(i)).$$
$$(4)$$

In Eq. 4, $\lambda_{k_1,k_2}^{(l,e)} \in \mathbb{R}^{R_{k_1,k_2}^{(l,e)}}$ is the hyperparameter of the distribution, and the hyperprior of $\lambda_{k_1,k_2}^{(l,e)}$ is shown as follow:

$$p(\lambda_{k_1,k_2}^{(l,e)}) = \prod_{i=1}^{R_{k_1,k_2}^{(l,e)}} \text{GIG}(\lambda_{k_1,k_2}^{(l,e)}(i) \mid a_{k_1,k_2}^{(l,e)}(i), b_{k_1,k_2}^{(l,e)}(i),$$
$$c_{k_1,k_2}^{(l,e)}(i)),$$
$$(5)$$

where $a_{k_1,k_2}^{(l,e)}, b_{k_1,k_2}^{(l,e)}, c_{k_1,k_2}^{(l,e)} \in \mathbb{R}^{R_{k_1,k_2}^{(l,e)}}$ are the hyperparameters. Moreover, $\text{GIG}(x \mid a,b,c) = \frac{\left(\frac{a}{b}\right)^{c/2}}{2K_c(\sqrt{ab})} x^{c-1} \exp\left(-\frac{1}{2}\left(ax + bx^{-1}\right)\right)$ is the generalized inverse Gaussian (GIG) distribution, and $K_{\cdot}(\cdot)$ is the modified Bessel function of the second kind.

To investigate the sparsity-inducing ability of the above Gaussian-GIG prior, we marginalize $\lambda_{k_1,k_2}^{(l,e)}$ to obtain the marginal distribution $p(s_{k_1,k_2}^{(l,e)})$ as follows:

$$p(s_{k_1,k_2}^{(l,e)}(i)) = \int p(s_{k_1,k_2}^{(l,e)}(i) \mid \lambda_{k_1,k_2}^{(l,e)}(i)) p(\lambda_{k_1,k_2}^{(l,e)}(i)) d_{\lambda(i)}$$

$$= C \frac{K_{c_{k_1,k_2}^{(l,e)}(i)-\frac{1}{2}}\left(\sqrt{a_{k_1,k_2}^{(l,e)}(i)(b_{k_1,k_2}^{(l,e)}(i) + s_{k_1,k_2}^{(l,e)}(i)^2)}\right)}{(b_{k_1,k_2}^{(l,e)}(i) + s_{k_1,k_2}^{(l,e)}(i)^2)^{\frac{1}{2} - \frac{c_{k_1,k_2}^{(l,e)}(i)}{2}}}.$$
$$(6)$$

$p(s_{k_1,k_2}^{(l,e)}) = \prod_{i=1}^{R_{k_1,k_2}^{(l,e)}} p(s_{k_1,k_2}^{(l,e)}(i))$ is a multivariate generalized hyperbolic distribution with shape control by hyperparameters $\{a_{k_1,k_2}^{(l,e)}, b_{k_1,k_2}^{(l,e)}, c_{k_1,k_2}^{(l,e)}\}$, and $C$ is the coefficient. By setting the hyperparameters to different values, the generalized hyperbolic distribution can reduce to sparsity-inducing distributions that exhibit a wide range of central and tail behaviors (Thabane & Safiul Haq, 2004; Babacan et al., 2014), including the Laplacian distribution (equivalently, the $\ell_1$ penalty (Zheng et al., 2024)). Thus, it can provide a more flexible TN-SS capability.

Although recent work has also explored the combination of Gaussian–GIG for learning the structure of TN (Zeng et al., 2024b), it directly assigns priors on the tensor cores, following the common strategy in Bayesian TN learning (Long et al., 2021; Hawkins & Zhang, 2021). However, in the following theorem, we provide a rigorous analysis of how sparsity is actually imposed on the tensor cores and highlight a potential flaw in this modeling framework for learning TN structures, thereby motivating our proposed framework.

**Theorem 2.1** (Sparsity-Inducing Effect on TN Representation in (Zeng et al., 2024b))**. *Given any $k, f = 1, \ldots, N$ and $k < f$, the model settings in (Zeng et al., 2024b) result in a sparsity-inducing distribution placed over the TN-cores $\mathcal{G}_k$ and $\mathcal{G}_f$, detailed as: (1) $p\left(\mathcal{G}_k(\ldots, r_{k-1,k}, :, r_{k,k+1}, \ldots, r_{k,f-1}, :, r_{k,f+1}, \ldots)\right) \propto$*

$$\prod_{r_{k,f}} \left(\sum_{i_k} \mathcal{G}_k^2(r_{1,k}, \ldots, i_k, \ldots, r_{k,N})\right)^{-\frac{I_k}{2}}; \qquad (2)$$
*$p(\mathcal{G}_f(\ldots, r_{k-1,f}, :, r_{k+1,f}, \ldots, r_{f-1,f}, :, r_{f,f+1}, \ldots)) \propto$*

$$\prod_{r_{k,f}} \left(\sum_{i_f} \mathcal{G}_f^2(r_{1,f}, \ldots, i_f, \ldots, r_{f,N})\right)^{-\frac{I_f}{2}}.$$
*These hold for any choice of the rank indices $r_{1,k}, \ldots, r_{k,N}, r_{1,f}, \ldots, r_{f,N}$ when $a^{(k,f)} \to 0$, $b^{(k,f)} \to 0$, and $c^{(k,f)} \to 0$.*

The proof of the theorem is provided in the Appendix B. From Theorem 2.1, we can observe that a sparsity penalty is indeed imposed on the rank $R_{k,f}$, but it is separately applied to the two corresponding tensor cores and across other rank indices $r_{1,k}, \ldots, r_{k,N}, r_{1,f}, \ldots, r_{f,N}$. However, ideally, the penalty should be applied jointly, which is equivalent to constraining the sparsity of the combined slices over the rank $R_{k,f}$ across both cores.

In comparison, through the introduction of the threshold vector, the proposed framework achieves this desired effect and provides a theoretically more consistent formulation, demonstrating the theoretical advantages of the proposed Bayesian TN modeling framework.

**Log joint distribution.** Existing studies on Bayesian TN learning typically need to explicitly compute the posterior distribution, which can be computationally expensive when estimating uncertainty. To accelerate uncertainty quantification in TN learning, we propose an efficient two-step algorithm. The main computational cost of the method lies in solving the MAP problem in the first step, while the second step enables lightweight uncertainty estimation through simple post-processing of the trained model, as demonstrated in Section 3. The minimization objective of the MAP problem is shown as follows:

$$
\min_{\Theta, \{\lambda_{k_1,k_2}^{(l,e)}\}} -\log(p(\mathcal{Y} \mid \mathcal{X}, \Theta)) - \sum_{l,e,i} \log(p(\text{vec}(\mathcal{G}_i^{(l,e)})))
$$
$$
- \sum_{l,e,k_1,k_2} \left( \log(p(s_{k_1,k_2}^{(l,e)} \mid \lambda_{k_1,k_2}^{(l,e)})) + \log(p(\lambda_{k_1,k_2}^{(l,e)})) \right).
$$
(7)

### 2.3. Application in Parameter-efficient Fine-tuning (PEFT)

In this subsection, we introduce the application of MTNL to PEFT, with a specific focus on the Low-Rank Adaptation (LoRA) method as demonstrated below.

$$
h = \mathbf{W}_0 a + \Delta \mathbf{W} a = \mathbf{W}_0 a + \mathbf{B}\mathbf{A}a. \tag{8}
$$

Here, fine-tuning $\mathbf{B} \in \mathbb{R}^{O \times r}$ and $\mathbf{A} \in \mathbb{R}^{r \times I}$, with $r \ll \min(O, I)$, contains far fewer parameters compared to fine-tune pre-trained weight $\mathbf{W}_0$, making LoRA an efficient method for PEFT.

However, matrix decomposition may not fully exploit the higher-order information in the weight matrix, resulting in sub-optimal performance. To overcome this limitation, recent studies have explored tensor decomposition to enhance LoRA performance (Hu et al., 2025; Anjum et al., 2024; Chen et al., 2024a; Bershatsky et al., 2024; Tao et al., 2025). These approaches decompose the higher-order transformation perturbation matrix using fixed TN structures. Although promising, reliance on fixed TN architectures may limit the effectiveness of TN-based PEFT. To address this issue, we have the following model:

$$
\mathcal{H}^l = \mathcal{W}_0^l \mathcal{A}^l + \Delta \mathcal{W}^l \mathcal{A}^l = \mathcal{W}_0^l \mathcal{A}^l + f^l(\{\{\mathcal{G}_i^{(l,e)}\}_{i=1}^T,
$$
$$
\{\mathbf{S}_{k_1,k_2}^{(l,e)}\}_{1 \le k_1 < k_2 \le T}\}_{e=1}^E, \mathcal{A}^l).
$$
(9)

Eq. 9 illustrates the $l$-th layer of the fine-tuning model, where $f^l(\cdot)$ uses a sum of TNs to represent the fine-tuning

weights contracted with the layer input $\mathcal{A}^l$. Under this formulation, the proposed MTNL model can be viewed as a prototype for the final fine-tuning model, providing the theoretical mechanism for structure learning. Compared to other TN-based PEFT methods, our method not only adaptively learns suitable TN structures from the data but also introduces a latent factor TN representation to the perturbation matrix. This design enhances the effectiveness of TN-based methods for high-rank fine-tuning. Beyond the sparsity prior, Theorem 2.2 reveals an interesting structure learning mechanism of the proposed MTNL, aligning well with the practical TN-SS needs of PEFT (Zhang et al., 2023; Ye et al., 2026).

**Theorem 2.2** (Implicit Structure Learning in MTNL). *Given a TN representation of this paper, we can express it in a normalized rank-one representation format (Zhao et al., 2016) by summing the weighted outer product of its normalized mode fibers, denoted as $\mathcal{U}_i^{(l,e)}(r_{1,i}^{(l,e)}, \ldots, r_{i-1,i}^{(l,e)}, :, r_{i,i+1}^{(l,e)}, \ldots, r_{i,T}^{(l,e)})$, and weights tensor $\mathcal{Z}^{(l,e)}(r_{1,2}^{(l,e)}, \ldots, r_{T-1,T}^{(l,e)}) = \prod_{i=1}^T g_{i_{r_{1,i}^{(l,e)}, \ldots, r_{i-1,i}^{(l,e)}, r_{i,i+1}^{(l,e)}, \ldots, r_{i,T}^{(l,e)}}}^{(l,e)}$. Based on the normalized rank-one form, the MTNL model can be reformulated as $\mathcal{F}(\cdot, \Theta^1)$, where $\Theta^1 = \{\mathcal{Z}^{(l,e)}, \{\mathcal{U}_i^{(l,e)}\}_{i=1}^T, \{\mathbf{S}_{k_1,k_2}^{(l,e)}\}_{1 \le k_1 < k_2 \le T}\}_{e=1,l=1}^{E,L}$, and $\mathcal{U}_i^{(l,e)}$ denotes the mode-fiber-wise normalized core tensor. The lower bound optimization problem of Eq. 7 is shown as follow:*

$$
\min_{\Theta^1, \{\lambda_{k_1,k_2}^{(l,e)}\}} -\log(p(\mathcal{Y} \mid \mathcal{X}, \Theta^1)) + \frac{\mu}{2} \prod_{l,e} \|\text{vec}(\mathcal{Z}^{(l,e)})\|_{2/T}
$$
$$
- \text{term}_3,
$$
(10)

*where* $\text{term}_3$ *denotes the last term of Eq. 7 and $\| \cdot \|_{2/T}$ is the $\ell_{2/T}$ norm of a vector. By optimizing Eq. 7, it implicitly optimizes Eq. 10 with $\mathcal{G}_i^{(l,e)}(r_{1,i}^{(l,e)}, \ldots, r_{i-1,i}^{(l,e)}, :, r_{i,i+1}^{(l,e)}, \ldots, r_{i,T}^{(l,e)}) = C_i^{(l,e)}(\mathcal{Z}^{(l,e)}(r_{1,2}^{(l,e)}, ., r_{T-1,T}^{(l,e)}))^{1/T} \mathcal{U}_i^{(l,e)}(r_{1,i}^{(l,e)}, ., r_{i-1,i}^{(l,e)}, :, r_{i,i+1}^{(l,e)}, \ldots, r_{i,T}^{(l,e)})$, where $C_i^{(l,e)}$ is a constant related to the dimension of the TN.*

The proof of the theorem is provided in the Appendix C. From the lower bound optimization model of MTNL presented in Theorem 2.2, we observe that MTNL implicitly enforces sparsity on the rank-one tensors of each TN by applying the $\ell_{2/T}$ norm to the elements of the coefficient tensor $\mathcal{Z}^{(l,e)}$. Since the number of rank-one tensors in a TN corresponds to the product of its TN-ranks, this constraint effectively promotes learning a compact TN structure. Moreover, Theorem 2.2 reveals that MTNL introduces a sparsity-reweighting mechanism across different layers of TNs through an equivalent transformation of the regularization term: $\prod_{l,e} \|\text{vec}(\mathcal{Z}^{(1,e)})\|_{2/T} \rightarrow$

$\sum_l w_l \prod_e \|\text{vec}(\mathcal{Z}^{(l,e)})\|_{2/T}, w_l = \frac{1}{L} \prod_{i=1, i \neq l}^{L} \prod_e \|\text{vec}(\mathcal{Z}^{(i,e)})\|_{2/T}$. This transformation implies that TNs in layers with lower ranks are adaptively subjected to higher penalties. As demonstrated in (Zhang et al., 2023; Ye et al., 2026), certain fine-tuning layers are significantly more suitable for compression than others, which aligns with the proposed model's adaptive structure learning ability.

## 3. Fast Uncertainty Quantification Algorithm

In this section, we will introduce a fast uncertainty quantification algorithm that requires only lightweight post-processing after solving the optimization problem in Eq. 7. Before going into the details of the algorithm, we first define dropout TN based on the TN formulation used in this paper.

For $l = 1, \ldots, L, e = 1, \ldots, E, 1 \leq k_1 < k_2 \leq T$, we define random matrix $\mathbf{V}_{k_1,k_2}^{(l,e)} = \text{diag}(v_{k_1,k_2}^{(l,e)})$, where $v_{k_1,k_2}^{(l,e)} \in \mathbb{R}^{R_{k_1,k_2}^{(l,e)}}$ is a random vector whose elements are i.i.d Bernoulli($\theta$). Dropout TN is a randomized TN that simultaneously contracts the threshold matrix $\mathbf{S}_{k_1,k_2}^{(l,e)}$ and the random matrix $\mathbf{V}_{k_1,k_2}^{(l,e)}$ on a given rank $R_{k_1,k_2}^{(l,e)}$. Taking matrix decomposition $x_{i,j} = \sum_{k,b} a_{i,k} s_{k,b} b_{b,j}$ as an example, its dropout form is expressed as $x_{i,j} = \sum_{k,b} a_{i,k} v_{k,b} s_{k,b} b_{b,j}$. After defining the dropout TN, we have the following theorem.

**Theorem 3.1** (Implicit Dropout Learning). *Based on the dropout TN, we refer to a dropout MTNL model whenever using the model parameters $\widetilde{\Theta} = \{\{\mathbf{V}_{k_1,k_2}^{(l,e)}\}_{1 \leq k_1 < k_2 \leq T}, \{\mathcal{G}_i^{(l,e)}\}_{i=1}^{T}, \{\mathbf{S}_{k_1,k_2}^{(l,e)}\}_{1 \leq k_1 < k_2 \leq T}\}_{e,l=1}^{E,L}$. If we drop the tilde and omit the random matrices, using $\Theta$ instead, we refer to the standard MTNL model. Moreover, in this theorem, we consider only the linearized MTNL model with identical activation. In the following equation, we demonstrate that the dropout variant of the MAP problem in Eq. 7 has an upper bound.*

$$- \mathbb{E}_{[\mathbf{V}]} \left[ \log(p(\mathcal{Y} \mid \mathcal{X}, \tilde{\Theta})) \right] \quad (11)$$

$$- \sum_{l,e,k_1,k_2} \left( \log(p(s_{k_1,k_2}^{(l,e)} \mid \lambda_{k_1,k_2}^{(l,e)})) + \log(p(\lambda_{k_1,k_2}^{(l,e)})) \right)$$

$$(12)$$

$$\leq -\log(p(\mathcal{Y} \mid \mathcal{X}, \Theta)) + \sum_{l,e,i} k_i^{(l,e)}(\theta) \|\mathcal{G}_i^{(l,e)}\|_F^2 \quad (13)$$

$$+ \sum_{l,e,k_1,k_2} b_{k_1,k_2}^{(l,e)}(\theta) \frac{1}{2\min(\lambda_{k_1,k_2}^{(l,e)})} \|\mathbf{S}_{k_1,k_2}^{(l,e)}\|_F^2 - \log(p(\lambda_{k_1,k_2}^{(l,e)})).$$

$$(14)$$

*Here, $k_i^{(l,e)}(\theta)$ and $b_{k_1,k_2}^{(l,e)}(\theta) \in [0, +\infty]$ are functions controlled by the parameter $0 \leq \theta \leq 1$, and they are not related to the optimization variables of the problem.*

---

**Algorithm 1** MTNL-Dropout

**Input:** The optimal results $\Theta_{\text{MAP}}$ to Eq. 7, the testing data $\mathcal{X}_{\text{test}}$, dropout rate $\theta$, sample number $N$.

**Initialization:** The dropout MTNL model $\mathcal{F}_{\text{Drop}}(\cdot, \{\Theta_{\text{MAP}}, \{\{\mathbf{V}_{k_1,k_2}^{(l,e)}\}_{1 \leq k_1 < k_2 \leq T}\}_{l=1,e=1}^{L,E}\})$.

$Y \leftarrow [\,]$.

**for** $i = 1$ **to** $N$ **do**

   1. Sample $\{\{\mathbf{V}_{k_1,k_2}^{(l,e)}\}_{1 \leq k_1 < k_2 \leq T}\}_{l=1,e=1}^{L,E}$ according to dropout rate $\theta$.

   2. $Y \leftarrow Y \bigcup [$prediction of the dropout MTNL with $\mathcal{X}_{\text{test}}$ and $\{\{\mathbf{V}_{k_1,k_2}^{(l,e)}\}_{1 \leq k_1 < k_2 \leq T}\}_{l=1,e=1}^{L,E}]$.

**end for**

**Output:** Uncertainty: variance($Y$).

---

The proof of the theorem is provided in the Appendix D. As we can see from Eq. 12 and Eq. 11, it is a dropout MTNL with regularization. The upper bound of this problem is in Eq. 14 and Eq. 13, and as we expand Eq. 7, they are similar. Moreover, since the coefficients $k_i^{(l,e)}(\theta)$ and $b_{k_1,k_2}^{(l,e)}(\theta) \in [0, +\infty]$, by appropriately choosing $\theta$, the optimization of the MAP problem implicitly optimizes a dropout-based problem. It is worth noting that the coefficient of $\mathbf{S}_{k_1,k_2}^{(l,e)}$ in Eq. 7 is actually lower bounded by $\frac{1}{2\max(\lambda_{k_1,k_2}^{(l,e)})}$. When some ranks of the TN are unnecessary, the difference between $\max(\lambda_{k_1,k_2}^{(l,e)})$ and $\min(\lambda_{k_1,k_2}^{(l,e)})$ can be several orders of magnitude. In such a case, we may have to choose $\theta = 1$, which makes the result meaningless. However, as is common in the Bayesian tensor learning, we can prune the ranks dynamically during training. In that case, the resulting model's $\lambda_{k_1,k_2}^{(l,e)}$ values will be of similar magnitude, allowing $\theta$ within a reasonable range.

Thus, this finding motivates an MC-dropout-like uncertainty quantification approach (Gal & Ghahramani, 2016) in MTNL. Specifically, after acquiring $\Theta_{\text{MAP}}$, we sample several different dropout MTNL models $\mathcal{F}_{\text{Drop}}(\cdot, \{\Theta_{\text{MAP}}, \{\{\mathbf{V}_{k_1,k_2}^{(l,e)}\}_{1 \leq k_1 < k_2 \leq T}\}_{l=1,e=1}^{L,E}\})$ and combine their predictions for efficient uncertainty quantification. The pseudo-code is provided in Alg. 1.

## 4. Experiments

In this section, we conduct experiments on tensor recovery, PEFT of LLM, and tensor regression to validate the effectiveness of the proposed MTNL framework. Additional details are given in Appendix H.

### 4.1. Tensor Recovery Experiments

In this subsection, we conduct real-world tensor completion experiments on four multispectral images (MSIs) from the

*Table 1.* Comparison of tensor recovery performance across different methods on four MSIs. In the table, the cells corresponding to a missing rate of 95% are shaded in gray for better visualization.

| MSI | TRALS | | BCPF | | BTucker | | BTT | | TW | | BayesTNSS | | MTNL | |
|---|---|---|---|---|---|---|---|---|---|---|---|---|---|---|
| | 85% | 95% | 85% | 95% | 85% | 95% | 85% | 95% | 85% | 95% | 85% | 95% | 85% | 95% |
| **MSI1** | 0.0412 | 0.1016 | 0.0553 | 0.1052 | 0.0458 | 0.0939 | 0.0360 | 0.0872 | 0.0353 | 0.0984 | 0.0278 | 0.0779 | **0.0251** | **0.0758** |
| **MSI2** | 0.0544 | 0.1232 | 0.0489 | 0.0956 | 0.0471 | 0.1067 | 0.0401 | 0.0804 | 0.0453 | 0.1013 | 0.0356 | 0.0813 | **0.0330** | **0.0773** |
| **MSI3** | 0.0925 | 0.3599 | 0.0822 | 0.2744 | 0.1046 | 0.4752 | 0.0889 | 0.1899 | 0.0868 | 0.2200 | 0.0732 | 0.1968 | **0.0617** | **0.1828** |
| **MSI4** | 0.0709 | 0.1739 | 0.0817 | 0.1495 | 0.0702 | 0.1569 | 0.0573 | 0.1267 | 0.0610 | 0.1740 | 0.0494 | 0.1314 | **0.0458** | **0.1214** |
| **Average** | 0.0648 | 0.1897 | 0.0670 | 0.1562 | 0.0669 | 0.2082 | 0.0556 | 0.1211 | 0.0571 | 0.1484 | 0.0465 | 0.1219 | **0.0414** | **0.1143** |

*Table 2.* Illustration of the LLM fine-tuning results. In the table, #Params indicates the proportion of fine-tuning model parameters relative to the full-tuning model. Results for other methods are from (Chen et al., 2024a).

| Model | FT | MoRA | LoRETTA | QuanTA | $MTNL_l$ | MTNL |
|---|---|---|---|---|---|---|
| $F_1$ **Score** (↑) | 59.4 | 58.9 | 59.1 | 59.5 | **60.1** | 59.6 |
| **#Params** (↓) | 100% | 0.996% | 1.254% | 0.041% | 0.044% | 0.053% |

CAVE dataset (Yasuma et al., 2010) and compare the recovery performance of MTNL against other tensor completion methods. Six methods are considered for comparison: (1) TRALS (Wang et al., 2017), a TR-based completion method using alternating least squares optimization; (2) BCPF (Zhao et al., 2015), Bayesian low-rank CP completion; (3) BTucker (Tong et al., 2023), Bayesian low-rank Tucker completion; (4) BTT (Xu et al., 2023), Bayesian low-rank TT completion; (5) TW (Wu et al., 2022), Tensor Wheel (TW) completion; (6) BayesTNSS (Zeng et al., 2024b), a continuous Bayesian TN-SS method for completion. For all comparison methods, we fine-tuned their hyperparameters to achieve the best results.

Furthermore, we consider two missing rates, 85% and 95%, and use the relative standard error (RSE) over the missing entries as the evaluation metric. For MTNL, we set the structure as $E = 2, L = 2$. In addition, we incorporate skip connections (He et al., 2016) into each layer of the model to improve gradient flow. In Appendix G, we further show that the theoretical analyses presented in this paper can be extended to this structural enhancement

The completion results are shown in Table 1. From the table, we can observe a notable performance gap between the proposed MTNL and other TN completion methods. Specifically, even though the comparison methods already fully utilize the representation power of TNs, the proposed MTNL, which integrates concepts from both TN and DL, still demonstrates superior recovery performance. These results highlight that, *compared with traditional tensor recovery methods, MTNL is capable of enhancing TN learning performance with a more expressive model structure*. Be-

sides the DL interpretation, MTNL also admits a generalized TN recovery model interpretation, where elements such as nonlinearity are added between TN cores to increase the expressiveness, as discussed in Appendix F. Moreover, MTNL also shares a similar intact optimization form with other TN baselines, largely improving the practicality of the MTNL method (see Appendix H.2).

### 4.2. Parameter-efficient Fine-tuning (PEFT) Experiments

In this subsection, we aim to evaluate the LLM PEFT performance of the proposed method. Specifically, we fine-tune LLaMA2-7B (Touvron et al., 2023) on the DROP dataset (Chen et al., 2024a) and compare its performance with other (tensor-based) fine-tuning methods.

Table 2 presents the results of this experiment. The result of $MTNL_l$ corresponds to applying a rank-cutting threshold of $10^{-1}$ to the final MTNL model and then retraining the searched model. As shown, $MTNL_l$ achieves better performance than the original MTNL, indicating that a more compact structure is learned. Furthermore, compared to QuanTA, the structure learned by $MTNL_l$ exhibits similar parameter complexity but yields a significant improvement in the $F_1$ score. Additionally, as we analyze the statistics of the learned TN parameters used to represent the $Q$ and $V$ matrices in Appendix H.3, we find that our method consistently allocates more parameters to $V$ than to $Q$, which aligns with findings in (Hu et al., 2022; Ye et al., 2026), supporting the implicit structure learning capability of MTNL (Theorem 2.2). Overall, these experiments demonstrate that *the MTNL framework can enhance (tensor-based) PEFT performance with more expressive TN structures*.

### 4.3. Tensor Regression Experiments

In the regression experiment, we aim to evaluate the uncertainty quantification capability of the MTNL-dropout in responding to distribution shifts in the test data. First, we generate a training set with inputs as $20 \times 20$ matrices independently drawn from a Gaussian distribution, and we

*Table 3.* Change in prediction mean squared error (upper rows) and variance (lower rows) under different levels of distribution shift. Training time and average uncertainty estimation time are shown below each model.

| Model | Distribution Shift | | | |
|---|---|---|---|---|
| $(E, L)$ | 0 | 3 | 6 | 9 |
| **(4,4)** | 0.0224 | 0.0288 | 0.0992 | 0.2273 |
| 36.0/**0.3** | 0.0001 | 0.0002 | 0.0008 | 0.0017 |
| **(5,5)** | 0.3859 | 1.8563 | 6.1476 | 14.5465 |
| 54.8/**0.5** | 0.0004 | 0.0015 | 0.0062 | 0.0161 |

*Table 4.* NLL achieved by SVGD with $n$ times higher computational cost than the proposed method. Values in brackets show the proposed method's NLL.

| Cost | 120 | 160 | 200 | 240 | 280 |
|---|---|---|---|---|---|
| **NLL** ($\downarrow$) [1.984] | 1634.914 | 1295.855 | 1369.410 | 9.836 | **-0.083** |

*Table 5.* Ablation results for TN-SS components. Each method achieves $\text{RSE}<10^{-5}$. The learned parameters, the iterations required (in square brackets, $\times 1000$), and the original model parameters (in round brackets) are shown.

| Method | Proposed | w/o core prior | w/o threshold prior |
|---|---|---|---|
| **D=6** (300) | **178** [**655**] | 248 [3215] | 280 [6795] |
| **D=8** (384) | **328** [**460**] | 338 [5455] | 454 [8125] |

forward-propagate them through the MTNL model to obtain the labels. For testing, we create four test sets drawn from the same Gaussian distribution but with means shifted by 0, 3, 6, and 9. In this experiment, we consider two MTNL models with $(E, L) = (4, 4)$ and $(E, L) = (5, 5)$. Since the number of training samples is smaller than the number of model parameters, the models tend to overfit, making it increasingly difficult to extrapolate to distributionally shifted test data. This setting provides an effective test bed for assessing the uncertainty quantification ability of the proposed MTNL-dropout.

In Table 3, we present the results of this experiment. For both models, the MSE increases as the test data distribution shifts further, and the unreliable predictions are correspondingly captured by MTNL-dropout through increasing predictive variance. Moreover, as shown in the table, the uncertainty quantification time accounts for only 8%-9% of the total model training time, demonstrating the efficiency of the MTNL-dropout algorithm.

Furthermore, we evaluate the uncertainty quantification performance of the proposed algorithm on large-scale problems with $64 \times 64$ inputs and compare it with the SVGD method (Hawkins & Zhang, 2021), using the negative log-likelihood (NLL) metric (Lind et al., 2024). The results are demonstrated in Table 4. As shown, the proposed al-

gorithm achieves comparable performance to SVGD while reducing computational cost by up to $200\times$, demonstrating its strong practical potential. These tensor regression experiments indicate that *MTNL-dropout is an efficient and resource-friendly uncertainty quantification algorithm for TN learning*.

### 4.4. Ablations

In this experiment, we conduct ablation studies to verify the effectiveness of different TN-SS components. Specifically, we generate synthetic tensors using the MTNL model and train an overparameterized MTNL on the tensors. Finally, we compare the learned model parameters with those of the tensor-generating model. Here, we consider three-order tensors with mode dimensions set to 6 and 8. The experimental results are presented in Table 5. From the table, we can observe that the proposed method successfully learns a compact model structure from the data. When each TN-SS component is removed from the model, the TN-SS performance begins to drop, demonstrating the importance of each component. On the other hand, as seen from the result of the w/o threshold prior, the model still retains TN-SS capability, which verifies the implicit structure learning ability of the MTNL. This also helps to explain the accelerated convergence observed when the proposed method and the w/o core prior variant achieve similar performance.

## 5. Conclusion

In this paper, we propose MTNL, a unified framework that bridges two major research directions in TN learning. This framework provides a more systematic perspective for addressing key challenges in existing TN models, including expressiveness, flexibility, and efficiency. Specifically, MTNL meets two core requirements of general TN learning through non-trivial extensions of the model and the prior over existing studies. By introducing a novel Bayesian TN modeling framework, MTNL enables flexible learning of multiple TN structures, supported by explicit prior distribution analysis and the implicit structure learning theorem. Furthermore, MTNL establishes the implicit dropout learning theorem, which forms the basis for an efficient and lightweight uncertainty quantification algorithm. Through extensive experiments across various tasks, we demonstrate that MTNL enhances the expressiveness of TN learning and improves tensor recovery performance, enhances PEFT with more expressive TN structures, and provides efficient uncertainty quantification in predictions. We hope that this work can inspire new perspectives in the design of TN-based models in the future.

**Limitation.** When unsupervised and supervised TN learning meet in this work, they inspire new model design degrees of freedom that are shown to be important for TN learning.

However, this also increases the method's complexity, and how to make this method more efficient, for example, by exploring more effective optimization methods, will be the focus of our future work.

## Impact Statement

This paper presents work focusing on using TN to represent high-dimensional problems. It has the potential to reduce computational cost and even energy consumption in tackling such problems. Beyond this, we do not identify any other impacts that must be specifically highlighted here.

## Acknowledgements

We appreciate the anonymous (meta-)reviewers for their helpful comments, and Junhua is grateful to Yumeng Ma for her valuable discussions throughout this work. The work was supported in part by the Inner Mongolia University High-level Talent Project through grant 10000-A25206022, in part by the Guangdong Natural Science Foundation under Grant 2024A1515010114, in part by JSPS KAKENHI (Grant No. JP24K20849, JP24K03005, JP25KJ1215 and JP23K28109), and in part by the JSPS Bilateral Program Number JPJSBP120257420.

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

# A. Basic Formulation

In this section, we provide explicit forms for several important formulations of the work that were not included in the main text.

First, we provide an explicit form for the MAP problem presented in Eq. 7 of the main text.

$$
\min_{\left\{\{\mathcal{G}_i^{(l,e)}\}_{i=1}^T, \{\mathbf{S}_{k_1,k_2}^{(l,e)}\}_{1\le k_1 < k_2 \le T}, \{\lambda_{k_1,k_2}^{(l,e)}\}_{1\le k_1 < k_2 \le T}\right\}_{l=1,e=1}^{L,E}}
$$

$$
- \log(p(\mathcal{Y} \mid \mathcal{X}, \Theta))
$$

$$
+ \sum_{l,e,t}^{L,E,T} \frac{\tau_t^{(l,e)}}{2} \left\| \mathcal{G}_t^{(l,e)} \right\|_F^2
$$

$$
+ \sum_{l,e}^{L,E} \mathrm{Sum}\left( s_{1,2}^{(l,e)} \circledast s_{1,2}^{(l,e)} \circledast \left( 1/\left(2\lambda_{1,2}^{(l,e)}\right)\right)\right) + \cdots + \mathrm{Sum}\left( s_{T-1,T}^{(l,e)} \circledast s_{T-1,T}^{(l,e)} \circledast \left( 1/\left(2\lambda_{T-1,T}^{(l,e)}\right)\right)\right)
$$

$$
+ \sum_{l,e}^{L,E} \mathrm{Sum}\left( \frac{1}{2}\log\left(\lambda_{1,2}^{(l,e)}\right) - \left(c_{1,2}^{(l,e)} - 1\right) \circledast \log\left(\lambda_{1,2}^{(l,e)}\right) + \frac{1}{2}\left( a_{1,2}^{(l,e)} \circledast \lambda_{1,2}^{(l,e)} + b_{1,2}^{(l,e)} \circledast \left(\lambda_{1,2}^{(l,e)}\right)^{-1}\right)\right)
$$

$$
+ \cdots + \mathrm{Sum}\left( \frac{1}{2}\log\left(\lambda_{T-1,T}^{(l,e)}\right) - \left(c_{T-1,T}^{(l,e)} - 1\right) \circledast \log\left(\lambda_{T-1,T}^{(l,e)}\right) + \frac{1}{2}\left( a_{T-1,T}^{(l,e)} \circledast \lambda_{T-1,T}^{(l,e)} + b_{T-1,T}^{(l,e)} \circledast \left(\lambda_{T-1,T}^{(l,e)}\right)^{-1}\right)\right).
$$

(15)

Here, $\mathrm{Sum}(\cdot)$ denotes the summation over the elements of a vector, and $\circledast$ represents the element-wise product of two vectors. Additionally, all operations over vectors in the last three terms of Eq. 15 are performed element-wise. The first term in Eq. 15 represents the problem-specific loss function for tensor learning problems. For the tensor completion problem, this loss term is expressed as: $\sum_{i=1}^N \frac{\epsilon}{2}\|y_i - \mathcal{F}(\mathcal{X}_i; \Theta)\|_F^2$, where $N$ denotes the number of observed elements, $\mathcal{X}_i$ is the coordinate tensor indicating the location of the observed element $y_i$. Furthermore, to optimize the above problem, a straightforward approach is to utilize the automatic differentiation tool along with gradient descent available in modern deep learning frameworks. Alternatively, we can employ a mixed optimization strategy: the loss term can be optimized using gradient-based methods, while the regularization terms admit a closed-form solution to accelerate convergence.

In the following, we review the SVD-inspired TN representation form (Zheng et al., 2024) for completeness.

$$
\begin{aligned}
\mathcal{X}(i_1, i_2, \cdots, i_N) = &\sum_{r_{1,2}=1}^{R_{1,2}} \sum_{r_{1,3}=1}^{R_{1,3}} \cdots \sum_{r_{1,N}=1}^{R_{1,N}} \sum_{r_{2,3}=1}^{R_{2,3}} \cdots \sum_{r_{2,N}=1}^{R_{2,N}} \cdots \sum_{r_{N-1,N}=1}^{R_{N-1,N}} \\
& \mathbf{S}_{1,2}(r_{1,2}, r_{1,2}) \mathbf{S}_{1,3}(r_{1,3}, r_{1,3}) \cdots \mathbf{S}_{1,N}(r_{1,N}, r_{1,N}) \\
& \mathbf{S}_{2,3}(r_{2,3}, r_{2,3}) \cdots \mathbf{S}_{2,N}(r_{2,N}, r_{2,N}) \cdots \\
& \mathbf{S}_{N-1,N}(r_{N-1,N}, r_{N-1,N}) \\
& \mathcal{G}_1(i_1, r_{1,2}, r_{1,3}, \cdots, r_{1,N}) \\
& \mathcal{G}_2(r_{1,2}, i_2, r_{2,3}, \cdots, r_{2,N}) \cdots \\
& \mathcal{G}_k(r_{1,k}, r_{2,k}, \cdots, r_{k-1,k}, i_k, r_{k,k+1}, \cdots, r_{k,N}) \cdots \\
& \mathcal{G}_N(r_{1,N}, r_{2,N}, \cdots, r_{N-1,N}, i_N),
\end{aligned}
$$

(16)

In notation, the SVD-inspired TN representation is also denoted as $\mathcal{X} = \mathrm{SVD}_{ins}\mathrm{TN}(\{\mathcal{G}_i\}_{i=1}^N, \{\mathbf{S}_{k_1,k_2}\}_{1\le k_1 < k_2 \le N})$.

In the main text, we propose several extensions to the above TN formulation, including the normalized rank-one format (Theorem 2.2) and the Dropout format (Theorem 3.1). Here, we provide detailed element-wise expressions of these extensions based on Eq. 16, which will be useful for the subsequent proofs of these theorems.

First, we present an equivalent normalized rank-one format of the SVD-inspired TN in Eq. 16 as follows:

$$
\begin{aligned}
\mathcal{X} = &\sum_{r_{1,2}=1}^{R_{1,2}} \sum_{r_{1,3}=1}^{R_{1,3}} \cdots \sum_{r_{1,N}=1}^{R_{1,N}} \sum_{r_{2,3}=1}^{R_{2,3}} \cdots \sum_{r_{2,N}=1}^{R_{2,N}} \cdots \sum_{r_{N-1,N}=1}^{R_{N-1,N}} \\
& \mathcal{Z}(r_{1,2}, r_{1,3}, \ldots, r_{1,N}, r_{2,3}, \ldots, r_{2,N}, \ldots, r_{N-1,N}) \\
& \mathbf{S}_{1,2}(r_{1,2}, r_{1,2}) \mathbf{S}_{1,3}(r_{1,3}, r_{1,3}) \cdots \mathbf{S}_{1,N}(r_{1,N}, r_{1,N}) \\
& \mathbf{S}_{2,3}(r_{2,3}, r_{2,3}) \cdots \mathbf{S}_{2,N}(r_{2,N}, r_{2,N}) \cdots \\
& \mathbf{S}_{N-1,N}(r_{N-1,N}, r_{N-1,N}) \\
& \mathcal{U}_1(:, r_{1,2}, r_{1,3}, \cdots, r_{1,N}) \circ \\
& \mathcal{U}_2(r_{1,2}, :, r_{2,3}, \cdots, r_{2,N}) \circ \cdots \circ \\
& \mathcal{U}_k(r_{1,k}, r_{2,k}, \cdots, r_{k-1,k}, :, r_{k,k+1}, \cdots, r_{k,N}) \circ \cdots \circ \\
& \mathcal{U}_N(r_{1,N}, r_{2,N}, \cdots, r_{N-1,N}, :),
\end{aligned}
\tag{17}
$$

where $\mathcal{Z}(r_{1,2}, r_{1,3}, \ldots, r_{1,N}, r_{2,3}, \ldots, r_{2,N}, \ldots, r_{N-1,N}) = \prod_{i=1}^{N} g_{r_{1,i}, \ldots, r_{i-1,i}, r_{i,i+1}, \ldots, r_{i,N}}^{i}$, and $g_{r_{1,i}, \ldots, r_{i-1,i}, r_{i,i+1}, \ldots, r_{i,N}}^{i} = \|\mathcal{G}_i(r_{1,i}, r_{2,i}, \cdots, r_{i-1,i}, :, r_{i,i+1}, \cdots, r_{i,N})\|_2$. Moreover, $\mathcal{G}_i(r_{1,i}, r_{2,i}, \cdots, r_{i-1,i}, :, r_{i,i+1}, \cdots, r_{i,N}) = g_{r_{1,i}, \ldots, r_{i-1,i}, r_{i,i+1}, \ldots, r_{i,N}}^{i} \mathcal{U}_i(r_{1,i}, r_{2,i}, \cdots, r_{i-1,i}, :, r_{i,i+1}, \cdots, r_{i,N})$. In Eq. 17, there are total $R_{1,2} \times R_{1,3} \times \cdots \times R_{1,N} \times R_{2,3} \times \cdots \times R_{2,N} \times \cdots \times R_{N-1,N}$ terms of summation of the rank-one tensor. In notation, the rank-one format of the SVD-inspired TN representation is also denoted as $\mathcal{X} = \text{SVD}_{one} \text{TN}(\mathcal{Z}, \{\mathcal{U}_i\}_{i=1}^{N}, \{\mathbf{S}_{k_1,k_2}\}_{1 \le k_1 < k_2 \le N})$.

Second, given the randomized dropout matrices $\{\mathbf{V}_{k_1,k_2}\}_{1 \le k_1 < k_2 \le N}$, the Dropout format of the SVD-inspired TN in Eq. 16 is shown as follows:

$$
\begin{aligned}
\mathcal{X}(i_1, i_2, \cdots, i_N) = &\sum_{r_{1,2}=1}^{R_{1,2}} \sum_{r_{1,3}=1}^{R_{1,3}} \cdots \sum_{r_{1,N}=1}^{R_{1,N}} \sum_{r_{2,3}=1}^{R_{2,3}} \cdots \sum_{r_{2,N}=1}^{R_{2,N}} \cdots \sum_{r_{N-1,N}=1}^{R_{N-1,N}} \\
& \mathbf{S}_{1,2}(r_{1,2}, r_{1,2}) \mathbf{S}_{1,3}(r_{1,3}, r_{1,3}) \cdots \mathbf{S}_{1,N}(r_{1,N}, r_{1,N}) \\
& \mathbf{S}_{2,3}(r_{2,3}, r_{2,3}) \cdots \mathbf{S}_{2,N}(r_{2,N}, r_{2,N}) \cdots \\
& \mathbf{S}_{N-1,N}(r_{N-1,N}, r_{N-1,N}) \\
& \mathbf{V}_{1,2}(r_{1,2}, r_{1,2}) \mathbf{V}_{1,3}(r_{1,3}, r_{1,3}) \cdots \mathbf{V}_{1,N}(r_{1,N}, r_{1,N}) \\
& \mathbf{V}_{2,3}(r_{2,3}, r_{2,3}) \cdots \mathbf{V}_{2,N}(r_{2,N}, r_{2,N}) \cdots \\
& \mathbf{V}_{N-1,N}(r_{N-1,N}, r_{N-1,N}) \\
& \mathcal{G}_1(i_1, r_{1,2}, r_{1,3}, \cdots, r_{1,N}) \\
& \mathcal{G}_2(r_{1,2}, i_2, r_{2,3}, \cdots, r_{2,N}) \cdots \\
& \mathcal{G}_k(r_{1,k}, r_{2,k}, \cdots, r_{k-1,k}, i_k, r_{k,k+1}, \cdots, r_{k,N}) \cdots \\
& \mathcal{G}_N(r_{1,N}, r_{2,N}, \cdots, r_{N-1,N}, i_N),
\end{aligned}
\tag{18}
$$

In notation, the Dropout format of the SVD-inspired TN representation is also denoted as $\mathcal{X} = \text{SVD}_{drop} \text{TN}(\{\mathcal{G}_i\}_{i=1}^{N}, \{\mathbf{S}_{k_1,k_2}, \mathbf{V}_{k_1,k_2}\}_{1 \le k_1 < k_2 \le N})$.

## B. Proof of Theorem 2.1

In this section, we demonstrate the sparsity-inducing prior distribution that is actually placed over the TN-cores with the model assumptions made in (Zeng et al., 2024b). Before presenting the detailed proof of the theorem, we first review the basic model assumptions made in (Zeng et al., 2024b).

Firstly, the vanilla TN representation format is adopted as shown below:

$$
\begin{aligned}
\mathcal{X}(i_1, i_2, \cdots, i_N) = \sum_{r_{1,2}=1}^{R_{1,2}} \sum_{r_{1,3}=1}^{R_{1,3}} \cdots \sum_{r_{1,N}=1}^{R_{1,N}} \sum_{r_{2,3}=1}^{R_{2,3}} \cdots \sum_{r_{2,N}=1}^{R_{2,N}} \cdots \sum_{r_{N-1,N}=1}^{R_{N-1,N}} \\
\mathcal{G}_1(i_1, r_{1,2}, r_{1,3}, \cdots, r_{1,N}) \\
\mathcal{G}_2(r_{1,2}, i_2, r_{2,3}, \cdots, r_{2,N}) \cdots \\
\mathcal{G}_k(r_{1,k}, r_{2,k}, \cdots, r_{k-1,k}, i_k, r_{k,k+1}, \cdots, r_{k,N}) \cdots \\
\mathcal{G}_N(r_{1,N}, r_{2,N}, \cdots, r_{N-1,N}, i_N).
\end{aligned}
\tag{19}
$$

For the TN-core $\mathcal{G}_k$, it follows a Gaussian distribution of the following form:

$$
p(\mathcal{G}_k | \lambda^{(1,k)}, \ldots, \lambda^{(k,N)}) = \prod_{i_k, r_{1,k}, \ldots, r_{k,N}} \mathcal{N}(\mathcal{G}_k(r_{1,k}, \ldots, i_k, \ldots, r_{k,N}) | 0, \lambda_{r_{1,k}}^{(1,k)} \cdots \lambda_{r_{k,N}}^{(k,N)}).
\tag{20}
$$

In order to learn the TN-ranks, a hyperprior is placed on $\lambda^{(k_1, k_2)}$, as shown below:

$$
p(\lambda^{(k_1, k_2)}) = \prod_{r_{k_1, k_2}} \mathrm{GIG}(\lambda_{r_{k_1, k_2}}^{(k_1, k_2)} | a_{r_{k_1, k_2}}^{(k_1, k_2)}, b_{r_{k_1, k_2}}^{(k_1, k_2)}, c_{r_{k_1, k_2}}^{(k_1, k_2)}).
\tag{21}
$$

In the following, we provide theoretical results demonstrating how the model assumptions above are capable of enforcing sparsity over the TN-cores.

**Theorem B.1** (Sparsity-Inducing Effect on TN Representation in (Zeng et al., 2024b))**.** *Given any $k, f = 1, \ldots, N$ and $k < f$, the model settings in (Zeng et al., 2024b) result in a sparsity-promoting distribution placed over the TN-cores $\mathcal{G}_k$ and $\mathcal{G}_f$, detailed as: (1) $p\left(\mathcal{G}_k(r_{1,k}, \ldots, r_{k-1,k}, :, r_{k,k+1}, \ldots, r_{k,f-1}, :, r_{k,f+1}, \ldots, r_{k,N})\right) \propto \prod_{r_{k,f}} \left(\sum_{i_k} \mathcal{G}_k^2(r_{1,k}, \ldots, i_k, \ldots, r_{k,N})\right)^{-\frac{I_k}{2}}$; (2) $p\left(\mathcal{G}_f(r_{1,f}, \ldots, r_{k-1,f}, :, r_{k+1,f}, \ldots, r_{f-1,f}, :, r_{f,f+1}, \ldots, r_{f,N})\right) \propto \prod_{r_{k,f}} \left(\sum_{i_f} \mathcal{G}_f^2(r_{1,f}, \ldots, i_f, \ldots, r_{f,N})\right)^{-\frac{I_f}{2}}$. These hold for any choice of the rank indices $r_{1,k}, \ldots, r_{k,N}, r_{1,f}, \ldots, r_{f,N}$ when $a^{(k,f)} \to 0$, $b^{(k,f)} \to 0$, and $c^{(k,f)} \to 0$.*

*Proof.* Firstly, by multiplying the distributions of the TN-cores in Eq. 20 and $\lambda^{(k_1, k_2)}$ in Eq. 21, we obtain their joint probability distribution as follows:

$$
p(\mathcal{G}_1, \ldots, \mathcal{G}_N, \lambda^{(1,2)}, \ldots, \lambda^{(N-1,N)}) = \prod_{k=1}^{N} p(\mathcal{G}_k | \lambda^{(1,k)}, \ldots, \lambda^{(k,N)}) \prod_{k_1=1}^{N-1} \prod_{k_2=k_1+1}^{N} p(\lambda^{(k_1, k_2)}).
\tag{22}
$$

Then, we integrate out all the TN-cores except $\mathcal{G}_k$ and obtain the joint probability distribution of $\mathcal{G}_k$ and all the $\lambda^{(k_1, k_2)}$ as follows:

$$
p(\mathcal{G}_k, \lambda^{(1,2)}, \ldots, \lambda^{(N-1,N)}) = p(\mathcal{G}_k | \lambda^{(1,k)}, \ldots, \lambda^{(k,N)}) \prod_{k_1=1}^{N-1} \prod_{k_2=k_1+1}^{N} p(\lambda^{(k_1, k_2)}).
\tag{23}
$$

Similarly, we can integrate out all the $\lambda^{(k_1, k_2)}$ that are irrelevant to $\mathcal{G}_k$ and obtain the following joint distribution:

$$
p(\mathcal{G}_k, \lambda^{(1,k)}, \ldots, \lambda^{(k,N)}) = p(\mathcal{G}_k | \lambda^{(1,k)}, \ldots, \lambda^{(k,N)}) p(\lambda^{(1,k)}) \cdots p(\lambda^{(k,N)}).
\tag{24}
$$

Based on Eq. 24, we further integrate out all the elements of $\mathcal{G}_k$ except $\mathcal{G}_k(r_{1,k}, \ldots, r_{k-1,k}, :, r_{k,k+1}, \ldots, r_{k,f-1}, :, r_{k,f+1}, \ldots, r_{k,N})$ and obtain:

$$
\begin{aligned}
&p(\mathcal{G}_k(r_{1,k}, \ldots, r_{k-1,k}, :, r_{k,k+1}, \ldots, r_{k,f-1}, :, r_{k,f+1}, \ldots, r_{k,N}), \lambda^{(1,k)}, \ldots, \lambda^{(k,N)}) \\
&= \prod_{i_k, r_{k,f}} \mathcal{N}(\mathcal{G}_k(r_{1,k}, \ldots, i_k, \ldots, r_{k,N}) | 0, \lambda_{r_{1,k}}^{(1,k)} \cdots \lambda_{r_{k,N}}^{(k,N)}) p(\lambda^{(1,k)}) \cdots p(\lambda^{(k,N)}).
\end{aligned}
\tag{25}
$$

Following Eq. 25, for $\lambda^{(1,k)}, \ldots, \lambda^{(k,N)}$ except $\lambda^{(k,f)}$, we can further integrate out their elements except the $r_{1,k}, \ldots, r_{k,N}$-th elements, respectively, and obtain the following joint distribution:

$$
\begin{aligned}
&p(\mathcal{G}_k(r_{1,k}, \ldots, r_{k-1,k}, :, r_{k,k+1}, \ldots, r_{k,f-1}, :, r_{k,f+1}, \ldots, r_{k,N}), \lambda^{(1,k)}_{r_{1,k}}, \ldots, \lambda^{(k,N)}_{r_{k,N}}, \lambda^{(k,f)}) \\
&= \prod_{i_k, r_{k,f}} \mathcal{N}(\mathcal{G}_k(r_{1,k}, \ldots, i_k, \ldots, r_{k,N})|0, \lambda^{(1,k)}_{r_{1,k}} \cdots \lambda^{(k,N)}_{r_{k,N}}) p(\lambda^{(1,k)}_{r_{1,k}}) \cdots p(\lambda^{(k,N)}_{r_{k,N}}) p(\lambda^{(k,f)}) \\
&\propto p(\lambda^{(1,k)}_{r_{1,k}}) \cdots p(\lambda^{(k,N)}_{r_{k,N}}) \prod_{i_k, r_{k,f}} \frac{1}{\sqrt{\lambda^{(1,k)}_{r_{1,k}} \cdots \lambda^{(k,N)}_{r_{k,N}}}} \exp\left\{ -\frac{1}{2\lambda^{(1,k)}_{r_{1,k}} \cdots \lambda^{(k,N)}_{r_{k,N}}} \mathcal{G}^2_k(r_{1,k}, \ldots, r_{k,N}) \right\} \\
&\quad \prod_{r_{k,f}} \lambda^{(k,f)^{c^{(k,f)}_{r_{k,f}} - 1}}_{r_{k,f}} \exp\left\{ -\frac{1}{2}(a^{(k,f)}_{r_{k,f}} \lambda^{(k,f)}_{r_{k,f}} + b^{(k,f)}_{r_{k,f}} \lambda^{(k,f)^{-1}}_{r_{k,f}}) \right\} \\
&\propto p(\lambda^{(1,k)}_{r_{1,k}}) \cdots p(\lambda^{(k,N)}_{r_{k,N}}) \prod_{r_{k,f}} \lambda^{(1,k)^{-\frac{I_k}{2}}}_{r_{1,k}} \cdots \lambda^{(k,N)^{-\frac{I_k}{2}}}_{r_{k,N}} \exp\left\{ -\frac{1}{2\lambda^{(1,k)}_{r_{1,k}} \cdots \lambda^{(k,N)}_{r_{k,N}}} (\sum_{i_k} \mathcal{G}^2_k(r_{1,k}, \ldots, r_{k,N})) \right\} \\
&\quad \prod_{r_{k,f}} \lambda^{(k,f)^{c^{(k,f)}_{r_{k,f}} - 1}}_{r_{k,f}} \exp\left\{ -\frac{1}{2}(a^{(k,f)}_{r_{k,f}} \lambda^{(k,f)}_{r_{k,f}} + b^{(k,f)}_{r_{k,f}} \lambda^{(k,f)^{-1}}_{r_{k,f}}) \right\} \\
&\propto \lambda^{(1,k)^{-\frac{R_{k,f} I_k}{2}}}_{r_{1,k}} p(\lambda^{(1,k)}_{r_{1,k}}) \cdots \lambda^{(k,N)^{-\frac{R_{k,f} I_k}{2}}}_{r_{k,N}} p(\lambda^{(k,N)}_{r_{k,N}}) \prod_{r_{k,f}} \lambda^{(k,f)^{-\frac{I_k}{2} + c^{(k,f)}_{r_{k,f}} - 1}}_{r_{k,f}} \\
&\quad \exp\left\{ -\frac{1}{2}(a^{(k,f)}_{r_{k,f}} \lambda^{(k,f)}_{r_{k,f}} + (b^{(k,f)}_{r_{k,f}} + \frac{1}{\lambda^{(1,k)}_{r_{1,k}} \cdots \lambda^{(k,N)}_{r_{k,N}}} (\sum_{i_k} \mathcal{G}^2_k(r_{1,k}, \ldots, r_{k,N}))) \lambda^{(k,f)^{-1}}_{r_{k,f}}) \right\}.
\end{aligned}
\tag{26}
$$

Then, utilizing the property of the GIG distribution, we can integrate out the random variable $\lambda^{(k,f)}$ and obtain the following distribution:

$$
\begin{aligned}
&p(\mathcal{G}_k(r_{1,k}, \ldots, r_{k-1,k}, :, r_{k,k+1}, \ldots, r_{k,f-1}, :, r_{k,f+1}, \ldots, r_{k,N}), \lambda^{(1,k)}_{r_{1,k}}, \ldots, \lambda^{(k,N)}_{r_{k,N}}) \\
&\propto \lambda^{(1,k)^{-\frac{R_{k,f} I_k}{2}}}_{r_{1,k}} p(\lambda^{(1,k)}_{r_{1,k}}) \cdots \lambda^{(k,N)^{-\frac{R_{k,f} I_k}{2}}}_{r_{k,N}} p(\lambda^{(k,N)}_{r_{k,N}}) \\
&\quad \prod_{r_{k,f}} \frac{K_{c^{(k,f)}_{r_{k,f}} - \frac{I_k}{2}}\left( \sqrt{a^{(k,f)}_{r_{k,f}}} \sqrt{b^{(k,f)}_{r_{k,f}} + \frac{1}{\lambda^{(1,k)}_{r_{1,k}} \cdots \lambda^{(k,N)}_{r_{k,N}}}(\sum_{i_k} \mathcal{G}^2_k(r_{1,k}, \ldots, r_{k,N}))} \right)}{(b^{(k,f)}_{r_{k,f}} + \frac{1}{\lambda^{(1,k)}_{r_{1,k}} \cdots \lambda^{(k,N)}_{r_{k,N}}}(\sum_{i_k} \mathcal{G}^2_k(r_{1,k}, \ldots, r_{k,N})))^{\frac{I_k}{4} - \frac{c^{(k,f)}_{r_{k,f}}}{2}}}.
\end{aligned}
\tag{27}
$$

When $a^{(k,f)} \to 0$, Eq. 27 becomes the following (Babacan et al., 2014):

$$
\begin{aligned}
&p(\mathcal{G}_k(r_{1,k}, \ldots, r_{k-1,k}, :, r_{k,k+1}, \ldots, r_{k,f-1}, :, r_{k,f+1}, \ldots, r_{k,N}), \lambda^{(1,k)}_{r_{1,k}}, \ldots, \lambda^{(k,N)}_{r_{k,N}}) \\
&\propto \lambda^{(1,k)^{-\frac{R_{k,f} I_k}{2}}}_{r_{1,k}} p(\lambda^{(1,k)}_{r_{1,k}}) \cdots \lambda^{(k,N)^{-\frac{R_{k,f} I_k}{2}}}_{r_{k,N}} p(\lambda^{(k,N)}_{r_{k,N}}) \\
&\quad \prod_{r_{k,f}} \left( b^{(k,f)}_{r_{k,f}} + \frac{1}{\lambda^{(1,k)}_{r_{1,k}} \cdots \lambda^{(k,N)}_{r_{k,N}}} \left( \sum_{i_k} \mathcal{G}^2_k(r_{1,k}, \ldots, r_{k,N}) \right) \right)^{c^{(k,f)}_{r_{k,f}} - \frac{I_k}{2}}.
\end{aligned}
\tag{28}
$$

When $b^{(k,f)} \to 0$ and $c^{(k,f)} \to 0$, since we can separate the remaining random variables $\lambda$ from $\mathcal{G}_k(r_{1,k}, \ldots, r_{k-1,k}, :, r_{k,k+1}, \ldots, r_{k,f-1}, :, r_{k,f+1}, \ldots, r_{k,N})$, we can then easily obtain the distribution of the TN-core as:

$$
p(\mathcal{G}_k(r_{1,k}, \ldots, r_{k-1,k}, :, r_{k,k+1}, \ldots, r_{k,f-1}, :, r_{k,f+1}, \ldots, r_{k,N})) \propto \prod_{r_{k,f}} \left( \sum_{i_k} \mathcal{G}^2_k(r_{1,k}, \ldots, r_{k,N}) \right)^{-\frac{I_k}{2}}.
\tag{29}
$$

By keeping the same hyperparameter settings, $a^{(k,f)} \to 0$, $b^{(k,f)} \to 0$, and $c^{(k,f)} \to 0$, we continue to analyze the prior distribution that will be placed on the TN-core $\mathcal{G}_f$.

By following a similar process as for $\mathcal{G}_k$, we can obtain the joint distribution of $\mathcal{G}_f$ and its corresponding $\lambda$ as:

$$p(\mathcal{G}_f, \lambda^{(1,f)}, \ldots, \lambda^{(f,N)}) = p(\mathcal{G}_f | \lambda^{(1,f)}, \ldots, \lambda^{(f,N)}) p(\lambda^{(1,f)}) \cdots p(\lambda^{(f,N)}). \tag{30}$$

Then, we further integrate out the elements of $\mathcal{G}_f$ except $\mathcal{G}_f(r_{1,f}, \ldots, r_{k-1,f}, :, r_{k+1,f}, \ldots, r_{f-1,f}, :, r_{f,f+1}, \ldots, r_{f,N})$ and obtain:

$$
\begin{aligned}
&p(\mathcal{G}_f(r_{1,f}, \ldots, r_{k-1,f}, :, r_{k+1,f}, \ldots, r_{f-1,f}, :, r_{f,f+1}, \ldots, r_{f,N}), \lambda^{(1,f)}, \ldots, \lambda^{(f,N)}) \\
&= \prod_{i_f, r_{k,f}} \mathcal{N}(\mathcal{G}_f(r_{1,f}, \ldots, i_f, \ldots, r_{f,N}) | 0, \lambda_{r_{1,f}}^{(1,f)} \cdots \lambda_{r_{f,N}}^{(f,N)}) p(\lambda^{(1,f)}) \cdots p(\lambda^{(f,N)}).
\end{aligned} \tag{31}
$$

Furthermore, for $\lambda^{(1,f)}, \ldots, \lambda^{(f,N)}$ except $\lambda^{(k,f)}$, we can further integrate out their elements except the $r_{1,f}, \ldots, r_{f,N}$-th elements, respectively, and obtain the following joint distribution:

$$
\begin{aligned}
&p(\mathcal{G}_f(r_{1,f}, \ldots, r_{k-1,f}, :, r_{k+1,f}, \ldots, r_{f-1,f}, :, r_{f,f+1}, \ldots, r_{f,N}), \lambda_{r_{1,f}}^{(1,f)}, \ldots, \lambda_{r_{f,N}}^{(f,N)}, \lambda^{(k,f)}) \\
&= \prod_{i_f, r_{k,f}} \mathcal{N}(\mathcal{G}_f(r_{1,f}, \ldots, i_f, \ldots, r_{f,N}) | 0, \lambda_{r_{1,f}}^{(1,f)} \cdots \lambda_{r_{f,N}}^{(f,N)}) p(\lambda_{r_{1,f}}^{(1,f)}) \cdots p(\lambda_{r_{f,N}}^{(f,N)}) p(\lambda^{(k,f)}) \\
&\propto p(\lambda_{r_{1,f}}^{(1,f)}) \cdots p(\lambda_{r_{f,N}}^{(f,N)}) \prod_{i_f, r_{k,f}} \frac{1}{\sqrt{\lambda_{r_{1,f}}^{(1,f)} \cdots \lambda_{r_{f,N}}^{(f,N)}}} \exp\left\{ -\frac{1}{2\lambda_{r_{1,f}}^{(1,f)} \cdots \lambda_{r_{f,N}}^{(f,N)}} \mathcal{G}_f^2(r_{1,f}, \ldots, r_{f,N}) \right\} \\
&\quad \prod_{r_{k,f}} \lambda_{r_{k,f}}^{(k,f)^{c_{r_{k,f}}^{(k,f)}-1}} \exp\left\{ -\frac{1}{2}(a_{r_{k,f}}^{(k,f)} \lambda_{r_{k,f}}^{(k,f)} + b_{r_{k,f}}^{(k,f)} \lambda_{r_{k,f}}^{(k,f)^{-1}}) \right\} \\
&\propto p(\lambda_{r_{1,f}}^{(1,f)}) \cdots p(\lambda_{r_{f,N}}^{(f,N)}) \prod_{r_{k,f}} \lambda_{r_{1,f}}^{(1,f)^{-\frac{I_f}{2}}} \cdots \lambda_{r_{f,N}}^{(f,N)^{-\frac{I_f}{2}}} \exp\left\{ -\frac{1}{2\lambda_{r_{1,f}}^{(1,f)} \cdots \lambda_{r_{f,N}}^{(f,N)}} (\sum_{i_f} \mathcal{G}_f^2(r_{1,f}, \ldots, r_{f,N})) \right\} \\
&\quad \prod_{r_{k,f}} \lambda_{r_{k,f}}^{(k,f)^{c_{r_{k,f}}^{(k,f)}-1}} \exp\left\{ -\frac{1}{2}(a_{r_{k,f}}^{(k,f)} \lambda_{r_{k,f}}^{(k,f)} + b_{r_{k,f}}^{(k,f)} \lambda_{r_{k,f}}^{(k,f)^{-1}}) \right\} \\
&\propto \lambda_{r_{1,f}}^{(1,f)^{-\frac{R_{k,f} I_f}{2}}} p(\lambda_{r_{1,f}}^{(1,f)}) \cdots \lambda_{r_{f,N}}^{(f,N)^{-\frac{R_{k,f} I_f}{2}}} p(\lambda_{r_{f,N}}^{(f,N)}) \prod_{r_{k,f}} \lambda_{r_{k,f}}^{(k,f)^{-\frac{I_f}{2}+c_{r_{k,f}}^{(k,f)}-1}} \\
&\quad \exp\left\{ -\frac{1}{2}(a_{r_{k,f}}^{(k,f)} \lambda_{r_{k,f}}^{(k,f)} + (b_{r_{k,f}}^{(k,f)} + \frac{1}{\lambda_{r_{1,f}}^{(1,f)} \cdots \lambda_{r_{f,N}}^{(f,N)}} (\sum_{i_f} \mathcal{G}_f^2(r_{1,f}, \ldots, r_{f,N}))) \lambda_{r_{k,f}}^{(k,f)^{-1}}) \right\}.
\end{aligned} \tag{32}
$$

Then, we can integrate out the random variable $\lambda^{(k,f)}$ and obtain the following distribution:

$$
\begin{aligned}
&p(\mathcal{G}_f(r_{1,f}, \ldots, r_{k-1,f}, :, r_{k+1,f}, \ldots, r_{f-1,f}, :, r_{f,f+1}, \ldots, r_{f,N}), \lambda_{r_{1,f}}^{(1,f)}, \ldots, \lambda_{r_{f,N}}^{(f,N)}) \\
&\propto \lambda_{r_{1,f}}^{(1,f)^{-\frac{R_{k,f} I_f}{2}}} p(\lambda_{r_{1,f}}^{(1,f)}) \cdots \lambda_{r_{f,N}}^{(f,N)^{-\frac{R_{k,f} I_f}{2}}} p(\lambda_{r_{f,N}}^{(f,N)}) \\
&\quad \prod_{r_{k,f}} \frac{K_{c_{r_{k,f}}^{(k,f)}-\frac{I_f}{2}}\left( \sqrt{a_{r_{k,f}}^{(k,f)}} \sqrt{b_{r_{k,f}}^{(k,f)} + \frac{1}{\lambda_{r_{1,f}}^{(1,f)} \cdots \lambda_{r_{f,N}}^{(f,N)}} (\sum_{i_f} \mathcal{G}_f^2(r_{1,f}, \ldots, r_{f,N}))} \right)}{(b_{r_{k,f}}^{(k,f)} + \frac{1}{\lambda_{r_{1,f}}^{(1,f)} \cdots \lambda_{r_{f,N}}^{(f,N)}} (\sum_{i_f} \mathcal{G}_f^2(r_{1,f}, \ldots, r_{f,N})))^{\frac{I_f}{4} - \frac{c_{r_{k,f}}^{(k,f)}}{2}}}.
\end{aligned} \tag{33}
$$

Since $a^{(k,f)} \to 0$, $b^{(k,f)} \to 0$, and $c^{(k,f)} \to 0$, similarly to $\mathcal{G}_k$, the distribution over $\mathcal{G}_f$ is shown as follows:

$$p(\mathcal{G}_f(r_{1,f}, \ldots, r_{k-1,f}, :, r_{k+1,f}, \ldots, r_{f-1,f}, :, r_{f,f+1}, \ldots, r_{f,N})) \propto \prod_{r_{k,f}} \left( \sum_{i_f} \mathcal{G}_f^2(r_{1,f}, \ldots, r_{f,N}) \right)^{-\frac{I_f}{2}}. \tag{34}$$

This completes the proof. $\qquad \square$

## C. Proof of Theorem 2.2

In this section, we provide the detailed proof of the following theorem:

**Theorem C.1** (Implicit Structure Learning in MTNL). *Given a TN representation of this paper, we can express it in a normalized rank-one representation format based on Eq. 17. Based on the normalized rank-one form, the MTNL model can be reformulated as $\mathcal{F}(\cdot, \Theta^1)$, where $\Theta^1 = \{\mathcal{Z}^{(l,e)}, \{\mathcal{U}_i^{(l,e)}\}_{i=1}^T, \{\mathbf{S}_{k_1,k_2}^{(l,e)}\}_{1 \le k_1 < k_2 \le T}\}_{e=1,l=1}^{E,L}$, and $\mathcal{U}_i^{(l,e)}$ denotes the mode-fiber-wise normalized core tensor. The lower bound optimization problem of the MAP optimization problem is shown as follows:*

$$\min_{\Theta^1, \{\{\lambda_{k_1,k_2}^{(l,e)}\}_{1 \le k_1 < k_2 \le T}\}_{l=1,e=1}^{L,E}} -\log(p(\mathcal{Y} \mid \mathcal{X}, \Theta^1)) + \frac{\mu}{2} \prod_{l,e} \|\text{vec}(\mathcal{Z}^{(l,e)})\|_{2/T} - \text{term}_3, \tag{35}$$

*where* $\text{term}_3$ *denotes the last term of the MAP optimization problem, and $\|\cdot\|_{2/T}$ is the $\ell_{2/T}$ norm of a vector. By optimizing the MAP optimization problem, it implicitly optimizes Eq. 35 with $\mathcal{G}_i^{(l,e)}(r_{1,i}^{(l,e)}, \ldots, r_{i-1,i}^{(l,e)}, :, r_{i,i+1}^{(l,e)}, \ldots, r_{i,T}^{(l,e)}) = C_i^{(l,e)}(\mathcal{Z}^{(l,e)}(r_{1,2}^{(l,e)}, \ldots, r_{T-1,T}^{(l,e)}))^{1/T} \mathcal{U}_i^{(l,e)}(r_{1,i}^{(l,e)}, \ldots, r_{i-1,i}^{(l,e)}, :, r_{i,i+1}^{(l,e)}, \ldots, r_{i,T}^{(l,e)})$, where $C_i^{(l,e)}$ is a constant related to the dimension of the TN.*

*Proof.* First, we present the following equivalent optimization problem to the MAP optimization problem in Eq.7 of the main text:

$$\min_{\Theta, \Theta', \{\{\lambda_{k_1,k_2}^{(l,e)}\}_{1 \le k_1 < k_2 \le T}\}_{l=1,e=1}^{L,E}} -\log(p(\mathcal{Y} \mid \mathcal{X}, \Theta')) + \sum_{l,e,t} \frac{\tau_t^{(l,e)}}{2} \left\|\mathcal{G}_t^{(l,e)}\right\|_F^2 - \text{term}_3$$

$$s.t. \mathcal{T}^{(l,e)} = \text{SVD}_{ins}\text{TN}(\{\mathcal{G}_i^{(l,e)}\}_{i=1}^T, \{\mathbf{S}_{k_1,k_2}^{(l,e)}\}_{1 \le k_1 < k_2 \le T}) \tag{36}$$

$$l = 1, \ldots, L, e = 1, \ldots, E,$$

where, $\Theta' = \{\mathcal{T}^{(l,e)}\}_{l=1,e=1}^{L,E}$.

In the next step, we demonstrate that, given any $l = 1, \ldots, L, e = 1, \ldots, E$, the Frobenius norm regularization term of the tensor cores has an equivalent transformation form.

$$\sum_t \frac{\tau_t^{(l,e)}}{2} \|\mathcal{G}_t^{(l,e)}\|_F^2 =$$

$$\sum_{r_{1,2}=1}^{R_{1,2}} \sum_{r_{1,3}=1}^{R_{1,3}} \cdots \sum_{r_{1,N}=1}^{R_{1,N}} \sum_{r_{2,3}=1}^{R_{2,3}} \cdots \sum_{r_{2,N}=1}^{R_{2,N}} \cdots \sum_{r_{N-1,N}=1}^{R_{N-1,N}} C_1^{(l,e)} \|\mathcal{G}_1^{(l,e)}(:, r_{1,2}, r_{1,3}, \cdots, r_{1,T})\|_F^2 \tag{37}$$

$$+ \cdots + C_T^{(l,e)} \|\mathcal{G}_T^{(l,e)}(r_{1,T}, r_{2,T}, \cdots, r_{T-1,T}, :)\|_F^2.$$

To demonstrate how Eq. 37 holds, we aim to break the right-hand side of Eq. 37 into a summation over separate cores and show that each summation term is equal to $\frac{\tau_t^{(l,e)}}{2}\|\mathcal{G}_t^{(l,e)}\|_F^2$. For the first core, we have:

$$\sum_{r_{1,2}=1}^{R_{1,2}} \sum_{r_{1,3}=1}^{R_{1,3}} \cdots \sum_{r_{1,N}=1}^{R_{1,N}} \sum_{r_{2,3}=1}^{R_{2,3}} \cdots \sum_{r_{2,N}=1}^{R_{2,N}} \cdots \sum_{r_{N-1,N}=1}^{R_{N-1,N}} C_1^{(l,e)} \|\mathcal{G}_1^{(l,e)}(:, r_{1,2}, r_{1,3}, \cdots, r_{1,T})\|_F^2$$

$$= \sum_{r_{1,2}=1}^{R_{1,2}} \sum_{r_{1,3}=1}^{R_{1,3}} \cdots \sum_{r_{1,N}=1}^{R_{1,N}} C_1^{(l,e)} \|\mathcal{G}_1^{(l,e)}(:, r_{1,2}, r_{1,3}, \cdots, r_{1,T})\|_F^2 \sum_{r_{2,3}=1}^{R_{2,3}} \cdots \sum_{r_{2,N}=1}^{R_{2,N}} \cdots \sum_{r_{N-1,N}=1}^{R_{N-1,N}} 1 \tag{38}$$

$$= R_{2,3} \times \cdots \times R_{2,N} \times \cdots \times R_{N-1,N} \times C_1^{(l,e)} \|\mathcal{G}_1^{(l,e)}\|_F^2.$$

Then, if we want the last term of Eq. 38 to equal $\frac{\tau_1^{(l,e)}}{2}\|\mathcal{G}_1^{(l,e)}\|_F^2$, we only need to set $C_1^{(l,e)} = \frac{\tau_1^{(l,e)}}{2 \times R_{2,3} \times \cdots \times R_{2,N} \times \cdots \times R_{N-1,N}}$, and Eq. 37 holds with these settings of $C_t^{(l,e)}$.

In the following, we begin to show that the optimization problem in Eq. 36 has an equivalent optimization formulation based

on the rank-one format of the SVD-inspired TN representation given in Eq. 17, as shown below:

$$
\min_{\Theta,\Theta',\{\{\lambda_{k_1,k_2}^{(l,e)}\}_{1\le k_1 < k_2 \le T}\}_{l=1,e=1}^{L,E},\{g_{i_{r_{1,i},\ldots,r_{i-1,i},r_{i,i+1},\ldots,r_{i,T}}}^{(l,e)}\}} -\log(p(\mathcal{Y} \mid \mathcal{X}, \Theta')) - \text{term}_3
$$

$$
+ \sum_{l,e} \sum_{r_{1,2}=1} \sum_{r_{1,3}=1} \cdots \sum_{r_{1,T}=1} \sum_{r_{2,3}=1} \cdots \sum_{r_{2,T}=1} \cdots \sum_{r_{T-1,T}=1} C_1^{(l,e)} g_{1_{r_{1,2},r_{1,3},\cdots,r_{1,T}}}^{(l,e)^2}
$$

$$
+ \cdots + C_T^{(l,e)} g_{T_{r_{1,T},r_{2,T},\cdots,r_{T-1,T}}}^{(l,e)^2} \tag{39}
$$

$$
s.t. \mathcal{T}^{(l,e)} = \text{SVD}_{one}\text{TN}(\mathcal{Z}^{(l,e)}, \{\mathcal{U}_i^{(l,e)}\}_{i=1}^T, \{\mathbf{S}_{k_1,k_2}^{(l,e)}\}_{1\le k_1 < k_2 \le T})
$$

$$
\mathcal{Z}^{(l,e)}(r_{1,2}, r_{1,3}, \ldots, r_{1,T}, r_{2,3}, \ldots, r_{2,T}, \ldots, r_{T-1,T}) = \prod_{i=1}^T g_{i_{r_{1,i},\ldots,r_{i-1,i},r_{i,i+1},\ldots,r_{i,T}}}^{(l,e)}
$$

$$
l = 1, \ldots, L, e = 1, \ldots, E.
$$

With a slight abuse of notation, we define $\Theta = \left\{ \{\mathcal{U}_i^{(l,e)}\}_{i=1}^T, \{\mathbf{S}_{k_1,k_2}^{(l,e)}\}_{1\le k_1 < k_2 \le T}, \mathcal{Z}^{(l,e)} \right\}_{l=1,e=1}^{L,E}$. Here, $\mathcal{U}_i^{(l,e)}$ is a mode-fiber-wise normalized core tensor, and $g_{i_{r_{1,i},\ldots,r_{i-1,i},r_{i,i+1},\ldots,r_{i,T}}}^{(l,e)}$ is a positive real number. Relying on the relation $g_{i_{r_{1,i},\ldots,r_{i-1,i},r_{i,i+1},\ldots,r_{i,T}}}^{(l,e)} = \|\mathcal{G}_i^{(l,e)}(r_{1,i}, r_{2,i}, \cdots, r_{i-1,i}, :, r_{i,i+1}, \cdots, r_{i,N})\|_2$, Eq. 39 can thus be considered an equivalent optimization problem to Eq. 36.

Based on the optimization problem of Eq. 39, when we consider optimizing the set of variables $g_{i_{r_{1,i},\ldots,r_{i-1,i},r_{i,i+1},\ldots,r_{i,T}}}^{(l,e)}$ while keeping all other variables fixed, we obtain the following optimization problem:

$$
\min_{\{g_{i_{r_{1,i},\ldots,r_{i-1,i},r_{i,i+1},\ldots,r_{i,T}}}^{(l,e)}\}} \sum_{l,e} \sum_{r_{1,2}=1} \sum_{r_{1,3}=1} \cdots \sum_{r_{1,T}=1} \sum_{r_{2,3}=1} \cdots \sum_{r_{2,T}=1} \cdots \sum_{r_{T-1,T}=1}
$$

$$
C_1^{(l,e)} g_{1_{r_{1,2},r_{1,3},\cdots,r_{1,T}}}^{(l,e)^2} + \cdots + C_T^{(l,e)} g_{T_{r_{1,T},r_{2,T},\cdots,r_{T-1,T}}}^{(l,e)^2} \tag{40}
$$

$$
s.t. \mathcal{Z}^{(l,e)}(r_{1,2}, r_{1,3}, \ldots, r_{1,T}, r_{2,3}, \ldots, r_{2,T}, \ldots, r_{T-1,T}) = \prod_{i=1}^T g_{i_{r_{1,i},\ldots,r_{i-1,i},r_{i,i+1},\ldots,r_{i,T}}}^{(l,e)}
$$

$$
l = 1, \ldots, L, e = 1, \ldots, E.
$$

According to the inequality of arithmetic and geometric means, we have the following relation for the minimum value of Eq. 40:

$$
\sqrt[T]{(\sqrt{C_1^{(l,e)}} g_{1_{r_{1,2},r_{1,3},\cdots,r_{1,T}}}^{(l,e)})^2 \cdots (\sqrt{C_T^{(l,e)}} g_{T_{r_{1,T},r_{2,T},\cdots,r_{T-1,T}}}^{(l,e)})^2} \le
$$
$$
\frac{(\sqrt{C_1^{(l,e)}} g_{1_{r_{1,2},r_{1,3},\cdots,r_{1,T}}}^{(l,e)})^2 + \cdots + (\sqrt{C_T^{(l,e)}} g_{T_{r_{1,T},r_{2,T},\cdots,r_{T-1,T}}}^{(l,e)})^2}{T}. \tag{41}
$$

The equality holds when $(\sqrt{C_1^{(l,e)}} g_{1_{r_{1,2},r_{1,3},\ldots,r_{1,T}}}^{(l,e)})^2 = \cdots = (\sqrt{C_T^{(l,e)}} g_{T_{r_{1,T},r_{2,T},\ldots,r_{T-1,T}}}^{(l,e)})^2$. Combining this condition with the constraint in Eq. 40, and after some algebraic transformations, we find that the minimum of Eq. 40 is achieved when $(\sqrt{C_1^{(l,e)}} g_{1_{r_{1,2},r_{1,3},\ldots,r_{1,T}}}^{(l,e)})^2 = \cdots = (\sqrt{C_T^{(l,e)}} g_{T_{r_{1,T},r_{2,T},\ldots,r_{T-1,T}}}^{(l,e)})^2 = \left(\sqrt{C_1^{(l,e)} \cdots C_T^{(l,e)}} \mathcal{Z}^{(l,e)}(r_{1,2}, r_{1,3}, \ldots, r_{1,T}, r_{2,3}, \ldots, r_{2,T}, \ldots, r_{T-1,T})\right)^{\frac{2}{T}}$. Then, by substituting this into the objec-

tive function of Eq. 40, we find that the minimum value of Eq. 40 is:

$$T \sum_{l,e} \sum_{r_{1,2}=1} \sum_{r_{1,3}=1} \cdots \sum_{r_{1,T}=1} \sum_{r_{2,3}=1} \cdots \sum_{r_{2,T}=1} \cdots \sum_{r_{T-1,T}=1}$$

$$\left( \sqrt{C_1^{(l,e)} \cdots C_T^{(l,e)}} \, \mathcal{Z}^{(l,e)}(r_{1,2}, r_{1,3}, \ldots, r_{1,T}, r_{2,3}, \ldots, r_{2,T}, \ldots, r_{T-1,T}) \right)^{\frac{2}{T}} \tag{42}$$

$$= T \sum_{l,e} (\sqrt{C_1^{(l,e)} \cdots C_T^{(l,e)}})^{2/T} \|\mathrm{vec}(\mathcal{Z}^{(l,e)})\|_{2/T}^{2/T},$$

where $\|\cdot\|_{2/T}$ denotes the $\ell_{2/T}$-norm of a vector. Therefore, by substituting the right-hand side of Eq. 42 into Eq. 39 and eliminating irrelevant variables, we obtain the equivalent problem of Eq. 39 shown as follows:

$$\min_{\Theta,\Theta',\{\{\lambda_{k_1,k_2}^{(l,e)}\}_{1\leq k_1<k_2\leq T}\}_{l=1,e=1}^{L,E}} \quad -\log(p(\mathcal{Y} \mid \mathcal{X},\Theta')) - \mathrm{term}_3$$

$$+ T \sum_{l,e} (\sqrt{C_1^{(l,e)} \cdots C_T^{(l,e)}})^{2/T} \|\mathrm{vec}(\mathcal{Z}^{(l,e)})\|_{2/T}^{2/T} \tag{43}$$

$$s.t. \mathcal{T}^{(l,e)} = \mathrm{SVD}_{one}\mathrm{TN}(\mathcal{Z}^{(l,e)}, \{\mathcal{U}_i^{(l,e)}\}_{i=1}^T, \{\mathbf{S}_{k_1,k_2}^{(l,e)}\}_{1\leq k_1<k_2\leq T})$$

$$l = 1, \ldots, L, e = 1, \ldots, E.$$

Moreover, due to the equivalence among Eq. 36, Eq. 39, and Eq. 43, we obtain the following relation between their optimal solutions:

$$\mathcal{G}_i^{(l,e)}(r_{1,i}, \ldots, r_{i-1,i}, :, r_{i,i+1}, \ldots, r_{i,T}) = K_i^{(l,e)} \left( \mathcal{Z}^{(l,e)}(r_{1,2}, \ldots, r_{T-1,T}) \right)^{1/T} \mathcal{U}_i^{(l,e)}(r_{1,i}, \ldots, r_{i-1,i}, :$$

$$, r_{i,i+1}, \ldots, r_{i,T}), \text{ where } K_i^{(l,e)} = (\sqrt{C_1^{(l,e)} \cdots C_T^{(l,e)}})^{\frac{1}{T}}/\sqrt{C_i^{(l,e)}}.$$

In the following, we begin to show that Eq. 35 in the theorem serves as a lower-bound optimization problem to Eq. 43. First, with a specific value of $\omega$, the following optimization problem is equivalent to Eq. 43:

$$\min_{\Theta,\Theta',\{\{\lambda_{k_1,k_2}^{(l,e)}\}_{1\leq k_1<k_2\leq T}\}_{l=1,e=1}^{L,E}} \quad T \sum_{l,e} (\sqrt{C_1^{(l,e)} \cdots C_T^{(l,e)}})^{2/T} \|\mathrm{vec}(\mathcal{Z}^{(l,e)})\|_{2/T}^{2/T}$$

$$s.t. - \log(p(\mathcal{Y} \mid \mathcal{X},\Theta')) - \mathrm{term}_3 \leq \omega^2 \tag{44}$$

$$\mathcal{T}^{(l,e)} = \mathrm{SVD}_{one}\mathrm{TN}(\mathcal{Z}^{(l,e)}, \{\mathcal{U}_i^{(l,e)}\}_{i=1}^T, \{\mathbf{S}_{k_1,k_2}^{(l,e)}\}_{1\leq k_1<k_2\leq T})$$

$$l = 1, \ldots, L, e = 1, \ldots, E.$$

By defining $H := \min(\{(\sqrt{C_1^{(l,e)} \cdots C_T^{(l,e)}})^{2/T}\}_{l=1,e=1}^{L,E})$, we have:

$$T \sum_{l,e} (\sqrt{C_1^{(l,e)} \cdots C_T^{(l,e)}})^{2/T} \|\mathrm{vec}(\mathcal{Z}^{(l,e)})\|_{2/T}^{2/T} \tag{45}$$

$$\geq TH \sum_{l,e} \|\mathrm{vec}(\mathcal{Z}^{(l,e)})\|_{2/T}^{2/T} \tag{46}$$

$$= TH \sum_{l,e} (\|\mathrm{vec}(\mathcal{Z}^{(l,e)})\|_{2/T}^{1/T})^2 \tag{47}$$

$$\geq THLE(\prod_{l,e} \|\mathrm{vec}(\mathcal{Z}^{(l,e)})\|_{2/T}^{1/T})^{2/LE}. \tag{48}$$

Therefore, the lower-bound optimization problem to Eq. 44 is:

$$\min_{\Theta,\Theta',\{\{\lambda_{k_1,k_2}^{(l,e)}\}_{1\leq k_1<k_2\leq T}\}_{l=1,e=1}^{L,E}} \quad THLE(\prod_{l,e} \|\mathrm{vec}(\mathcal{Z}^{(l,e)})\|_{2/T}^{1/T})^{2/LE}$$

$$s.t. - \log(p(\mathcal{Y} \mid \mathcal{X},\Theta')) - \mathrm{term}_3 \leq \omega^2 \tag{49}$$

$$\mathcal{T}^{(l,e)} = \mathrm{SVD}_{one}\mathrm{TN}(\mathcal{Z}^{(l,e)}, \{\mathcal{U}_i^{(l,e)}\}_{i=1}^T, \{\mathbf{S}_{k_1,k_2}^{(l,e)}\}_{1\leq k_1<k_2\leq T})$$

$$l = 1, \ldots, L, e = 1, \ldots, E.$$

By dropping the constant coefficients, and also because the power functions of $\frac{1}{T}$ and $\frac{2}{LE}$ are strictly increasing functions, we can thus drop the coefficients and also the powers, to get the equivalent optimization problem:

$$\min_{\Theta,\Theta',\{\{\lambda_{k_1,k_2}^{(l,e)}\}_{1\leq k_1<k_2\leq T}\}_{l=1,e=1}^{L,E}} \prod_{l,e} \|\text{vec}(\mathcal{Z}^{(l,e)})\|_{2/T}$$

$$s.t. -\log(p(\mathcal{Y}\mid\mathcal{X},\Theta')) - \text{term}_3 \leq \omega^2 \tag{50}$$

$$\mathcal{T}^{(l,e)} = \text{SVD}_{one}\text{TN}(\mathcal{Z}^{(l,e)}, \{\mathcal{U}_i^{(l,e)}\}_{i=1}^T, \{\mathbf{S}_{k_1,k_2}^{(l,e)}\}_{1\leq k_1<k_2\leq T})$$

$$l = 1,\ldots,L, e = 1,\ldots,E.$$

This completes the proof by recasting the constraint into an optimization objective, and equivalently transforming it into the optimization problem in Eq. 35. □

## D. Proof of Theorem 3.1

In this section, we provide the detailed proof of the following theorem:

**Theorem D.1** (Implicit Dropout Learning). *Based on the dropout TN in Eq. 18, we refer to a dropout MTNL model whenever using the model parameters* $\widetilde{\Theta} = \{\{\mathbf{V}_{k_1,k_2}^{(l,e)}\}_{1\leq k_1<k_2\leq T}, \{\mathcal{G}_i^{(l,e)}\}_{i=1}^T, \{\mathbf{S}_{k_1,k_2}^{(l,e)}\}_{1\leq k_1<k_2\leq T}\}_{e=1,l=1}^{E,L}$. *If we drop the tilde and omit the random matrices, using* $\Theta$ *instead, we refer to the standard MTNL model. Moreover, in this theorem, we consider only the linearized MTNL model with identical activation. In the following equation, we demonstrate that the dropout variant of the MAP problem has an upper bound.*

$$-\mathbb{E}_{[\mathbf{V}]}\left[\log(p(\mathcal{Y}\mid\mathcal{X},\tilde{\Theta}))\right] - \sum_{l,e,k_1,k_2}\left(\log(p(s_{k_1,k_2}^{(l,e)}\mid\lambda_{k_1,k_2}^{(l,e)})) + \log(p(\lambda_{k_1,k_2}^{(l,e)}))\right) \tag{51}$$

$$\leq -\log(p(\mathcal{Y}\mid\mathcal{X},\Theta)) + \sum_{l,e,i}k_i^{(l,e)}(\theta)\|\mathcal{G}_i^{(l,e)}\|_F^2 + \sum_{l,e,k_1,k_2}b_{k_1,k_2}^{(l,e)}(\theta)\frac{1}{2\min(\lambda_{k_1,k_2}^{(l,e)})}\|\mathbf{S}_{k_1,k_2}^{(l,e)}\|_F^2$$

$$- \sum_{l,e,k_1,k_2}\log(p(\lambda_{k_1,k_2}^{(l,e)})). \tag{52}$$

*Here,* $k_i^{(l,e)}(\theta)$ *and* $b_{k_1,k_2}^{(l,e)}(\theta) \in [0,+\infty]$ *are functions controlled by the parameter* $0 \leq \theta \leq 1$, *and they are not related to the optimization variables of the problem.*

*Proof.* First, we start by giving an explicit expression for the problem loss of dropout-linearized MTNL, $-\mathbb{E}_{[\mathbf{V}]}\left[\log\left(p(\mathcal{Y}\mid\mathcal{X},\tilde{\Theta})\right)\right]$, and demonstrate its upper bound. The expression is shown as follows:

$$\mathbb{E}_{[\mathbf{V}]}\left[\frac{\epsilon}{2}\sum_{i=1}^N\left\|\mathcal{Y}^i - \frac{1}{\theta}\mathcal{F}_{lin}(\mathcal{X}^i,\widetilde{\Theta})\right\|_F^2\right] \tag{53}$$

Due to the omission of nonlinearity, the computation process of $\mathcal{F}_{\text{lin}}(\mathcal{X}^i,\widetilde{\Theta})$ proceeds by first contracting the input $\mathcal{X}^i$ in parallel with different TNs. Using a matrix decomposition as a demonstration, this can be written as $\mathbf{O}_1 = \mathbf{B}^{(1,1)}\mathbf{A}^{(1,1)}\mathbf{X} + \cdots + \mathbf{B}^{(1,E)}\mathbf{A}^{(1,E)}\mathbf{X}$, and this process is repeated in the second layer on $\mathbf{O}_1$, resulting in $\mathbf{O}_2 = \mathbf{B}^{(2,1)}\mathbf{A}^{(2,1)}\mathbf{O}_1 + \cdots + \mathbf{B}^{(2,E)}\mathbf{A}^{(2,E)}\mathbf{O}_1$. It is easy to see that, by some algebraic manipulations, this process is equivalent to summing over $E^L$ different deep matrix factorizations of the input $\mathbf{X}^i$, where at each layer, one matrix factorization is chosen for forward propagation, i.e., $\cdots\mathbf{B}^{(2,1)}\mathbf{A}^{(2,1)}\mathbf{B}^{(1,1)}\mathbf{A}^{(1,1)}\mathbf{X}^i\overbrace{+\cdots+}^{\times E^L}\cdots\mathbf{B}^{(2,E)}\mathbf{A}^{(2,E)}\mathbf{B}^{(1,E)}\mathbf{A}^{(1,E)}\mathbf{X}^i$. Although the TN case involves more complex contraction indices, it is essentially similar to its lower-order matrix counterpart, and thus Eq. 53 is equivalent to this formulation.

$$\frac{\epsilon}{2}\sum_{i=1}^N\mathbb{E}_{[\mathbf{V}]}\left[\left\|\mathcal{Y}^i - \frac{1}{\theta}\left(\text{TN}(\{\mathcal{G}_1\},\{\mathbf{V}_1\},\{\mathbf{S}_1\})\overbrace{+\cdots+}^{E^L}\text{TN}(\{\mathcal{G}_{E^L}\},\{\mathbf{V}_{E^L}\},\{\mathbf{S}_{E^L}\})\right) \times_{in\_index}\mathcal{X}^i\right\|_F^2\right]. \tag{54}$$

In this equation, $\text{TN}(\cdot)$ represents the same deep TN with internal cores, constructed from the tensor cores $\{\mathcal{G}_i\}$, threshold matrices $\{\mathbf{S}_i\}$, and dropout matrices $\{\mathbf{V}_i\}$, from TNs of different layers, where $i = 1, \ldots, E^L$. Moreover, this equation can also be expressed elementwise:

$$\frac{\epsilon}{2} \sum_{i=1}^{N} \sum_{out\_index}$$
$$\mathbb{E}_{[\mathbf{V}]} \left[ \left( \mathcal{Y}^i_{out\_index} - \frac{1}{\theta} \left( \left( \text{TN}(\{\mathcal{G}_1\}, \{\mathbf{V}_1\}, \{\mathbf{S}_1\}) \overbrace{+\cdots+}^{E^L} \text{TN}(\{\mathcal{G}_{E^L}\}, \{\mathbf{V}_{E^L}\}, \{\mathbf{S}_{E^L}\}) \right) \times_{in\_index} \mathcal{X}^i \right)_{out\_index} \right)^2 \right]. \tag{55}$$

Then, using the bias-variance decomposition $\mathbb{E}[r^2] = \mathbb{V}[r] + \mathbb{E}[r]^2$, which holds for a scalar random variable $r$, Eq. 55 can be decomposed into the sum of two terms. The first term is:

$$\frac{\epsilon}{2} \sum_{i=1}^{N} \sum_{out\_index}$$
$$\mathbb{E}_{[\mathbf{V}]} \left[ \mathcal{Y}^i_{out\_index} - \frac{1}{\theta} \left( \left( \text{TN}(\{\mathcal{G}_1\}, \{\mathbf{V}_1\}, \{\mathbf{S}_1\}) \overbrace{+\cdots+}^{E^L} \text{TN}(\{\mathcal{G}_{E^L}\}, \{\mathbf{V}_{E^L}\}, \{\mathbf{S}_{E^L}\}) \right) \times_{in\_index} \mathcal{X}^i \right)_{out\_index} \right]^2. \tag{56}$$

Since all of the operations in Eq. 56 are linear operations, and the dropout variables are i.i.d. distributed, it is easy to see that the expectation operator can be brought inside each TN and to the randomized dropout matrix $\mathbf{V}$, as expressed by the following:

$$\frac{\epsilon}{2} \sum_{i=1}^{N} \sum_{out\_index}$$
$$\left( \mathcal{Y}^i_{out\_index} - \frac{1}{\theta} \left( \left( \text{TN}(\{\mathcal{G}_1\}, \mathbb{E}(\{\mathbf{V}_1\}), \{\mathbf{S}_1\}) \overbrace{+\cdots+}^{E^L} \text{TN}(\{\mathcal{G}_{E^L}\}, \mathbb{E}(\{\mathbf{V}_{E^L}\}), \{\mathbf{S}_{E^L}\}) \right) \times_{in\_index} \mathcal{X}^i \right)_{out\_index} \right)^2. \tag{57}$$

Exploiting the analytical formula for the expected value of a Bernoulli($\theta$) distribution: $\mathbb{E}[r] = \theta$. Eq. 57 can be transformed into the following:

$$\frac{\epsilon}{2} \sum_{i=1}^{N} \sum_{out\_index} \left( \mathcal{Y}^i_{out\_index} - \left( \left( \text{TN}(\{\mathcal{G}_1\}, \{\mathbf{S}_1\}) \overbrace{+\cdots+}^{E^L} \text{TN}(\{\mathcal{G}_{E^L}\}, \{\mathbf{S}_{E^L}\}) \right) \times_{in\_index} \mathcal{X}^i \right)_{out\_index} \right)^2. \tag{58}$$

It is clear that this equation is the elementwise form of the problem loss of the deterministic linearized MTNL.

$$\frac{\epsilon}{2} \sum_{i=1}^{N} \left\| \mathcal{Y}^i - \left( \text{TN}(\{\mathcal{G}_1\}, \{\mathbf{S}_1\}) \overbrace{+\cdots+}^{E^L} \text{TN}(\{\mathcal{G}_{E^L}\}, \{\mathbf{S}_{E^L}\}) \right) \times_{in\_index} \mathcal{X}^i \right\|^2_F. \tag{59}$$

Eq. 59 can be further transformed into $\frac{\epsilon}{2} \sum_{i=1}^{N} \left\| \mathcal{Y}^i - \mathcal{F}_{lin}(\mathcal{X}^i, \Theta) \right\|^2_F$, which is equal to $-\log(p(\mathcal{Y} \mid \mathcal{X}, \Theta))$ in Eq. 52.

In the above, we have shown how the first decomposed term of Eq. 55 is related to Eq. 52. From now on, we begin to show how the second term is related. Furthermore, the detailed expression of the second term is given as follows:

$$\frac{\epsilon}{2} \sum_{i=1}^{N} \sum_{out\_index}$$
$$\mathbb{V}_{[\mathbf{V}]} \left[ \mathcal{Y}^i_{out\_index} - \frac{1}{\theta} \left( \left( \text{TN}(\{\mathcal{G}_1\}, \{\mathbf{V}_1\}, \{\mathbf{S}_1\}) \overbrace{+\cdots+}^{E^L} \text{TN}(\{\mathcal{G}_{E^L}\}, \{\mathbf{V}_{E^L}\}, \{\mathbf{S}_{E^L}\}) \right) \times_{in\_index} \mathcal{X}^i \right)_{out\_index} \right]. \tag{60}$$

Utilizing the property of variance computation, Eq. 60 is equal to the following:

$$\frac{\epsilon}{2}\sum_{i=1}^{N}\sum_{out\_index}\frac{1}{\theta^2}\mathbb{V}_{[\mathbf{V}]}\left[\left(\left(\mathrm{TN}\left(\{\mathcal{G}_1\},\{\mathbf{V}_1\},\{\mathbf{S}_1\}\right)\overbrace{+\cdots+}^{E^L}\mathrm{TN}\left(\{\mathcal{G}_{E^L}\},\{\mathbf{V}_{E^L}\},\{\mathbf{S}_{E^L}\}\right)\right)\times_{in\_index}\mathcal{X}^i\right)_{out\_index}\right].$$
(61)

Utilizing the variance property $\mathbb{V}(a_1+\cdots+a_n)\le n(\mathbb{V}(a_1)+\cdots+\mathbb{V}(a_n))$. Also since all of the operations in $\mathrm{TN}(\cdot)$ are linear, and the dropout variables are i.i.d. distributed, the variance operator can be brought inside each TN, down to the randomized dropout matrix $\mathbf{V}$ ($\mathbb{V}[r]=\theta(1-\theta)$), with all other TN components raised elementwise to the power of two. Combining the above factors, Eq. 61 can be upper bounded into the following:

$$\frac{(1-\theta)\epsilon}{2\theta}\sum_{i=1}^{N}\sum_{out\_index}\left(\left(\mathrm{TN}\left(\{\mathcal{G}_1^2\},\{\mathbf{S}_1^2\}\right)\overbrace{+\cdots+}^{E^L}\mathrm{TN}\left(\{\mathcal{G}_{E^L}^2\},\{\mathbf{S}_{E^L}^2\}\right)\right)\times_{in\_index}(\mathcal{X}^i)^2\right)_{out\_index}.$$
(62)

From now on, we start to prove the upper bound of the summation term in Eq. 62 over the $out\_index$.

$$\sum_{in\_index}\left(\mathrm{TN}\left(\{\mathcal{G}_1^2\},\{\mathbf{S}_1^2\}\right)\overbrace{+\cdots+}^{E^L}\mathrm{TN}\left(\{\mathcal{G}_{E^L}^2\},\{\mathbf{S}_{E^L}^2\}\right)\right)_{in\_index}(\mathcal{X}^i)^2_{in\_index}$$
(63)

$$\le\frac{1}{2}\sum_{in\_index}\left(\mathrm{TN}\left(\{\mathcal{G}_1^4\},\{\mathbf{S}_1^4\}\right)\overbrace{+\cdots+}^{E^L}\mathrm{TN}\left(\{\mathcal{G}_{E^L}^4\},\{\mathbf{S}_{E^L}^4\}\right)\right)_{in\_index}.$$
(64)

The inequality of Eq. 64 holds by utilizing the inequality of arithmetic and geometric means, $a^2b^2\le\frac{a^4+b^4}{2}$. Firstly, if we write Eq. 63 in an element-wise form, it is basically the product of several tensor elements. Then, by considering two parts of the product, we treat $(\mathcal{X}^i)^2$ as one part, and the product of the other tensors raised to the power of two as the second part. We can then use the relation $a^2b^2\le\frac{a^4+b^4}{2}$ to separate these two parts and derive Eq. 64, where each tensor component is raised to the power of 4 and multiplied by the coefficient $\frac{1}{2}$. The term $(\mathcal{X}^i)^4$ is dropped because it is not relevant to the optimization problem.

In the following, we will use the inequality $(a_1\cdots a_n)^4\le\frac{1}{n^{2n}}(a_1^2+\cdots+a_n^2)^{2n}$, which is derived from the inequality of arithmetic and geometric means. To incorporate Eq. 64 with this inequality and derive an upper bound for it, we first need to interpret it using its element-wise form. In the element-wise form of Eq. 64, for each TN, it is basically a series of tensor elements raised to the power of 4 and multiplied together. This can be seen as $(a_1\cdots a_n)^4$, where $n$ is the total number of tensor cores and threshold matrices in one deep TN, which we denote as $C$. Then Eq. 64 has the following upper bound:

$$\frac{1}{2C^{2C}}\sum_{in\_index}\sum_{TN\_index}(\mathcal{G}_{1_{index}}^2+\cdots+\mathbf{S}_{1_{index}}^2+\cdots)^{2C}\overbrace{+\cdots+}^{E^L}(\mathcal{G}_{E^L_{index}}^2+\cdots+\mathbf{S}_{E^L_{index}}^2+\cdots)^{2C}$$
(65)

$$\le\frac{1}{2C^{2C}}\sum_{t}^{E^L}(\sum_{in\_index}\sum_{TN\_index}\mathcal{G}_{t_{index}}^2+\cdots+\mathbf{S}_{t_{index}}^2+\cdots)^{2C}.$$
(66)

Eq. 66 is the final upper bound of the summation term in Eq. 62 over the $out\_index$, and the upper bound of Eq. 62 is shown as follows:

$$\frac{(1-\theta)\epsilon}{4\theta C^{2C}}\sum_{i=1}^{N}\sum_{out\_index}\sum_{t}(\sum_{in\_index}\sum_{TN\_index}\mathcal{G}_{t_{index}}^2+\cdots+\mathbf{S}_{t_{index}}^2+\cdots)^{2C}$$
(67)

$$\le\frac{(1-\theta)\epsilon}{4\theta C^{2C}}(\sum_{i=1}^{N}\sum_{out\_index}\sum_{t}\sum_{in\_index}\sum_{TN\_index}\mathcal{G}_{t_{index}}^2+\cdots+\mathbf{S}_{t_{index}}^2+\cdots)^{2C}$$
(68)

$$=\frac{(1-\theta)\epsilon}{4\theta C^{2C}}(\sum_{l,e,t}p_t^{(l,e)}\|\mathcal{G}_t^{(l,e)}\|_F^2+\sum_{l,e,k_1,k_2}q_{k_1,k_2}^{(l,e)}\|s_{k_1,k_2}^{(l,e)}\|_F^2)^{2C}.$$
(69)

In Eq. 69, $p_t^{(l,e)}$ and $q_{k_1,k_2}^{(l,e)}$ are constants related to the dimensions of the problem and are not optimization variables. Since Eq. 69 is actually a regularization term of the problem, for a similar reason as in the proof of Theorem 2.2 in Eq. 49, the power function with exponent up to $2C$ is strictly increasing. Thus, we consider dropping the power and obtain an equivalent regularization: $\sum_{l,e,t} \frac{(1-\theta)\epsilon}{4\theta C^{2C}} p_t^{(l,e)} \|\mathcal{G}_t^{(l,e)}\|_F^2 + \sum_{l,e,k_1,k_2} \frac{(1-\theta)\epsilon}{4\theta C^{2C}} q_{k_1,k_2}^{(l,e)} \|s_{k_1,k_2}^{(l,e)}\|_F^2$. Here, $\frac{(1-\theta)\epsilon}{4\theta C^{2C}} p_t^{(l,e)}$ and $\frac{(1-\theta)\epsilon}{4\theta C^{2C}} q_{k_1,k_2}^{(l,e)}$ are functions of $\theta$ that are not related to the optimization problem, and their range is $[0, +\infty]$. We denote them as $k_t^{(l,e)}(\theta)$ and $b_{k_1,k_2}^{(l,e)}(\theta)$, respectively.

Now, we start to show that the term $-\log(p(s_{k_1,k_2}^{(l,e)} \mid \lambda_{k_1,k_2}^{(l,e)}))$ in Eq. 51 has an upper bound. Specifically, $-\log\big(p(s_{k_1,k_2}^{(l,e)} \mid \lambda_{k_1,k_2}^{(l,e)})\big) = \text{Sum}\left(s_{k_1,k_2}^{(l,e)} \circledast s_{k_1,k_2}^{(l,e)} \circledast \left(\frac{1}{2\lambda_{k_1,k_2}^{(l,e)}}\right)\right) \le \frac{1}{2\min(\lambda_{k_1,k_2}^{(l,e)})} \left\|s_{k_1,k_2}^{(l,e)}\right\|_F^2$.

Finally, the upper bound of Eq. 51 is obtained by summing over Eq. 59, the equivalent transformation of Eq. 69, $\frac{1}{2\min(\lambda_{k_1,k_2}^{(l,e)})} \left\|s_{k_1,k_2}^{(l,e)}\right\|_F^2$, and $-\sum_{l,e,k_1,k_2} \log\big(p(\lambda_{k_1,k_2}^{(l,e)})\big)$. $\qquad\square$

# E. Computational Complexity

For simplicity, assuming the TN in the MTNL has the order of $2d$, and $d$ orders are for the input feature, $d$ orders for the output feature, assuming the dimension of each mode of the TN is $N$, meaning $h = N^d$, where $h$ is the input feature vector's dimension. For this TN, we assume all of the rank is $r$, then in the forward pass, the tensor that is kept in the memory is related to the step of the contraction $s$, which is when $1 \le s \le d$ the memory is $N^{(d-s)}r^{s(2d-s)}$ when $d+1 \le s \le 2d$ memory required is $N^{(s-d)}r^{s(2d-s)}$. For the computational complexity, it also in each step changes, when $1 \le s \le d$ complexity is $N^{(d-s)}r^{s(2d-s)}Nr^{(s-1)}$ when $d+1 \le s \le 2d$ complexity is $N^{(s-d)}r^{s(2d-s)}r^{(s-1)}$, therefore, the total computational complexity is $\sum_{s=1}^{d} N^{(d-s)}r^{s(2d-s)}Nr^{(s-1)} + \sum_{s=d+1}^{2d} N^{(s-d)}r^{s(2d-s)}r^{(s-1)}$. Moreover, the model parameters that need to be stored in the optimizer are $2d(Nr^{(2d-1)})$. Then, assume the MTNL in each layer has $E$ TNs, the memory requirement, computational complexity and trainable parameters need to be multiplied by $E$ based on the above statement. Therefore, as we can analyze from the results, the main complexity overhead of the MTNL is the order of the TNs, which cannot grow too large. However, as demonstrated in both the tensor recovery and PEFT experiments, a relatively low order is sufficient to outperform other methods.

# F. A Generalized TN Recovery Model

In this section, we show that the proposed MTNL model is a powerful TN modeling framework that induces a generalized TN recovery model. This view helps explain the source of MTNL's performance gains and also provides a more principled way for designing better MTNL models. First, we present the MTNL learning formulation in the following equation:

$$\mathcal{Y}_i = \mathcal{F}(\mathcal{X}_i; \Theta) = f^L(\{\mathcal{G}_i^{(L,e)}\}, \{\mathbf{S}_{k_1,k_2}^{(L,e)}\}\underbrace{,\dots,}_{\substack{\text{forward}\\ \times L}} f^1(\{\mathcal{G}_i^{(1,e)}\}, \{\mathbf{S}_{k_1,k_2}^{(1,e)}\}, \mathcal{X}_i)\overbrace{\cdots}^{\times L}). \tag{70}$$

In MTNL, the input tensor $\mathcal{X}_i$ is forward-propagated layer by layer to produce the final output, and each layer is constructed from several TN components. However, the detailed forward-propagation process within each layer is not explicitly specified in Eq. 70, thereby enabling a potential more flexible model design.

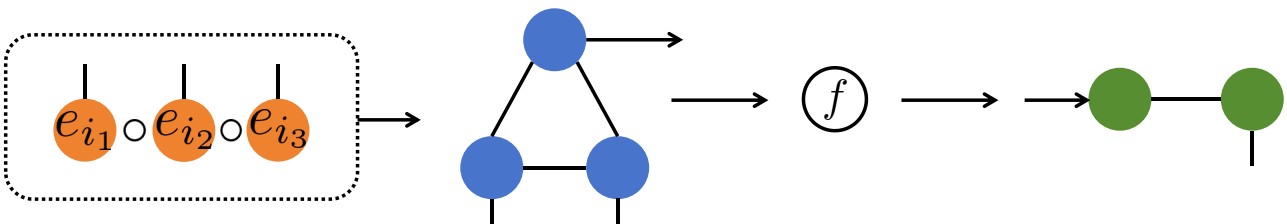

*Figure 2.* A special case of the MTNL model.

In Fig. 2, we illustrate a special case of the MTNL model. In this example, the input tensor $\mathcal{X}_i$ is constructed as the outer

product of basic vectors, which encode the location of each tensor element. In the first layer, the 3-order input tensor is contracted with the TN in the first layer along two edges (the downward-pointing edges), while the remaining edge of the input becomes one of the modes of the layer's output. After passing through the activation function, this remaining mode is then contracted with the TN in the second layer along the downward edge.

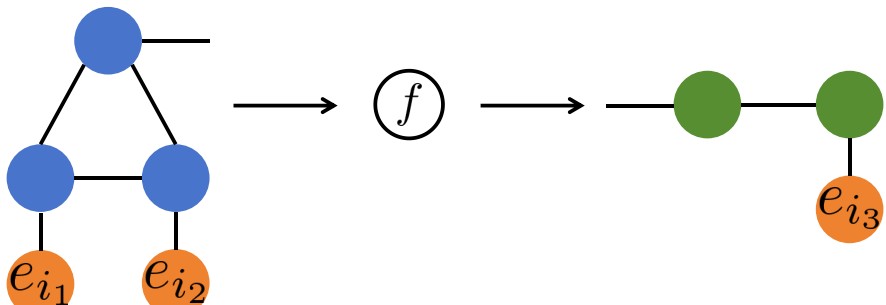

*Figure 3.* Generalized TN recovery model.

The model in Fig. 2 is an MTNL model with a specific design of the forward-propagation function. Most importantly, we claim that the model in Fig. 2 is equivalent to the model in Fig. 3. What's interesting about this interpretation is that, from the DL perspective, in many applications the inputs to the network are distributed across different layers rather than injected only at the first layer. For example, injecting auxiliary information at certain layers. From the TN recovery modeling perspective, it can be seen as a generalized TN design, where the correlation or connection between modes 1, 2 and mode 3 is strengthened by the nonlinear action and other DL-inspired components between them. More broadly speaking, this provides the insight that, for any traditional way of TN construction, we can add DL-inspired components between arbitrary connections (edges) to enhance its expressive power. From the opposite perspective, we can also design a suitable MTNL model by harnessing insights from traditional TN constructions.

In Fig. 4, we demonstrate a step-by-step guide showing how to convert the special case of the MTNL in Fig. 2 into the generalized TN recovery model in Fig. 3. In the first figure of Fig. 4, we forward-propagate the input to the first layer of the MTNL, where two of the input basis vectors are contracted with the TN. The output of the first layer passing through the nonlinear activation is illustrated in the second figure of Fig. 4. The ouput is a matrix constructed by the outer product of two vectors, one of which is the remaining basis vector. It is easy to verify that applying the element-wise nonlinear activation to this matrix is equivalent to performing the nonlinear activation directly on the non-basis vector, as shown in the third figure of Fig. 4. Finally, the MTNL output, shown in the last figure of Fig. 4, is equivalent to the forward-propagation process and the final layer result of the generalized TN recovery model in Fig. 3.

In the above demonstration, to simplify the process and more directly highlight the core proof idea, we only consider an MTNL with $E = 1$ and $L = 2$. However, the transformation process in Fig. 4, based on the basic TN algebra, can be similarly extended to MTNL with multiple layers and multiple TNs within a single layer. Furthermore, since all the models described above are special cases of the MTNL model and can be expressed within Eq. 70, and since all theoretical results are proven using the abstract form of Eq. 70, the main theoretical results of the paper, as well as the extended theoretical analysis in Section G, can be naturally generalized to these models.

## G. Extended Theoretical Analysis

In this section, we aim to demonstrate that the theoretical analysis presented in this paper can be generalized to several structural extensions of the MTNL, as shown in Fig. 5 and Fig. 6. The extended structure shown in Fig. 5 illustrates the skip connections introduced in the MTNL, which help enhance gradient flow. This structure is used in the tensor recovery and tensor regression experiments discussed in the main text. The second extended structure, shown in Fig. 6, is inspired by the architecture of LLMs. In this design, each layer of the MTNL contains many groups of TNs instead of one group, and the output tensors of these groups are further contracted along specific edges to produce the output of the layer. This process mimics the compression of the attention blocks.

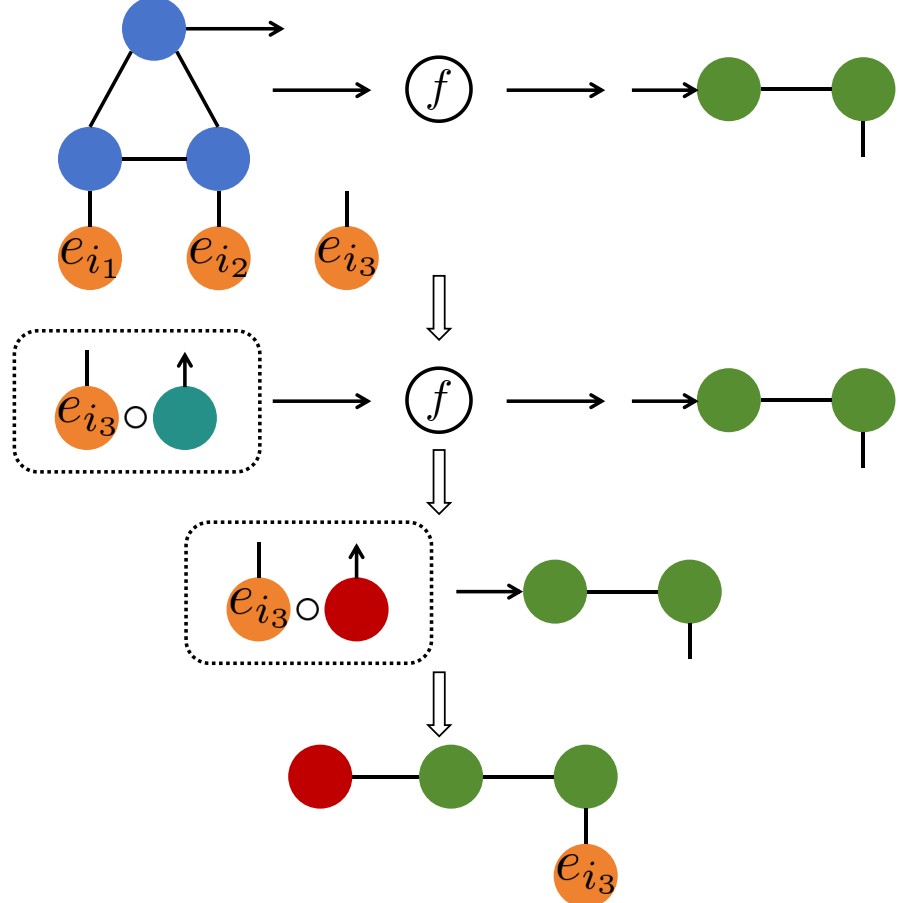

*Figure 4.* From MTNL to the generalized TN recovery model.

### G.1. Sparsity-Inducing Prior and Implicit Structure Learning

The extended theoretical analysis of the sparsity-inducing prior and the implicit structure learning effect is quite straightforward. As we can observe from the derivation process in the main text and in Appendix C, these results are developed separately from the model formulation.

For the sparsity-inducing prior, the prior modeling process is constructed independently of the main modeling procedure. Thus, we can safely impose priors on different TNs within the extended structures without affecting the marginalization process.

The same reasoning applies to the implicit structure learning analysis. The proof process remains orthogonal to the modeling process, with the only modification being the explicit construction of a different number of TNs in the constraints of the extended MTNL structure 2 compared to Eq. 36. The rest of the proof follows analogously.

### G.2. Implicit Dropout Learning

For the extended analysis of implicit dropout learning, the process is less straightforward compared to the sparsity-inducing prior and implicit structure learning, as the proof involves handling the model structure. The main difference lies in Eq. 54.

For the extended structure shown in Fig. 5, due to the introduction of the skip connection, Eq. 54 is not as straightforward as constructing a deep TN by simply selecting TNs in each layer. Instead, in the linear case, the extended structure requires that in every layer, the output tensor is multiplied by 2. As a result, the construction of the deep TN in Eq. 54 involves an accumulated scaling coefficient. Beyond this, the rest of the proof follows analogously.

For the extended structure 2 shown in Fig. 6, a more complex adjustment of the construction of the multiple deep TNs in

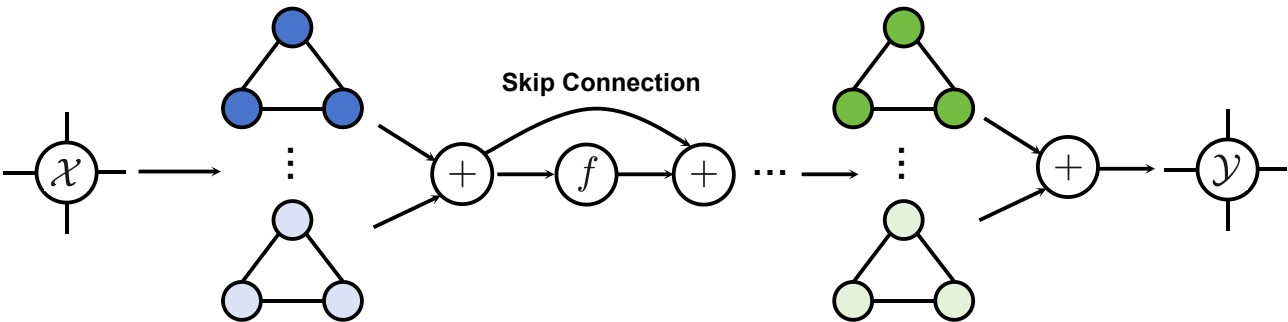

*Figure 5.* Extended MTNL structure 1.

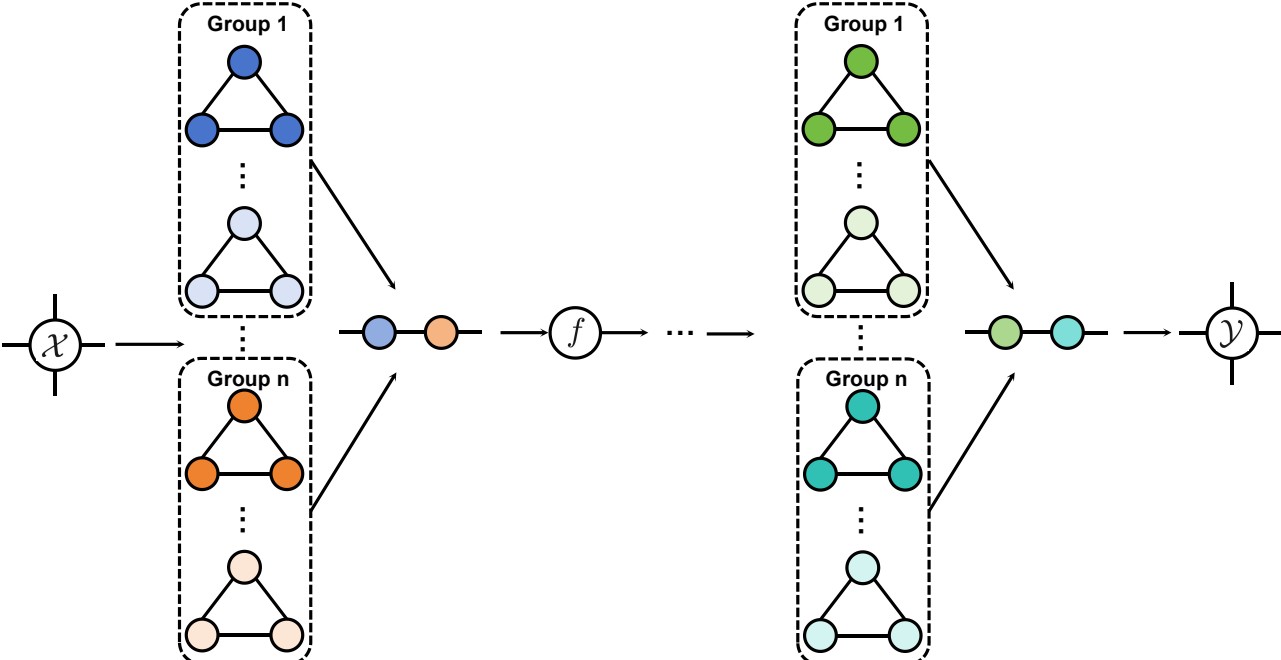

*Figure 6.* Extended MTNL structure 2.

Eq. 54 is required compared to extended structure 1. Specifically, since each layer of this extended structure involves an inner-layer contraction between different groups of TNs, it can be considered as increasing the number of parallel TNs per layer by a factor of $n$, compared to the MTNL in the main text or extended structure 1. Therefore, in Eq. 54, the summation over deep TNs becomes longer and contains $E^{nL}$ terms. Beyond this, the rest of the proof structure remains similar, and the argument follows analogously.

## H. Additional Experiment Details

In this section, we present the additional details of the experiments conducted in this paper, including how the datasets were generated or selected, the detailed parameter settings of the proposed method and the comparison methods, as well as some additional experimental results. The PEFT experiments were conducted on NVIDIA A100 40GB GPUs. The other experiments were conducted on an Apple M4 Max processor with 128GB of memory.

### H.1. Guidelines on Hyperparameter Selection

First, we enumerate the core hyperparameters in the model: depth $E$ and width $L$, activation function, TN structures in MTNL, TN-ranks, noise precision $\epsilon$, $\tau$ for the core prior, $a, b, c$ for the threshold prior, dropout rate $\theta$, and sample number

$N$.

Since nonlinearity is essential for distinguishing the proposed method from traditional methods, in practice, we should retain the ReLU activation. For $E$ and $L$, we can either tune both hyperparameters or fix $L = 2$ and tune only $E$. Fixing $E, L = 2$ was demonstrated to be effective in the tensor recovery experiment.

For the TN structure settings in each MTNL layer, determining the optimal configuration remains an open problem. For image data, one can follow the settings used in this paper, where the spectral mode is separated from the other modes.

For TN-ranks, they are adjusted jointly. In practice, we can directly tune them to align the number of model parameters with other TN models. Since MTNL is more expressive, this setting may already guarantee good performance. Alternatively, we can try several different integer values (e.g., 5, 10, etc.), which are commonly used in TN studies.

For $\epsilon$, this is data-dependent. If useful information is limited (e.g., few observed entries), $\epsilon$ should be small (e.g., 0.1, 0.01, ...). Otherwise, larger values (e.g., 1, 10, ...) can be used. For $\tau$, it controls regularization and is also related to the amount of available information. If information is limited, a large $\tau$ (e.g., 1, 5, ...) is preferred. Otherwise, smaller values (e.g., 0.1, 0.05, ...) can be used. Both $\epsilon$ and $\tau$ are common in ML models and can follow standard selection practices.

For $\{a, b, c\}$, in practice, we can first perform a coarse search over the four candidate sets: $\{(1e^{-6}, 1e^{-6}, -2), (1, 1e^{-6}, 5), (1, 1e^{-6}, 1), (1e^{-6}, 1, -5)\}$. Then, a fine-grained sparsity-level search can be conducted. Since $c$ mainly controls sparsity, if the problem contains limited information, smaller values (e.g., $-5$, $-15$, etc.) can be used; otherwise, larger values (e.g., 5, 15, etc.) are preferred. This strategy reduces the tuning complexity from combinatorial to linear.

Hyperparameters chosen from the above settings allow MTNL to be tuned along five "axes". In contrast, traditional TN methods typically involve three. The additional axes in MTNL are $\{E, L\}$ and $\{a, b, c\}$.

For $\theta$ and the sample number $N$, they can be determined using a validation set. For example, test $\theta$ from 0.1 to 0.9 and $N$ 50, 100, 150, etc. This process is nearly cost-free, as it is lightweight.

### H.2. Additional Details of the Tensor Recovery Experiment

In the MSI recovery, we adopt the MTNL structure described in Appendix F with $E = 2$ and $L = 2$. This structure implicitly models the spatial and spectral dimensions of the MSI separately, aligning with the stronger low-rank inductive bias in the spectral dimension of the MSI images, while adopting a structure similar to the tensor ring and enhancing the connections between spatial and spectral modes with DL-inspired designs.

In real-world image recovery tasks, the images usually involve millions of training samples, which can make the optimization slow. In the following, we first introduce an integrated formulation of the training objective that allows efficient full-batch gradient computation in a single backward pass. After this, we will present the detailed experimental settings.

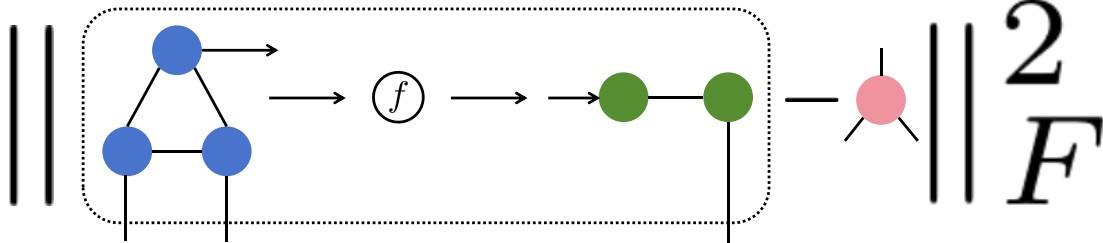

*Figure 7.* Optimization objective of the generalized TN recovery model.

**Efficient optimization algorithm.** In the general MTNL learning setup, a large amount of training data can create a computational bottleneck for the optimization process. Strategies such as stochastic gradient descent can help mitigate this issue. However, they introduce imprecise gradient estimation, which can slow convergence, and in other words, increase the overall training cost. Even though we introduced the concept of the generalized TN recovery model as a special case of the MTNL in Appendix F, it still remains just one step away from the traditional way of using TNs. Specifically, in existing studies, TN representations are directly constructed (contracted) from the tensor cores without the need to propagate each

individual element. Here, we aim to show that this gap in MTNL can be further closed, especially in the tensor recovery setup, making our term of the generalized TN recovery model more appropriate.

Specifically, we will show that the optimization of the MTNL model here takes a more integrated form, as illustrated in Fig. 7. In this optimization objective, we directly construct the representation of the pink tensor by contracting the TN layer by layer and applying the nonlinear activation function element-wisely. In this way, we do not need to propagate the training set, allowing accurate gradient estimation in a single backward propagation, and consequently improving the algorithm's efficiency.

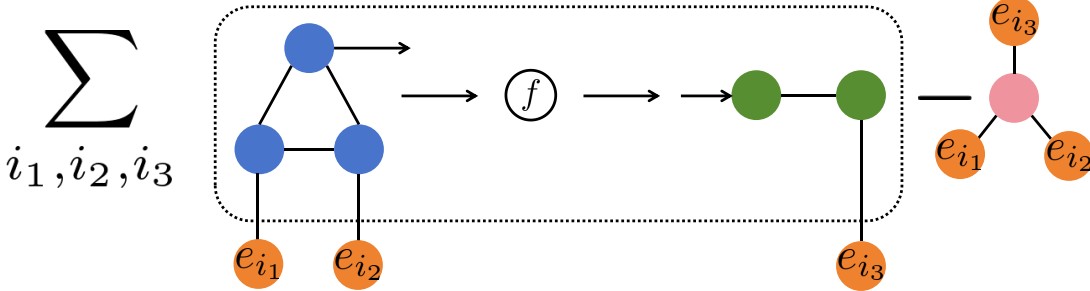

*Figure 8.* Element-wise form of the optimization objective.

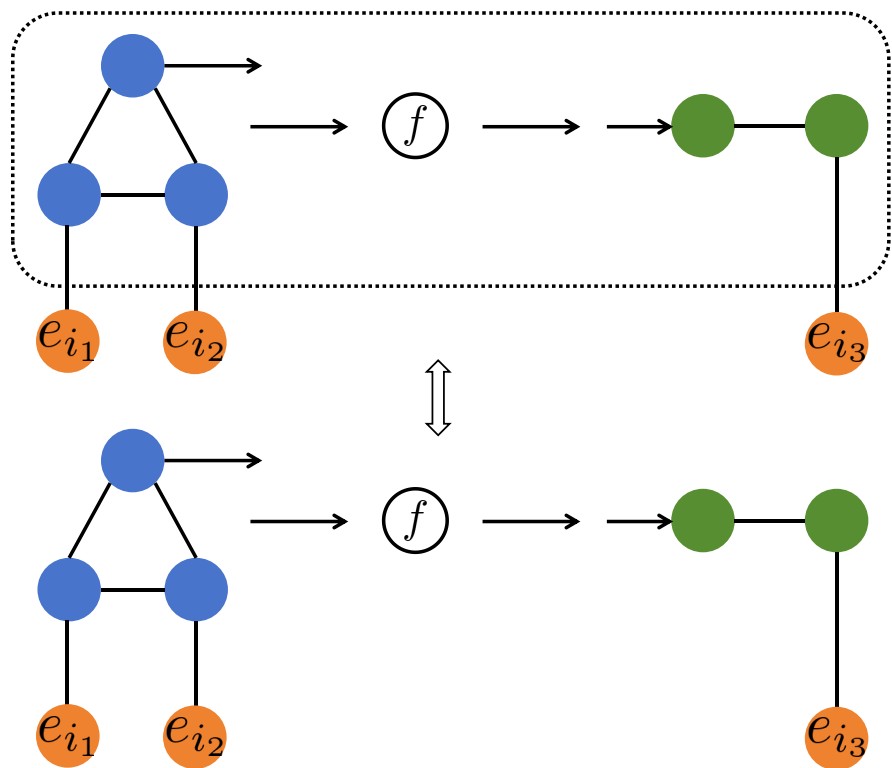

*Figure 9.* Equivalence between the MTNL objective and the generalized TN recovery objective.

First, we expand the optimization objective into its element-wise form, as shown in Fig. 8. Based on it, what we need to show next is the relationship demonstrated in Fig. 9 holds. To be specific, it presents the integrated way of conducting generalized TN recovery and the element-wise way. If the equivalence in Fig. 9 holds, then we can replace the middle term in Fig. 8, and the integrated objective is actually the same as the training-data propagation objective. Consequently, the gradient computed based on the objective in Fig. 7 is an accurate full-batch gradient of the MTNL model.

To illustrate that the relationship demonstrated in Fig. 9 holds, we require that relation 5 in Fig. 10 also holds. Specifically,

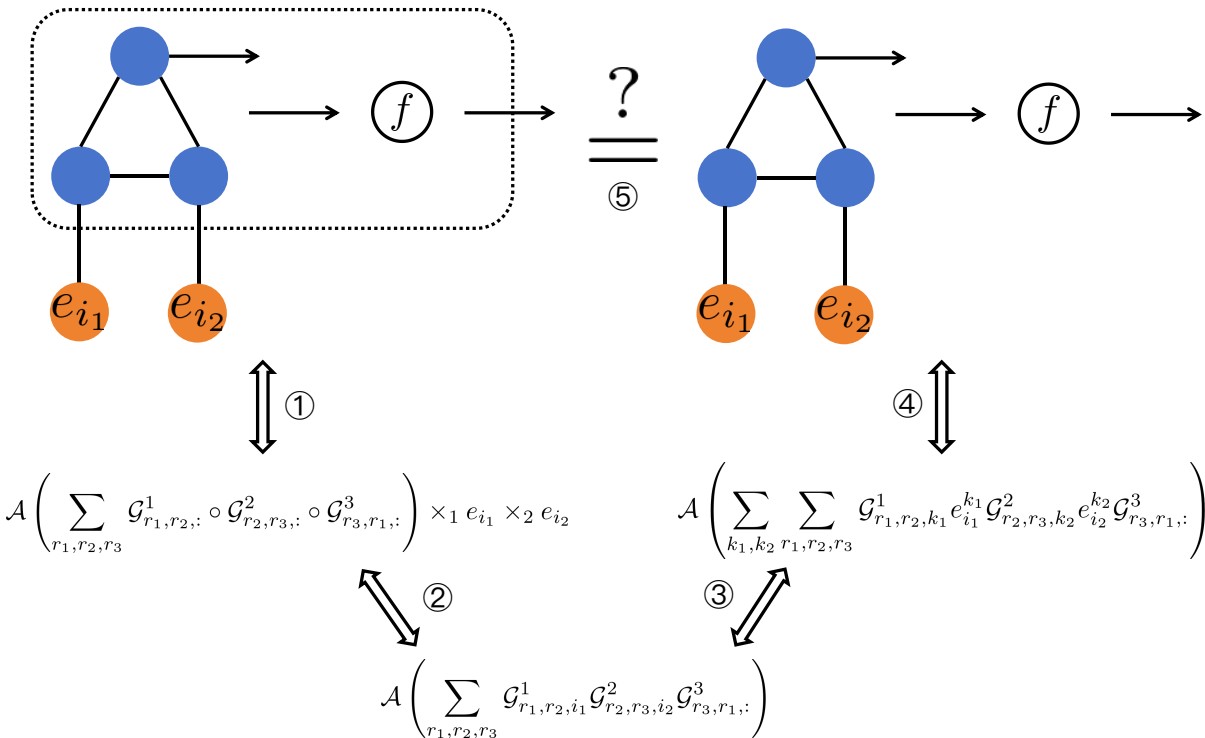

*Figure 10.* Relation between MTNL computation and generalized TN computation.

in Fig. 10, we demonstrate a chain of equalities from 1 to 4, which establishes the equivalence of the computation of 5. First, relation 1 expands the computation in the tensor formation, which is performed by first reconstructing the tensor ring tensor and then applying an activation function $\mathcal{A}(\cdot)$ on it. After that, we multiply the basis vectors along the first and second modes. Equality 2 holds because the mode-3 fibers of a tensor after the activation can be obtained by first taking the original mode-3 fibers before the activation and then applying the activation function to them. Moreover, this expression can be equivalently written as contracting the basis vectors with the corresponding input modes, which establishes equality 3. Finally, relation 4 expresses the computation using the graph.

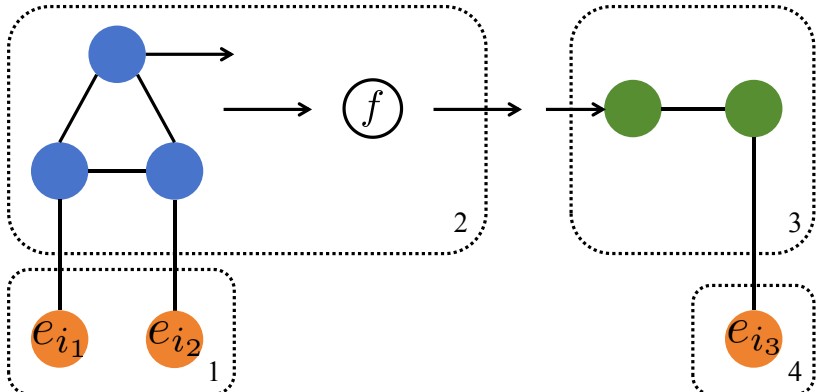

*Figure 11.* Block-wise computation of the generalized TN model.

To demonstrate that the relation in Fig. 9 holds, we first decompose the generalized TN computation in the upper figure of Fig. 9 into a 4-block tensor–contraction process, as shown in Fig. 11. Then, by following the contraction order $1 \rightarrow 2 \rightarrow 3 \rightarrow 4$, we see that the contraction of blocks 1 and 2 can be equivalently transformed into the data–propagation format through equality 5 in Fig. 10. Based on this, the remaining 3-block contraction involving blocks 3 and 4 can be further decomposed, thereby establishing the equivalence in Fig. 9.

In the above demonstration, to simplify the process and directly highlight the core proof idea, we only consider an MTNL model with $E = 1$ and $L = 2$. However, in the MSI recovery experiment, we adopt a model with $E = 2$ and $L = 2$, and additionally include a skip connection to enhance gradient flow. To prove that the integrated optimization objective still applies in this setting, the key step is to show that this modified model also admits a relation analogous to Fig. 10, but now with multiple TNs and the added skip connection. By explicitly writing out the expression as a multi-summation of tensor ring tensors, some passing through the activation $\mathcal{A}(\cdot)$ and some not, and then summing them together before multiplying by the two basis vectors, we obtain the counterpart of equality 1. Next, in the corresponding equality 2, we can change the explicit contraction of the basis vectors form into the explicit index-selection form. Because the expression contains multiple summed tensor ring terms, this index-selection operation can be pushed into each tensor-ring term individually, allowing equality 2 to hold. Similarly, this expression can be rewritten again in the basis-vector-contraction form, analogous to equality 3 in Fig. 10. Finally, we can construct the integrated optimization objective for the MTNL model used in the MSI recovery experiment, thereby accelerating the optimization process.

**Data Generation.** For the experimental images, we select chart_and_stuffed_toy (MSI1), strawberries (MSI2), sushi (MSI3), and flowers (MSI4). Each image is reshaped and rescaled into a tensor of size $256 \times 256 \times 31$ with element values normalized to the range $[0, 1]$.

**Hyperparameter Settings.** We fine-tune the hyperparameters of the comparison methods as follows. For TRALS, we fine-tune the TR-ranks within the range $\{2, 4, 6, 8, 10, 12, 14, 16, 18, 20\}$. For BCPF, we set the maximum CP-rank to 100. For BTucker, we set its rank to 130. For BTT, we set its rank to 125. For TW, we set $\rho = 0.001$ and select TW-ranks from the following configurations: the tensor ring related ranks are chosen from $\{2, 3, 4\} \times \{5, 10, 15, 20, 25\} \times \{2, 3\}$, and the tucker-related ranks are fixed as $\{2, 2, 2\}$, resulting in a total of 30 possible TW-ranks combinations. For BayesTNSS, we tune the rank-cutting threshold within $\{0.1, 0.3, 0.5\}$ and set the upper bound of the ranks to 20.

For the proposed method, we set the TNs' ranks to 14 and initialize the core tensors by independently sampling each element from the distribution $\mathcal{N}(0, 0.1)$. The threshold matrices are initialized as identity matrices. For model optimization, we employ ReLU activations with a skip connection. We use the Adam optimizer with a learning rate of 0.001, and set $\epsilon = 200$, $\tau = 1$, $a = 1$, $b = 1\mathrm{e}{-6}$, and $c = 5$.

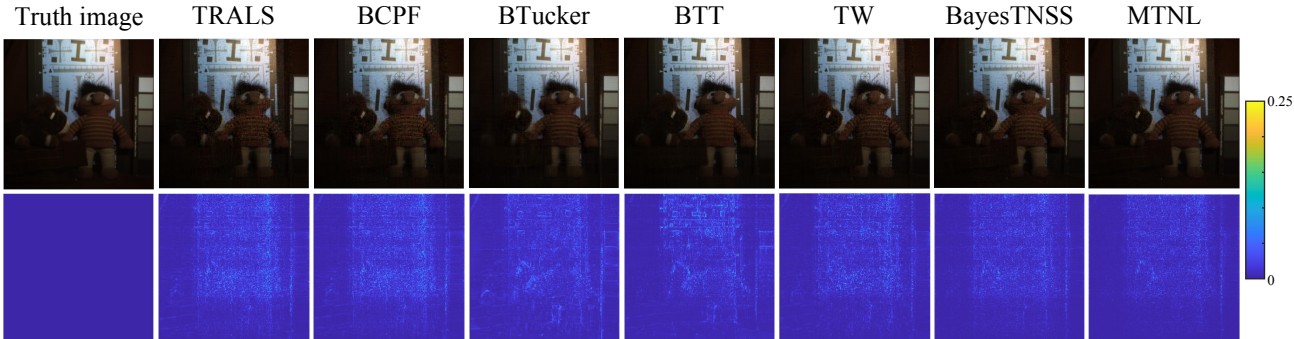

*Figure 12.* Recovered MSIs and residual images obtained by different methods.

**Additional Results.** In Fig. 12, we demonstrate the first three channels of the recovered MSI1 produced by different methods, along with their averaged error images. From the results, we can see that the proposed method achieves superior performance, with clearer image, fewer artifacts and a cleaner error image.

In Table 6, we demonstrate the average running time (in seconds) of different methods for recovering the four MSI images. We can see that MTNL requires a relatively higher runtime compared to other methods. This is mainly caused by the extended convergence process due to the nonlinearity in the model.

How to improve the convergence speed and reduce the running time of MTNL will be the focus of our future work. Here, we present preliminary results showing that the running time can be reduced. Specifically, in the original setting, we set $E = 2$, which may complicate the optimization dynamics, thus, we change it to 1. Moreover, we use a larger learning rate to improve convergence speed. With these changes, MTNL can achieve the same RSE while reducing the running time to 1236 (MR = 0.95) and 996 (MR = 0.85).

*Table 6.* Running time comparison of different methods.

| Missing Rate | TRALS | BCPF | BTucker | BTT | TW | BayesTNSS | MTNL |
|---|---|---|---|---|---|---|---|
| 0.95 | 12 | 113 | 2075 | 313 | 29 | 973 | 2530 |
| 0.85 | 53 | 173 | 2055 | 502 | 38 | 1248 | 1761 |

## H.3. Additional Details of the Parameter-efficient Fine-tuning (PEFT) Experiment

**Hyperparameter Settings.** For the proposed method, we consider compressing each $Q$ and $V$ matrix in LLaMA2 using a single TN. Each TN is initialized with rank 6 and is fully connected. Moreover, we set the hyperparameters as $\tau = 0.05$, $a = 0.2$, $b = 0.1$, $c = 50$, and $\epsilon = 1$. For model optimization, we use the AdamW optimizer and tune the learning rate within {5e-5, 1e-4, 5e-4, 1e-3, 5e-3} on the validation set to obtain the best model performance. To ensure fair comparison, we set the other hyperparameters of the model the same as in QuanTA (Chen et al., 2024a).

**Setup.** To demonstrate the TN-SS ability of the proposed method, we observe from the experimental results that although many elements of the threshold matrices gradually decrease, the training epochs are insufficient for them to reach sufficiently low values. Therefore, we apply a pruning strategy on the final results of the above method by cutting off the values of the threshold matrices that fall below a predefined threshold to check the TN-SS ability of the proposed method. In this experiment, we set the threshold value to $10^{-1}$. For the learned compressed LLaMA2 fine-tuning model, we set $\tau = a = b = c = 0$, while keeping all other procedures the same as those described above for the experiments.

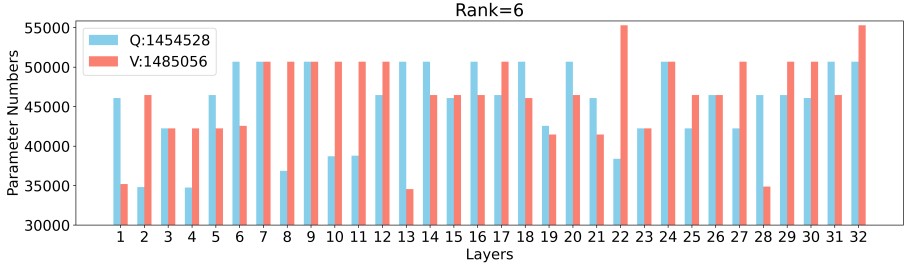

*Figure 13.* Parameter distribution of the learned TN structures for fine-tuning LLaMA2 with an initial rank of 6 on the DROP dataset. The x-axis denotes different layers of the LLaMA2 model, and the y-axis shows the number of TN parameters. The sky blue color represents the number of TN parameters used for fine-tuning the $Q$ matrix, while the salmon color represents those used for fine-tuning the $V$ matrix. The legend shows the total number of parameters used for fine-tuning these two types of matrices.

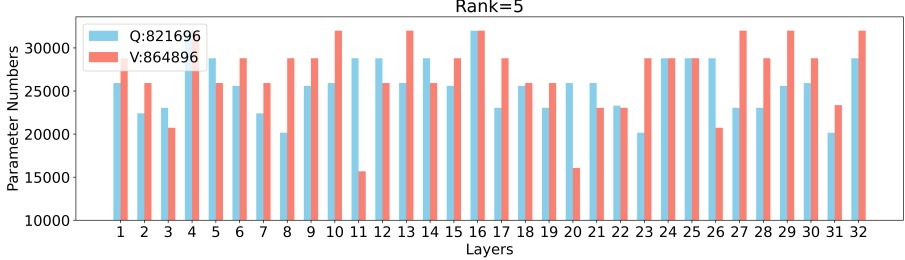

*Figure 14.* Parameter distribution of the learned TN structures for fine-tuning LLaMA2 with an initial rank of 5 on the DROP dataset. The x-axis denotes different layers of the LLaMA2 model, and the y-axis shows the number of TN parameters. The sky blue color represents the number of TN parameters used for fine-tuning the $Q$ matrix, while the salmon color represents those used for fine-tuning the $V$ matrix. The legend shows the total number of parameters used for fine-tuning these two types of matrices.

**Additional Results.** In Fig. 13, we present the learned TN structures for fine-tuning LLaMA2 on the DROP dataset. As shown in the figure, different layers and types of fine-tuning matrices require TNs with varying levels of complexity. Existing tensor-based methods typically rely on a fixed TN structure, which hinders their ability to flexibly adapt to this scenario, leading to suboptimal performance. In contrast, this work proposes a more expressive TN learning model that

satisfies these expressiveness requirements and learns diverse intrinsic TN structures through the flexible TN-SS prior, thus achieving superior fine-tuning performance.

Moreover, Fig. 13 shows that fewer parameters are used to fine-tune the $Q$ matrices compared to the $V$ matrices. This aligns with findings from the LoRA study (Hu et al., 2022), which indicate that the $V$ matrices are more critical for model performance and should be compressed less. A similar parameter allocation pattern is observed when we apply MTNL with an initial rank of 5, as shown in Fig. 14. These results further support the importance of the flexible TN-SS mechanism of the proposed method.

### H.4. Additional Details of the Tensor Regression Experiment

$$\text{Layer 1} = \begin{bmatrix} 20 & 2 & 2 \\ 2 & 20 & 2 \\ 2 & 2 & 2 \end{bmatrix} \quad \text{Layer 2\&3} = \begin{bmatrix} 2 & 2 & 2 \\ 2 & 2 & 2 \\ 2 & 2 & 2 \end{bmatrix} \quad \text{Layer 4} = \begin{bmatrix} 2 & 2 & 2 \\ 2 & 1 & 2 \\ 2 & 2 & 1 \end{bmatrix} \tag{71}$$

$$\text{Layer 1} = \begin{bmatrix} 20 & 2 & 2 \\ 2 & 20 & 2 \\ 2 & 2 & 2 \end{bmatrix} \quad \text{Layer 2\&3\&4} = \begin{bmatrix} 2 & 2 & 2 \\ 2 & 2 & 2 \\ 2 & 2 & 2 \end{bmatrix} \quad \text{Layer 5} = \begin{bmatrix} 2 & 2 & 2 \\ 2 & 2 & 2 \\ 2 & 2 & 1 \end{bmatrix} \tag{72}$$

**Data Generation.** In this experiment, the TN structures for each layer of the MTNL with configurations: $(E, L) = (4, 4)$ and $(E, L) = (5, 5)$ are demonstrated in Eq. 71 and Eq. 72, respectively. After defining the TN structures for MTNL, we initialize the core tensors and the threshold matrices. For the core tensors, each element is independently sampled from the distribution $\mathcal{N}(0, 0.3)$. The threshold matrices are initialized as identity matrices, with all diagonal elements set to one.

Once a specific MTNL is initialized, we propagate a set of 50 sample random $20 \times 20$ matrices sampled i.i.d. from $\mathcal{N}(0, 1)$ through the MTNL and collect the corresponding 50 output values as the training set. For the test set, we propagate a set of 20 sample random $20 \times 20$ matrices sampled i.i.d. from $\mathcal{N}(0, 1)$, $\mathcal{N}(3, 1)$, $\mathcal{N}(6, 1)$, and $\mathcal{N}(9, 1)$, and collect the corresponding 20 output values for four test sets.

For the larger-scale experiments, we adopt the same MTNL structure as in Eq. 72, but increase the input feature size to 64 and raise the TNs' ranks to 4. Moreover, we generate a training set of size 1000. Beyond these changes, all other data generation settings remain the same as in the small-scale experiments.

**Hyperparameter Settings.** For the proposed method, we consider TN structures of MTNL with rank-plus-one based on the data generation MTNL structures. Moreover, we initialize the core tensors by independently sampling each element from the distribution $\mathcal{N}(0, 0.1)$. The threshold matrices are initialized as identity matrices, with all diagonal elements set to one. For model optimization, we use the Adam optimizer with a learning rate of 0.001, and the maximum number of iterations is set to 5000. Moreover, we adopt a unified setting with $\tau = 0.0001$, $a = 1$, $b = 0$, $c = 5$, and $\epsilon = 100000$. For uncertainty estimation, we set the dropout rate $\theta = 0.1$ and use 20 samples. For the SVGD algorithm, we set the number of particles to 20 and tune the descending step size over $\{0.1, 0.001, 0.0001\}$, and we use the RBF kernel.

*Table 7.* Change in variance under different levels of noise.

| Dataset | Noise | | |
|---|---|---|---|
| | **10** | **20** | **30** |
| CT Data | 11834 | 41015 | 127451 |
| Superconductivity Data | 957 | 1268 | 1796 |

**Additional Results.** Here we include additional results on tensor regression experiments using real-world regression datasets. Two datasets from the UC Irvine Machine Learning (UCI) Repository are used: *Relative location of CT slices on axial axis* and *Superconductivity Data*. After training, we gradually add increasing levels of noise to the test data and report the variance obtained by MTNL-Dropout in Table 7. In the table, we can see that MTNL-Dropout still captures the uncertainty with increased variances. Moreover, for the CT data, the proposed method achieves an NLL of 6.1, while SVGD

achieves an NLL of 6.0 but requires $48\times$ more computational resources. For the Superconductivity Data, the proposed method achieves an NLL of 5.4, while SVGD achieves an NLL of 5.1 but requires $60\times$ more computational resources.

### H.5. Additional Details of the Ablation Experiment

$$\text{Layer 1} = \begin{bmatrix} 6/8 & 3 & 3 \\ 3 & 6/8 & 3 \\ 3 & 3 & 2 \end{bmatrix} \quad \text{Layer 2} = \begin{bmatrix} 6/8 & 3 \\ 3 & 2 \end{bmatrix} \tag{73}$$

In the ablation experiments, we adopt the data generating MTNL structure shown in Eq. 73 with $E = 2$ and $L = 2$, similar to the structure used in the MSI recovery experiment. After defining the TN structures, we initialize the core tensors and threshold matrices. Each element of the core tensors is independently sampled from $\mathcal{N}(0, 0.1)$. The threshold matrices are initialized as identity matrices. The tensor elements are then generated by feeding the coordinate tensor into MTNL and forward propagating the results. To fit the tensors, we consider a rank-plus-one MTNL structure and use the adam optimizer with a learning rate of 0.001. We set $\epsilon = 10^{10}$, $\tau = 1$, $a = 10^{-6}$, $b = 10^{-6}$, and $c = -2$.

*Table 8.* Ablation study on the model width $E$.

| Metric | 1 | 2 | 4 | 10 |
|---|---|---|---|---|
| RSE | 0.329 | 0.302 | 0.283 | 0.335 |

*Table 9.* Ablation study on the model depth $L$.

| Metric | 1 | 2 | 4 | 10 |
|---|---|---|---|---|
| RSE | 0.357 | 0.302 | 0.498 | 0.641 |

*Table 10.* Ablation study on model nonlinearity.

| Metric | W | W/O |
|---|---|---|
| RSE | 0.302 | 0.433 |

*Table 11.* Ablation study for $a$.

| Metric | 1e-6 | 1e-3 | 1 | 1e3 | 1e6 |
|---|---|---|---|---|---|
| RSE | 0.303 | 0.303 | 0.302 | 0.338 | 0.339 |

*Table 12.* Ablation study for $b$.

| Metric | 1e-6 | 1e-3 | 1 | 1e3 | 1e6 |
|---|---|---|---|---|---|
| RSE | 0.302 | 0.301 | 0.302 | 0.303 | 0.303 |

*Table 13.* Ablation study for $c$.

| Metric | -15 | -5 | 5 | 15 | 25 |
|---|---|---|---|---|---|
| RSE | 0.347 | 0.340 | 0.302 | 0.309 | 0.309 |

**Sensitivity Analysis.** Here we conduct a sensitivity analysis on the model hyperparameters, including the width $E$, depth $L$, with or without nonlinearity and the threshold prior hyperparameters $a$, $b$, and $c$. We take MTNL with $E = L = 2$ as the "center" configuration and vary different hyperparameter settings around it for MSI recovery. The ablation results for these hyperparameters are presented in Tables 8, 9, 10, 11, 12, and 13, respectively.

It can be seen from the results that both $E$ and $L$ are beneficial to model performance. However, overly deep or wide models may cause overfitting, with the depth having a larger impact on performance, especially for MSI recovery using the MTNL layer structure adopted in this paper. Moreover, as shown in Table 10, nonlinearity is essential for MTNL.

Furthermore, as shown in Tables 11, 12, and 13, within the threshold prior, hyperparameter $c$ mainly determines the sparsity level. A smaller $c$ leads to stronger sparsity, and vice versa. From the results, larger $c$ values generally lead to better performance, indicating that the model needs to fit more complex patterns. However, excessively large $c$ values may also degrade performance. Hyperparameter $a$ controls the tail behavior. A smaller $a$ preserves larger signals, which is beneficial for fitting, while a larger $a$ tends to weaken this effect. Hyperparameter $b$ controls the mass of the distribution around zero. Smaller $b$ values lead to stronger sparsity near zero, although the influence of $b$ on the overall model performance is relatively weaker compared to $c$ and $a$. Overall, the prior combine hyperparameters $a$, $b$, and $c$ to induce different sparsity behaviors.

