# OpenReview forum: "MTNL: A Unified Modeling Perspective for Enhancing Tensor Network Learning"
_ICML.cc/2026/Conference — ICML 2026 regular_

### Official Review · Reviewer_4B11 · 2026-03-05

**Soundness:** 3
**Presentation:** 3
**Significance:** 3
**Originality:** 3
**Overall Recommendation:** 4
**Confidence:** 4

**Summary:**

This paper proposes mixed tensor network learning (MTNL), attempting to merge the unsupervised and supervised research directions of tensor networks (TNs) from a unified modeling perspective. At the model level, it fuses multiple TNs in a deep-network style to enhance expressive power , while introducing a more flexible tensor network structure search (TN-SS) prior within a Bayesian framework and providing corresponding theoretical analysis. Furthermore, the authors state that the maximum a posteriori (MAP) training objective is connected to its dropout-based counterpart problem, leading to a lightweight uncertainty estimation method that requires only simple post-processing. This framework is experimentally validated on tasks including tensor recovery, parameter-efficient fine-tuning (PEFT) for LLMs, and tensor regression/uncertainty estimation.

**Compliance With Llm Reviewing Policy:**

Affirmed.

**Final Justification:**

This paper proposes a unified tensor network learning framework with stronger model expressiveness, flexible structure learning, and efficient uncertainty estimation. I find the paper interesting and technically solid, with clear originality and useful theoretical contributions.

(1) The main strengths are the unified modeling perspective, the nontrivial theoretical analysis, and the practical uncertainty estimation method.

(2) My original concerns were about limited sensitivity analysis, incomplete ablations, and relatively narrow experimental coverage.

(3) After the rebuttal, I think these concerns were addressed sufficiently well. The added sensitivity studies, clearer ablation explanation, and broader additional results improved my confidence in the work.

Overall, the rebuttal positively changed my assessment. Some weaknesses remain, but they no longer outweigh the strengths, so I increased my score.

**Key Questions For Authors:**

1. A more systematic ablation study is recommended to isolate the primary source of performance gains.
2. The inaccurate claim of using "dynamic pruning" to mitigate $\theta=1$ should be corrected, as the actual implementation appears to be post-processing pruning followed by retraining.
3. Since results are currently limited to LLaMA2-7B on the DROP dataset, it is necessary to clarify whether the consistent structural allocation patterns and performance gains hold across a wider range of tasks and datasets.

**Limitations:**

yes

**Strengths And Weaknesses:**

Strengths:
1. The research problem and motivation are clear. The paper bridges the gap between unsupervised and supervised tensor network learning from a unified modeling perspective.
2. The theoretical analysis provided is substantial. Specifically, Theorem 2.2 regarding implicit structure learning is an interesting result, revealing that the model implicitly applies an $l_{2/T}$ regularization and introduces an adaptive layer-wise sparsity-reweighting mechanism.
3. The proposed MC-dropout-style post-processing algorithm is a practical contribution. The authors demonstrate that it can reduce the computational cost by up to 200x compared to the SVGD baseline.

Weaknesses:
1. The paper lacks a systematic sensitivity analysis for its core hyperparameters. The impact of varying the network width ($E$), depth ($L$), and the Bayesian prior hyperparameters on the final performance is not sufficiently explored.
2. The evaluated models and datasets are narrow. For instance, the PEFT experiments are restricted to LLaMA2-7B on DROP, and the regression experiments rely on synthetically generated Gaussian data.
3. Despite highlighting the combined contributions of deep/nonlinear multi-TN fusion, flexible priors, and dropout UQ, the paper lacks comprehensive ablations to isolate the primary source of performance gains and reveal how these components interact.
4. The paper points out that a large magnitude difference between $\max(\lambda)$ and $\min(\lambda)$ can result in $\theta=1$, making the results meaningless and requiring dynamic pruning during training. However, the paper fails to analyze the actual impact of this situation.

---

> ### Author Rebuttal · Authors · 2026-03-30
>
> We sincerely appreciate the valuable comments provided by the reviewer. Below, we provide a point-by-point response to the main concerns raised in the review. We have added **additional ablation studies and experimental results** to enhance the paper's completeness. Due to the 5000 character limit, we refer to your comments using the corresponding indices and aim to make the response as informative as possible to assist with your evaluation. Thanks again.
>
> > **Weaknesses1: The..**
>
> Thank you. In the follow, we provide a sensitivity analysis over the network width $E$, network depth $L$, and Bayesian prior hyperparameters. We consider MTNL with $E,L=2$ as the ”center”, and vary different model settings around it to recover the MSI. The RSEs are as below:
>
> Varying $E$
> |1|2|3|4|
> |-|-|-|-|
> |0.329|0.302|0.281|0.283|
>
> Varying $L$
> |1|2|3|4|
> |-|-|-|-|
> |0.357|0.302|0.434|0.498|
>
> Varying Bayesian sparsity prior hyperparameters (a,b,c)
> |1e-6,1e-6,-2|1,1e-6,5|1,1e-6,1|1e-6,1,-5|
> |-|-|-|-|
> |0.327|0.302|0.309|0.317|
>
> It can be seen that increasing the $E$, $L$ can enhance model performance, while too deep and wide models may cause overfitting. Varying the sparsity prior hyperparameters injects different structure learning abilities into the method, which affects performance, and we mainly select the best hyperparameters from these four choices.
>
> > **Weaknesses2: The..**
>
> Thank you. In the follow, we present the results of PEFT on LLaMA2-7B for the Math10K dataset.
>
> | |#Para|AQuA|GSM8K|MAWPS|SVAMP|Avg.|
> |-|-|-|-|-|-|-|
> |QuanTA|0.19%|16.7| 67|94.3|80.3|64.6|
> |MTNL|**0.7%**|19.7| 65.2|93.7|80.9|**64.9**|
>
> From the table, we can see that MTNL achieves a higher average accuracy than QuanTA with fewer parameters. Due to time constraints, we can not conduct more PEFT experiments. We will extend the results in the revised version of the work.
>
> For regression, we test on two real-world regression datasets from the UCI repository and report the performance of the proposed method in the following:
>
> CT Data
> |Noise|10|20|30|
> |-|-|-|-|
> |**Variance**|11834|41015|127451|
>
> Superconductivity Data
> |Noise|10|20|30|
> |-|-|-|-|
> |**Variance**|957|1268|1796|
>
> In the table, we gradually add larger noise to the test data to create outliers, and we can see that MTNL still captures the uncertainty with increase variances. Moreover, for CT data, the NLL achieved by the proposed method is 6.1, while the NLL achieved by SVGD is 6.0 but requires 48$\times$ more computational resources.
>
>
> > **Weaknesses3: Despite..**
>
> Thank you. We first conduct ablations to identify the sources of performance gains in the multi-TN model. Specifically, we generate synthetic data using different model settings, then apply traditional TN method to recover the synthetic data. The high recovery RSE can help to reflect the complexity of the original model.
>
> |Settings|1|2|3|4|5|
> |-|-|-|-|-|-|
> |Depth|X|√|√|√|√|
> |Width|√|X|X|√|√|
> |Nonlinearity|X|X|√|X|√|
> |RSE|0.611|0.749|0.879|0.784|0.884|
>
> Comparing settings 2, 3 and 4, 5, we can see that nonlinearity is the key factor that separates the traditional methods from MTNL. On the other hand, depth and width can enhances the expressiveness of the model but too deep of a model may overfit the data and make training difficult (as reflect in response 1), and adding width can alleviate this issue.
>
> Further, for flexible priors, they serve as a core auxiliary mechanism for adjusting the model complexity to better fit the data. For dropout UQ, it does not affect performance but is used for outputting uncertainty.
>
> > **Weaknesses4: The..**
>
> Theoretically, with a large magnitude difference, if no pruning, we must set $\theta=1$ to construct the lower-bound problem. To show the actual effect on UQ, we report the following UQ results (using the NLL metric) along with the dropout rate used (square bracket).
>
> |Pruning|W/O Pruning|
> |-|-|
> |5.33[0.4]|5.79[0.6]|
>
> The results show that the pruning strategy uses a lower dropout rate and achieves better NLL. This is because when zero rank-slices exist, the sample becomes less efficient, which may affect the UQ results and increases dropout rate.
>
>
> > **Q1: A..**
>
> Thank you for your valuable suggestions. A more systematic ablation study is in the response letter.
>
> > **Q2: The..**
>
> In the PEFT experiment, we use post-processing pruning due to the low training epoch setting. In other situations, such as the regression experiments in the paper, we can use dynamic pruning. We will clarify these points in the revised version of the paper.
>
> > **Q3: Since..**
>
> For the Math10K dataset in the response, the method allocates 935680 and 860032 parameters to the Q, V matrices. As we analyzed, their difference becomes larger during the training process, increasing from 45184 to 75648, showing an allocation preference. Compared to the pattern for DROP, this may indicate that the allocation preference is data-dependent. In the revised version, we will extend the analysis to a wider range of tasks.

---

> > ### Author Rebuttal · Reviewer_4B11 · 2026-04-02
> >
> > Thank you for the detailed follow-up response. The revised rebuttal is clearer and more helpful.
> >
> > (1) The extended sensitivity analysis over (E), (L), and the prior settings addresses my earlier robustness concern better than the first rebuttal.
> >
> > (2) The additional ablation discussion makes the roles of model bias and sparsity prior clearer, which helps explain the source of the gains.
> >
> > (3) The added results beyond the original setup improve the empirical support and make the paper’s scope clearer.
> >
> > Overall, while some limitations remain, I think the authors have made a meaningful effort to address the main concerns, so I have increased my score. Best of luck with the paper, and thank you for the constructive discussion.

---

> > > ### Author Response · Authors · 2026-04-03
> > >
> > > Thank you for your comments. In the following response, we provide a more **in-depth ablation analysis, summarize the main results from the first response and include some additional results.**
> > >
> > > > **..limited in scope with only a small set of E,L and prior settings..not yet a systematic robustness study. The main paper also uses fixed settings..**
> > >
> > > **E and L Sensitivity Analysis Extension.** In the following, we extend the range of the ablation study for both E and L (1-4 to 1-10), and the RSEs are as follows:
> > >
> > > Varying E
> > > |1|2|3|4|6|8|10|
> > > |-|-|-|-|-|-|-|
> > > |0.329|0.302|0.281|0.283|0.307|0.314|0.335|
> > >
> > > Varying L
> > > |1|2|3|4|6|8|10|
> > > |-|-|-|-|-|-|-|
> > > |0.357|0.302|0.434|0.498|0.548|0.621|0.641|
> > >
> > > We can see that E and L are beneficial to model performance. However, as model expressiveness increases, large E and L can lead to overfitting. In comparison, L is more affected. These results suggest that in practice, we should find a balance when choosing the model structures, considering more E for L.
> > >
> > > **About fixed settings as E,L=2.** the main purpose of the MSI recovery experiments is to show that the new concepts of width and depth are indeed valuable. But based on the ablation results, E,L=2 is also a good choice.
> > >
> > > **Prior Settings Sensitivity Analysis Extension.** Here, we conduct a wider range ablation experiment for the hyperparameters a,b,c, using a=1,b=1e-6,c=5 as the ablation “center”. The RSEs are as below:
> > >
> > > Varying c
> > > |-15|-5|5|15|25|
> > > |-|-|-|-|-|
> > > |0.347|0.340|0.302|0.309|0.309|
> > >
> > > Varying a
> > > |1e-6|1e-3|1|1e3|1e6|
> > > |-|-|-|-|-|
> > > |0.303|0.303|0.302|0.338|0.339|
> > >
> > > Varying b
> > > |1e-6|1e-3|1|1e3|1e6|
> > > |-|-|-|-|-|
> > > |0.302|0.301|0.302|0.303|0.303|
> > >
> > > In the prior, c determines the sparsity level. A smaller c leads to stronger sparsity, and vice versa. Thus from the results, **larger c generally leads to better performance, indicating that the model needs to fit more complex patterns, however, overly large c can also degrade performance.** Parameter ‘a’ controls the tail behavior. **A smaller ‘a’ preserves larger signals, which is beneficial for fitting, while larger ‘a’ tends to reduce this effect.** Parameter b controls the mass of the distribution around zero. **Smaller b leads to stronger sparsity near zero, but b’s influence on overall model performance is lighter compared to c and ‘a’.** Overall, the proposed prior combines ‘a’, b, and c to produce different sparsity-inducing behaviors.
> > >
> > >
> > > > **..still does not fully isolate the main source of the gains..**
> > >
> > > At a higher level, the performance gains of the proposed method come from **two aspects, the model bias and the sparsity prior**. Both components work together and are indispensable in TN learning.
> > >
> > > **Isolating the main source of gains from model bias.** In our previous response, we have shown that depth and width significantly affect model expressiveness, and nonlinearity is key for enhancing model performance.
> > >
> > > **Isolating the main source of gains from the sparsity prior.** Table 5 in the main paper shows that removing the threshold prior leads to a more significant performance drop, indicating that it plays a crucial role in the overall sparsity prior. However, using only the threshold prior does not achieve the best performance, demonstrating the importance of combining both the core and threshold priors.
> > >
> > > In summary, we **fully isolate and analyze** the main sources of the gains from the two complementary aspects, which directly contribute to the final model performance gains. Moreover, the results also indicate that the combination of all proposed components leads to an overall synergistic improvement.
> > >
> > >
> > > > **The rebuttal format is also difficult to follow..**
> > >
> > > A core result in the first rebuttal focuses on extending our method to **more models and datasets.**
> > >
> > > **For PEFT**, we present results using LLaMA2-7B on the **Math10K dataset**. The results show that MTNL achieves higher average accuracy than QuanTA while using fewer parameters.
> > >
> > > Moreover, we also extend our method to **CLIP’s vision and text encoders for a few-shot classification** task, and the results are provided below.
> > >
> > > ||RandLoRA [1]|MTNL|
> > > |-|-|-|
> > > |Acc|69.0| **69.4**|
> > > |#Para|3.45%|**0.82%**|
> > >
> > > It can be seen that the proposed method also achieves superior results compared to the SoTA RandLoRA.
> > >
> > > **For regression,** we test on **two real-world datasets** and demonstrate the proposed method can capture uncertainty with increased variances, and also requires 48x less resources compared to SVGD.
> > >
> > >
> > > **To summarize the rebuttal**, we have added a systematic sensitivity analysis over the model components and isolated the sources of the performance gains. Moreover, we evaluate our method on two types of PEFT settings, covering both vision and NLP tasks, across two datasets. We also verify the uncertainty quantification ability on two real datasets. These help demonstrate the effectiveness and generality of the proposed method.
> > >
> > >
> > > [1]. RandLoRA: Full rank parameter-efficient fine-tuning of large models. ICLR 2025

---

### Official Review · Reviewer_hC77 · 2026-03-10

**Soundness:** 4
**Presentation:** 4
**Significance:** 3
**Originality:** 4
**Overall Recommendation:** 6
**Confidence:** 5

**Summary:**

This paper proposes a unified modeling framework named Mixed Tensor Network Learning (MTNL), aiming to integrate the strengths of tensor networks in both unsupervised and supervised learning domains. By incorporating core concepts from deep learning—depth, breadth, and nonlinearity—the authors construct a more expressive Bayesian TN learning framework. Key contributions include: a flexible Gaussian-GIG prior capable of automatically learning multiple TN structures, and a lightweight uncertainty quantification mechanism derived from theoretical proofs. Experiments demonstrate MTNL's superior performance in multispectral image restoration, parameter-efficient fine-tuning of large language models, and tensor regression tasks.

**Compliance With Llm Reviewing Policy:**

Affirmed.

**Final Justification:**

The authors' response addresses my concerns, particularly regarding the selection of Gaussian-GIG prior hyperparameters and their impact on model performance. The addition of a systematic sensitivity analysis will help clarify the role of these parameters. The use of the EM algorithm to reduce model tuning complexity is also convincing. The authors' approach to mitigating gradient issues in deep MTNL structures through Gaussian initialization strengthens my confidence in the model's stability. Although a discussion on computational efficiency would still be valuable, overall, I am satisfied with the response.
This work represents a new exploration in the field of tensor networks and introduces a novel research direction in tensor analysis; therefore, I have increased my score.

**Key Questions For Authors:**

1. How were the parameters a, b, and c in the Gaussian-GIG prior selected? If the data missing rate increases from 85% to 90+%, can this parameter set still learn a reasonable structure through the implicit mechanism of Theorem 2.2?
2. Are gradient vanishing or exploding issues more severe in deep MTNL structures compared to standard TN decompositions? What initialization strategies did you employ to mitigate these problems?
3. Providing computational resource and efficiency metrics would significantly enhance the paper's completeness.

**Limitations:**

yes

**Strengths And Weaknesses:**

Strengths
- The paper bridges the modeling gap between unsupervised and supervised tasks for TNs, offering a universal deep modeling perspective for high-dimensional data processing through multi-layer TN fusion.
- The proposed MTNL-Dropout algorithm achieves uncertainty estimation without significantly increasing computational burden, reducing computational cost by nearly 200 times compared to traditional SVGD methods.
- Experiments not only achieve SOTA performance in image restoration but also demonstrate superiority over QuanTA in fine-tuning LLaMA2-7B, proving the method's potential in modern large-scale machine learning tasks.

Weaknesses
- The Gaussian-GIG prior introduces multiple hyperparameters (e.g., a, b, c). While specific values are provided experimentally, a systematic sensitivity analysis on how these parameters influence final structural choices is lacking.
- The introduction of depth and width significantly increases design freedom (number of layers L, number of TNs per layer E). While this enhances performance, it also increases model tuning complexity and computational overhead.

---

> ### Author Rebuttal · Authors · 2026-03-30
>
> We sincerely appreciate the valuable comments provided by the reviewer. Below, we provide a point-by-point response to the review.
>
> > **The Gaussian-GIG prior introduces multiple hyperparameters (e.g., a, b, c). While specific values are provided experimentally, a systematic sensitivity analysis on how these parameters influence final structural choices is lacking.**
>
> Thank you for your comments. In this paper, the Gaussian-GIG prior hyperparameter set {a,b,c} is chosen from the following four sets: {1e-6,1e-6,-2} {1,1e-6,5} {1,1e-6,1} {1e-6,1,-5}. These settings allow the Gaussian-GIG prior to exhibit four different sparsity-inducing behaviors from the Laplacian distribution, Students’s t distribution, and their variants. We demonstrate the ablation results for these hyperparameter choices on recovering an MSI image in the following table, and the corresponding RSE values are reported.
>
> Varying Gaussian-GIG prior hyperparameters
> |1e-6,1e-6,-2|1,1e-6,5|1,1e-6,1|1e-6,1,-5|
> |-|-|-|-|
> |0.327|0.302|0.309|0.317|
>
> From the table, we can see that different hyperparameter settings affect the final model performance, implying that different model structures are used to represent the data. The Gaussian-GIG prior provides the flexibility to adapt to the data.
>
> Broadly speaking, the parameters a, b, and c play different roles in the prior. Parameter c determines the sharpness of the distribution at zero, where lower values strengthen sparsity. Parameter a governs the tail heaviness, determining whether large signals are preserved. Parameter b controls the concentration of mass near zero, with smaller values enhancing sparsity. In the revised version of the paper, we will include a more systematic sensitivity analysis of these hyperparameters.
>
> > **The introduction of depth and width significantly increases design freedom (number of layers L, number of TNs per layer E). While this enhances performance, it also increases model tuning complexity and computational overhead.**
>
> Thank you for your comments. The tensor recovery experiments demonstrate that we can use a low-width and low-depth MTNL to achieve superior performance. Furthermore, through several ablation experiments, we identify that considering width and depth alone can also benefit model performance, which helps reduce the design degrees of freedom of the model.
>
> Moreover, we investigate an EM algorithm to update the hyperparameters a, b, and c automatically. Prior studies demonstrate that, even with a non-informative prior setting, the algorithm can still find a good structure from an over-parameterized initialization. This reduces the tuning complexity of MTNL, making it comparable to existing methods. We will extend this discussion in the revised version of the paper.
>
> > **How were the parameters a, b, and c in the Gaussian-GIG prior selected? If the data missing rate increases from 85% to 90+%, can this parameter set still learn a reasonable structure through the implicit mechanism of Theorem 2.2?**
>
> Thank you for the interesting question. In this paper, the Gaussian-GIG prior hyperparameter set {a, b, c} is chosen from the following four sets: {1e-6,1e-6,-2} {1,1e-6,5} {1,1e-6,1} {1e-6,1,-5}.
>
> If the data missing rate increases from 85% to over 90+%, we first tune the noise precision $\epsilon$ to a more suitable value. Furthermore, we also investigate whether the implicit mechanism described in Theorem 2.2 alone can learn a reasonable structure. Specifically, we conduct an MSI recovery experiment with a missing rate of 99%, set the regularization related to the threshold to 0, and observe that the model achieves a recovery RSE of 0.5539 (by shrinking the model structure), whereas the model without any regularization achieves an RSE of 0.9658.
>
> > **Are gradient vanishing or exploding issues more severe in deep MTNL structures compared to standard TN decompositions? What initialization strategies did you employ to mitigate these problems?**
>
> Thank you for the question. Deep MTNL structures indeed have gradient issues. Currently, we use Gaussian initialization and try several different variances to alleviate these problems.
>
> > **Providing computational resource and efficiency metrics would significantly enhance the paper's completeness.**
>
> Thank you for your valuable suggestion. In the revised version of the paper, we will include computational resource and efficiency metrics.

---

> > ### Author Rebuttal · Reviewer_hC77 · 2026-04-04
> >
> > The authors' response addresses my concerns, particularly regarding the selection of Gaussian-GIG prior hyperparameters and their impact on model performance. The addition of a systematic sensitivity analysis will help clarify the role of these parameters. The use of the EM algorithm to reduce model tuning complexity is also convincing. The authors' approach to mitigating gradient issues in deep MTNL structures through Gaussian initialization strengthens my confidence in the model's stability. Although a discussion on computational efficiency would still be valuable, overall, I am satisfied with the response.
> > This work represents a new exploration in the field of tensor networks and introduces a novel research direction in tensor analysis.

---

> > > ### Author Response · Authors · 2026-04-06
> > >
> > > We sincerely thank the reviewer for the constructive feedback and recognition of our work and responses. We have included a detailed discussion of the computational efficiency of the proposed method. In short, **the computational time of the proposed method can reach a level comparable to existing methods. This work focuses more on the theoretical level, but further improving efficiency is very promising**, as shown in our response to reviewer LwRN. Thank you again!

---

### Official Review · Reviewer_Uj5J · 2026-03-12

**Soundness:** 2
**Presentation:** 2
**Significance:** 2
**Originality:** 2
**Overall Recommendation:** 3
**Confidence:** 1

**Summary:**

This paper focuses on merging supervised and unsupervised learning within the Tensor Network (TN) framework, which have traditionally been treated separately. This framework enhances model expressiveness by fusing multiple TNs in a deep-network style, incorporating depth, nonlinearity, and width into traditional tensor modeling. It introduces a flexible Bayesian prior that enables provable structure search and adaptive parameter allocation across different layers. Furthermore, this paper establishes a theoretical link between the model’s optimization and dropout, inducing a lightweight and efficient uncertainty quantification mechanism.

**Compliance With Llm Reviewing Policy:**

Affirmed.

**Final Justification:**

I maintain a neutral stance on this paper and keep my original score.

**Key Questions For Authors:**

See weakness

**Limitations:**

yes

**Strengths And Weaknesses:**

To be honest, the core of this paper is a bit outside my primary area of expertise, but I have done my best to go through the material and provide what I hope is constructive feedback.

Strengths:
1. I found it quite impressive that the proposed MTNL allows for a simple post-processing of the solution using model dropout. This seems like a very practical way to handle things.
2. The theoretical link the authors draw between the MAP solution of MTNL and its dropout-based optimization counterpart is a nice touch and seems solid from what I can follow.

Weaknesses
1. The writing overall could use some work, as it got a bit confusing in places. For instance, the paragraph describing the "first contribution" is worded in a way that’s hard to parse.
2. I also struggled with Table 1. The authors don't clearly state the evaluation metric. Should I be looking for higher or lower values? Why? On top of that, using four decimal places makes the table feel cluttered and actually makes it harder to see the performance gains, which don't feel particularly significant at first glance.
3. The paper mentions that the model "should incorporate the key characteristics of modern DL frameworks, such as depth, nonlinearity, and width." However, I didn't see any experiments proving which of these characteristics is actually the most effective. Is there a ranking or an ablation study showing the impact of one versus the others?
4. In Table 2, the performance jump seems very marginal. I’m not entirely convinced that these small gains justify the added complexity of the framework.

---

> ### Author Rebuttal · Authors · 2026-03-30
>
> We sincerely appreciate the valuable and constructive comments provided by the reviewer and the effort put into reviewing our work. Below, we provide a point-by-point response to the review.
>
> > **The writing overall could use some work, ..."first contribution" is worded in a way that’s hard to parse.**
>
> Thank you for your valuable comments. The "first contribution" of this work is the proposal of MTNL, which not only fuses TN with several key characteristics of modern DL but also addresses the TN-SS challenge in learning multiple TN structures.
>
> To achieve this, we propose a Bayesian learning framework that combines a novel model formulation (multiple TNs fused in a deep-network style) with a novel prior design (providing provable multiple TN-SS capability). Moreover, we demonstrate that the MAP solution of MTNL enables lightweight uncertainty estimation based on post-processing model dropout, which significantly improves uncertainty estimation efficiency.
>
> In short, the "first contribution" summarizes the overall contribution and novelty of the work. In the revised version of the paper, we will improve the paper writing to make it more structured and easier to follow.
>
> > **I also struggled with Table 1...**
>
> Thank you for your comments. The evaluation metric used in Table 1 is the relative standard error (RSE), which is a commonly used metric in the tensor recovery literature. For RSE, lower values indicate better performance.
>
> The RSE is defined as: $||\mathcal{X} - \mathcal{T}||_F \/ ||\mathcal{T}||_F$ where $\mathcal{T}$ is the ground-truth tensor (e.g., an MSI image) and $\mathcal{X}$ is the recovered tensor (image) produced by the algorithm. If the recovery is extremely good, $\mathcal{X}$ will be very similar to $\mathcal{T}$, resulting in a small RSE. Conversely, if the recovery is poor, the RSE will be large.
>
> In the following table, we reorganize the average RSE of different methods for recovering four MSI images.
>
> |Missing Rate|TRALS|BCPF|BTucker|BTT|TW|BayesTNSS|MTNL|
> |-|-|-|-|-|-|-|-|
> |95%|0.19|0.16|0.21|0.12|0.15|0.12|**0.11**|
> |85%|0.06|0.07|0.07|0.06|0.06|0.05|**0.04**|
>
> We can see that, under different missing rates, the proposed MTNL achieves superior performance compared to other methods due to the more expressive model structure introduced by MTNL.
>
> > **...I didn't see any experiments proving which of these characteristics is actually the most effective. Is there a ranking or an ablation study showing the impact of one versus the others?**
>
> Thank you for your comments. In the following table, we provide ablation studies on the effects of nonlinearity, depth, and width in the model. In the ablation, we fix the nonlinear MTNL with depth and width equal to 2 as the “center” and vary different model settings along one of the “axes” to recover the MSI image. The corresponding RSE values are shown below:
>
> Varying width
> |1|2|3|4|
> |-|-|-|-|
> |0.329|0.302|0.281|0.283|
>
> Varying depth
> |1|2|3|4|
> |-|-|-|-|
> |0.357|0.302|0.434|0.498|
>
> Ablation of nonlinearity
> |W|W/O|
> |-|-|
> |0.302|0.433|
>
> As we can see, nonlinearity is the key factor that separates MTNL from traditional linear methods. In addition, both depth and width are important for model performance. While an overly deep model can lead to overfitting, increasing the model width can alleviate this issue. In the Table 1 experiment, we use a nonlinear MTNL with depth and width equal to 2 and achieve superior performance.
>
>
> > **In Table 2, the performance jump seems very marginal. I’m not entirely convinced that these small gains justify the added complexity of the framework.**
>
> Thank you for your comments. In Table 2, the complexity of the proposed PEFT method is actually similar to other methods, as we only consider the width. The main performance gains come from learning better TN structures with the multiple TN-SS prior proposed in this work.
>
> One of the contributions of this work is tackling the multiple TN-SS challenge, which exists in tensor-based PEFT methods. MTNL can be considered an abstract underlying theoretical model that unifies tensor-based PEFT approaches and provides multiple TN-SS priors in a plug-and-play manner.
>
>
> **Thank you again for reading the rebuttal letter. In summary**, this work can be considered as providing a novel perspective in TN studies from a deep learning integration angle. This represents an elegant extension and is also necessary for handling complex tasks, as noted by reviewer LwRN.
>
> Moreover, we include substantial theoretical analysis, which not only demonstrates the flexibility of the proposed prior, producing interesting results as highlighted by both reviewers 4B11 and LwRN, but also leads to the novel finding of a practical uncertainty estimation method. Therefore, **this paper contributes to both modeling and theoretical perspectives in TN studies.**
>
> We hope this rebuttal letter can help you reevaluate the value of this paper. If you have any further questions or concerns, we are happy to discuss them.

---

> > ### Author Rebuttal · Reviewer_Uj5J · 2026-04-03
> >
> > Thank you to the authors for their response. However, I still have concerns regarding the limited performance gains in terms of RSE, and I remain unconvinced that these relatively small improvements justify the added complexity of the proposed framework. Therefore, I choose to keep my original score.
> > As I mentioned in my review, the core topic of this paper is somewhat outside my primary area of expertise, so my assessment is necessarily limited to the aspects I can reasonably evaluate.

---

> > > ### Author Response · Authors · 2026-04-04
> > >
> > > Thank you for your comments. Actually, the 0.01 RSE gain is **not a limited gain**. It has **meaningful improvement** in tensor recovery, **aligned with gain reported in studies published at top-tier conferences**, and it is also **scientifically meaningful**. In the following, we provide a more detailed discussion of these points.
> > >
> > >
> > > **Fine-grained noise removal.** As we can see from Figure 12 in the appendix of the main paper, the recovery image from MTNL is less noisy than other methods, reflected by: (1) the residual images below show the absolute difference between the recovered image and the original image. In the blue background residual images, the number and intensity of bright spots reflect the noise level of the recovered images. The MTNL residual image has fewer and weaker bright spots; (2) the recovery image of MTNL also has clearer structures and fewer artifacts.
> > >
> > > Making the recovery image less noisy is important, as it provides a clearer image for downstream tasks and improves their quality, and even after fine-grained hyperparameter tuning, the comparison methods still cannot reach a recovery quality as clean as MTNL.
> > >
> > > The reason that noise removal is not significantly reflected in the RSE is that the numerator, the Frobenius norm, is more sensitive to large signals and less sensitive to small ones. Furthermore, the denominator is the Frobenius norm of the original image, which is typically large, these factors make the effect of further noise reduction less noticeable. But in fact, a small change in RSE corresponds to a visibly clearer recovered image.
> > >
> > > **Peer Recognized.** A 0.01 RSE improvement is also a recognized gain in [1,2]. In [1],  the work similarly introduces higher complexity to TN to improve recovery performance. As shown in Figure 4 of that work, the RSE gain over the second best method is 0.006, which is even smaller than the 0.01 gain in our work.
> > >
> > > **Scientific Value.** Moreover, method such as BayesTNSS is theoretically SoTA among traditional TN methods, and it already achieve quite low RSE. In such low RSE circumstances, every further improvement is a non-trivial task. Specifically, a model must capture complex higher-order correlations aligned with the true problem bias in order to remove subtle noise. The further 0.01 improvement is evidence that a new model bias is emerging from added complexity MTNL, providing expressiveness beyond the reach of traditional methods. Such advancements in model bias enhance the ability of TNs to handle more complex problems, supporting potentially higher-impact research of TNs, which demonstrates clear scientific significance.
> > >
> > > **In summary, whether from the practical application or from a broader impact perspective, the 0.01 RSE gain justifies the added complexity of the proposed framework.**
> > >
> > > **Reference**
> > >
> > > [1]. Zheng Y B, Huang T Z, Zhao X L, et al. Fully-connected tensor network decomposition and its application to higher-order tensor completion[C]//Proceedings of the AAAI conference on artificial intelligence. 2021, 35(12): 11071-11078.
> > >
> > > [2]. Yuan L, Li C, Mandic D, et al. Tensor ring decomposition with rank minimization on latent space: An efficient approach for tensor completion[C]//Proceedings of the AAAI conference on artificial intelligence. 2019, 33(01): 9151-9158.

---

### Official Review · Reviewer_LwRN · 2026-03-18

**Soundness:** 2
**Presentation:** 2
**Significance:** 3
**Originality:** 3
**Overall Recommendation:** 3
**Confidence:** 3

**Summary:**

The work proposes a deep tensor network leaning framework named as mixed tensor network learning (MTNL) that can be leveraged for both supervised and unsupervised tensor tasks. The framework’s learning paradigm follows a Bayesian approach enforcing sparsity on core tensors and uses MAP optimization problem. The MTNL framework can be adapted for several tasks including parameter-efficient fine tuning and fast uncertainty estimation. Several experiments are presented to substantiate the claims.

**Compliance With Llm Reviewing Policy:**

Affirmed.

**Key Questions For Authors:**

Please see “Weaknesses” section. In addition, I have a few other questions.

1.	Figure 1 needs quite a lot of improvement in illustrating MTNL model. It is not technically deep in conveying the framework.

2.	“Second, while core concepts from DL have been incorporated to enhance the TN model, these attempts remain shallow and still fall short of fully exploring the insights of the modern DL framework, leaving room for further improvements.” This sentence in the introduction section reads vague, does not convey what is the shortcoming of the existing approaches employing DL in TN models.

3.	How are the threshold matrices applied? There is no mathematical relation defined relative to that.

4.	“To incorporate the model structure with input-output data pair, the order of the TN must satisfy T>= max(d,c)”. The reasoning here is unclear. Please clarify.

5.	It would be better if you can somehow simplify the notations for better readability. For example, equations 4,5, and equation after that could be represented without a lot of subscripts and superscripts for easy discussion.

6.	Theorem 2.1 and the explanation after that “However, ideally, the penalty should be applied jointly”. What is the “idea” scenario that is referred here? And why is it so?

7.	What is meant by rank cutting threshold in Table 2? What does the “iterations” refer to in Table 5? What is D parameter in Table 5? In addition, please indicate what are the hyperparameter setting in each experiment setting and what is the rationale for the choice.

**Limitations:**

yes

**Strengths And Weaknesses:**

Strengths:

1.	The framework is an elegant extension of tensor networks combining nonlinearity and sparsity needed for complex tasks.

2.	Experiments across diverse tasks demonstrate the flexibility of the framework.

3.	Theoretical results, especially Theorem 2.2, that brings out the adaptive sparsity weightage seems interesting.

Weaknesses:

1.	Several claims supporting the proposed architecture over existing methods has not been empirically demonstrated in depth. For example, sparsity prior, the choice of nonlinearity, number of layers, number of TNs in each layer may need detailed analysis on real data to understand the performance advantage in Table 1.

2.	The proposed learning framework has large number of hyperparameters such as those defined in Eq. 5, the ranks, activation function used etc. These hyperparameter selections need some guidelines in each setting, which is currently lacking in experiment section.

3.	The MAP estimation problem looks computationally intensive and hence it is important to present the comparison of runtime and computational complexity with respect to the baselines.

4.	Section 2.3 describing PEFT may need more clarity, especially in terms of the tensor dimensions in each layer. In addition, more discussion is needed how it is conceptually when compared to LoRA as applying f^l (with a nonlinear activation function) fundamentally breaks the low-rank residue.

---

> ### Author Rebuttal · Authors · 2026-03-30
>
> We sincerely appreciate the valuable comments provided by the reviewer. Below, we provide a point-by-point response to the main concerns raised in the review. For your comments, we refer to them with the corresponding indices. Due to the 5000-character limit, we try to make the response as informative as possible to assist with your evaluation. Thanks again.
>
> > **Weaknesses1: Several..**
>
> Thank you. In the following, we provide ablation studies on the sparsity prior, nonlinearity, number of layers $L$, and number of TNs in each layer $E$. In the ablation, we consider MTNL with $E,L=2$ as the ”center”, and vary different model settings around it to recover the MSI. The RSEs are as follows:
>
> Varying $E$
> |1|2|3|4|
> |-|-|-|-|
> |0.329|0.302|0.281|0.283|
>
> Varying $L$
> |1|2|3|4|
> |-|-|-|-|
> |0.357|0.302|0.434|0.498|
>
> Ablation of nonlinearity
> |W|W/O|
> |-|-|
> |0.302|0.433|
>
> Ablation of sparsity prior
> |W|W/O|
> |-|-|
> |0.302|0.439|
>
> We can observe that $L$ and $E$ are beneficial for model performance, while too deep and wide models may cause overfitting. Moreover, both nonlinearity and sparsity priors are essential for distinguishing the model from linear TN methods and for adapting model complexity to better fit the data. In the experiment, we choose MTNL with $E,L=2$, nonlinearity and sparsity priors to demonstrate superior performance.
>
> > **Weaknesses2: The..**
>
> Thank you. For the hyperparameters in Eq. 5, we pick them from four sets: {1e-6,1e-6,-2}, {1,1e-6,5}, {1,1e-6,1}, {1e-6,1,-5}, which account for a wide range of sparsity patterns. For the hyperparameters in Eq. 3 and the noise precision, we tune them in patterns such as 0.1, 0.001, ... We keep the ranks the same and tune them simultaneously to a relatively high value, since the sparsity prior can adjust the model complexity. Moreover, we mainly use the ReLU activation in the paper, as it is essential.
>
> > **Weaknesses3: The MAP..**
>
> Thank you. The computational complexity of the MTNL is provided in Appendix E, and MTNL shares a similar complexity with existing TN methods like BayesTNSS. Here, we provide the runtime of the tensor recovery experiments:
>
> | |TRALS|BCPF|BTucker|BTT|TW|BayesTNSS|MTNL|
> |-|-|-|-|-|-|-|-|
> |0.95|12|113|2075|313|29|973|2530|
> |0.85|53|173|2055|502|38|1248|1761|
>
> We can see that MTNL requires a relatively higher runtime compared to other methods and is similar to BTucker. This is mainly caused by the extended convergence process due to the nonlinearity in the model. We will focus on increasing the convergence speed in future work, while this work is the first to introduce the novel concepts for enhancing TN learning.
>
> > **Weaknesses4: Section..**
>
> Thank you. For PEFT, we first tensorized the fine-tuning weights. In LLaMA2-7B, we tensorize the weights to $64\times64\times64\times64$. Moreover, the proposed method does not break the low-rank residue. After training, the weights can still be merged first. Our model only uses a linear TN representation (sum of TNs) for the fine-tuning weights. We abstract this layer as the $l$-th layer of MTNL. We understand that the notation in Eq. 9 is misleading and will make it more accurate in the revision.
>
> > **Q1: Figure..**
>
> Thank you for your valuable suggestions. We will make Figure 1 more informative in the revision.
>
> > **Q2: Second..**
>
> Thank you. Several existing TN studies can be considered as using only parts of DL concepts, such as depth or nonlinearity. Therefore, the contribution of this paper is to “fully” fuse TN with DL to increase the expressiveness of the TN method for handling complex tasks.
>
> > **Q3: How..**
>
> Thank you. The precise mathematical description is provided in Eq. 16 in the Appendix.
>
> > **Q4: To..**
>
> Thank you. Since the model should receive the input data tensor and outputs the label tensor, the corresponding weight tensors must have higher dimensions. To simplify notation, we strictly set $T \geq \max(d, c)$. However, in practice, MTNL can have input weight tensors with dimensions higher than $d$, and output weight tensors with dimensions higher than $c$, respectively.
>
> > **Q5: It..**
>
> Thank you for your valuable suggestions.
>
> > **Q6: Theorem..**
>
> Thank you. Shrinking the unnecessary rank of a TN is equivalent to removing all the elements from the corresponding rank slice. This means ideally we should enumerate all the other indices to form a big vector, and apply a penalty to shrink it all at once. However, existing frameworks break this big vector into many smaller ones and do not apply a penalty to shrink them simultaneously, which may produce different results compared with the former approach in shrinking the TN ranks.
>
> > **Q7: What..**
>
> Thank you. The rank-cutting threshold refers to the threshold value we apply to the threshold matrix's elements. “Iterations” refer to the number of iterations in running the method. D in Table 5 denotes the mode dimension of the synthetic tensors. We describe the hyperparameter selections in response 2.

---

> > ### Author Rebuttal · Reviewer_LwRN · 2026-04-03
> >
> > Thank you for some of the clarifications. Yet, the concerns about lacking practical guidelines on choosing different hyperparameters as well as high computational complexity still remain. I also do not see that addressed in the current version of the revised manuscript. I would like to keep the score same.

---

> > > ### Author Response · Authors · 2026-04-05
> > >
> > > Thank you for your comments. In the following, we **provide detailed and comprehensive practical guidelines for choosing the core model hyperparameters in the paper, and discuss the high computational complexity.**
> > >
> > > **Practical guidelines.** First, we enumerate the core hyperparameters in the model: network depth E and width L, activation function, TN structures in MTNL, TN-ranks, noise precision $\epsilon$, $\tau$ for the core prior, a,b,c for the threshold prior, dropout rate $\theta$, and sample number $N$.
> > >
> > > As shown in the first response, nonlinearity is important. In practice, we should keep the ReLU activation. For E and L, we can tune them jointly, or fix L at 2 and only tune E, as indicated in the first response ablation study. Besides, fixing E, L = 2 is an experimentally guided choice.
> > >
> > > For the TN structures in MTNL, for image data, one can follow the settings used in this paper. More generally, for arbitrary-order data, empirically, we can group redundant modes, separate them across layers, and design the corresponding TN structures.
> > >
> > > For TN-ranks, they are adjusted jointly. In practice, we can directly tune (set) them to align the number of model parameters with other TN methods. Since our model is more expressive, this setting guarantees good performance. Alternatively, we can try several increasing integers (e.g., 5, 10, …), **common in TN studies**.
> > >
> > > For $\epsilon$, this is data-dependent. If useful information is limited (e.g., few observed entries), $\epsilon$ should be small (e.g., 0.1, 0.01, …). Otherwise, larger values (e.g., 1, 10, …) can be used. For $\tau$, it controls regularization and is also related to the amount of available information. If information is limited, a large $\tau$ (e.g., 1, 5, …) is preferred. Otherwise, smaller values (e.g., 0.1, 0.05, …) can be used. Both $\epsilon$ and $\tau$ are common in ML models and can **follow standard selection practices**.
> > >
> > > For {a,b,c}, in practice, we can first perform a coarse search using the four sets provided in the first response to find a suitable sparsity pattern. Then, a fine-grained sparsity level search can be conducted. As shown in the ablation experiments for reviewer 4B11, c mainly controls sparsity. If the problem has limited information, smaller values (e.g., -5, -15, …) can be used. Otherwise, larger values (e.g., 5, 15, …) are preferred. This strategy reduces the tuning from combinatorial to linear complexity.
> > >
> > > Hyperparameters chosen from the above settings allow MTNL to achieve superior performance. By **practically avoiding joint tuning**, we only need to tune along five “axes”.  Traditional TN methods typically involve three, the additional axes in MTNL are {E,L} and {a,b,c}, which are **reasonable add complexity. If we fix E,L, the additional axis is reduced to only one**.
> > >
> > > For $\theta$ and the sample number $N$, they are determined using a validation set with NLL. For example, test $\theta$ from 0.1 to 0.9 and $N$ 50, 100, 150, etc. This process is **nearly cost-free**, as it is lightweight.
> > >
> > > **High computational complexity.** As analyzed in the first response, the longer running time mainly comes from the extended convergence process. Here, we show that the higher running time can be reduced. Specifically, in the original setting, E=2, which complicates the optimization dynamics, we can set it to 1. Moreover, we use a larger learning rate. Through this empirical improvement, MTNL can reach the same RSE while converging much faster. **The running time is now 1236 s (MR = 0.95) and 996 s (MR = 0.85). MTNL even requires 252 s less average running time in the 0.85 recovery case compared to BayesTNSS.**
> > >
> > > **The contribution of the work.**
> > > This work represents a novel attempt to fuse TN with DL. Although we perform fine-grained tuning for traditional TN methods, their performance still cannot match MTNL. This reflects a limitation of traditional methods in capturing complex higher-order correlations in real-world data. MTNL introduces a new paradigm (model bias) in TN research to address this issue, recognized as a novel direction by reviewer hC77. Breaking these limitations **inevitably involves additional complexity, but the gain is meaningful.**
> > >
> > > The focus of this work is to **first explore** this new paradigm and **demonstrate its effectiveness**. **Improving efficiency is a promising** future direction, since there are many optimization techniques from TN and DL that can accelerate convergence and enable tuning-free. Moreover, in the high computational complexity section in this response, we have shown that an empirical adjustment can already significantly speed up convergence and reduce running time to a level comparable with existing methods.
> > >
> > >
> > > **The revised manuscript.** During this phase, ICML **does not support uploading a revised manuscript**. However, we have provided a clear description of the practical guidelines for hyperparameter selection and a discussion of computational complexity in this response.

---

### Decision · Program_Chairs · 2026-04-30

**Decision:**

Accept (regular)

**Comment:**

This work proposes the mixed-tensor network learning (MTNL) method, which
combines Bayesian tensor network learning with deep networks.  The authors give
a dropout-based algorithm with theoretical guarantees and a comprehensive
empirical study (tensor completion, tensor regression, and LLM applications).

**Decision.**
There is some discrepancy in the reviewer recommendations, but on average the reviews lean towards "Weak Accept" (including a high-confidence strong accept).